# The effects of morphology, mobility size and SOA material coating on the ice nucleation activity of black carbon in the cirrus regime

Cuiqi Zhang[1,2], Yue Zhang[3,4,5], Martin J. Wolf[2], Leonid Nichman[6], Chuanyang Shen[2,7], Timothy B. Onasch[4,5], Longfei Chen[1], and Daniel J. Cziczo[2,8,9]

[1]School of Energy and Power Engineering, Beihang University, Beijing, China
[2]Department of Earth, Atmospheric, and Planetary Sciences, Massachusetts Institute of Technology, Cambridge, MA 02139, United States
[3]Department of Environmental Sciences and Engineering, University of North Carolina at Chapel Hill, Chapel Hill, NC 27599, United States
[4]Aerodyne Research Incorporated, Billerica, MA 01821, United States
[5]Department of Chemistry, Boston College, Chestnut Hill, MA 02467, United States
[6]National Research Council Canada, Flight Research Laboratory, Ottawa, ON, K1V 9B4, Canada
[7]Department of Atmospheric and Oceanic Sciences, Peking University, Beijing, China
[8]Department of Civil and Environmental Engineering, Massachusetts Institute of Technology, Cambridge, MA 02139, United States
[9]Department of Earth, Atmospheric, and Planetary Sciences, Purdue University, West Lafayette, IN 47907, United States

*Correspondence to*: Longfei Chen (chenlongfei@buaa.edu.cn)

**Abstract.** There is evidence that black carbon (BC) particles may affect cirrus formation and hence global climate by acting as potential ice nucleating particles (INPs) in the troposphere. Nevertheless, the ice nucleation (IN) ability of bare BC and BC coated with secondary organic aerosol (SOA) material remains uncertain. We have systematically examined the IN ability of 100-400 nm size-selected BC particles with different morphologies and different SOA coatings representative of anthropogenic (toluene and *n*-dodecane) and biogenic (*β*-caryophyllene) sources in the cirrus regime (-46 to -38 °C). Several BC proxies were selected to represent different particle morphologies and oxidation levels. Atmospheric aging was further replicated with exposure of SOA-coated BC to OH. The results demonstrate that the 400 nm hydrophobic BC types nucleate ice only at or near the homogeneous freezing threshold. Ice formation at cirrus temperatures below homogeneous freezing thresholds, as opposed to purely homogeneous freezing, was observed to occur for some BC types between 100-200 nm within the investigated temperature range. More fractal BC particles did not consistently act as superior INPs over more spherical ones. SOA coating generated by oxidizing *β*-caryophyllene with $O_3$ did not seem to affect BC IN ability, probably due to an SOA phase state transition. However, SOA coatings generated from OH oxidation of various organic species did exhibit higher IN onset supersaturation ratio with respect to ice ($SS_i$) compared with bare BC particles, with toluene SOA coating showing an increase of $SS_i$ by 0.1-0.15 while still below the homogeneous freezing threshold. Slightly oxidized toluene SOA coating seemed to have a stronger deactivation effect on BC IN ability than highly oxidized toluene SOA, which might be caused by oligomer formation and phase state transition of toluene SOA under different oxidation levels. *n*-dodecane and *β*-caryophyllene-derived SOA coated BC only froze in the homogeneous regime. We attribute the inhibition of IN ability to the filling of the pores on the BC surface by the SOA material coating. OH exposure levels of *n*-dodecane and *β*-caryophyllene

SOA coating experiments, from an equivalent atmospheric 10 to 90 days, did not render significant differences in IN potential. Our study of selected BC types and sizes suggests that increase in diameter, compactness, and/or surface oxidation of BC particles lead to more efficient IN via pore condensation freezing (PCF) pathway, and that coatings of common SOA materials can inhibit the formation of ice.

## 1 Introduction

Cirrus clouds affect the global energy balance predominantly by more effectively trapping long-wave terrestrial radiation than reflecting solar energy (e.g., Kärcher et al., 2007; Heymsfield et al., 2017; Kärcher, 2018). In cirrus clouds, ice crystals can form via two pathways, i.e. homogeneous and heterogeneous ice nucleation (IN) (Pruppacher and Klett, 2010). Homogeneous freezing is the spontaneous freezing of solution droplets without any foreign surfaces aiding the process (Pruppacher and Klett, 2010). Heterogeneous IN occurs more readily than homogeneous IN due to the presence of an ice nucleating particle (INP) at a lower supersaturation with respect to ice ($SS_i$) or warmer temperature (DeMott et al., 2003; Vali et al., 2015; Kanji et al., 2017). Deposition IN is one classical heterogeneous IN mode, in which solid ice is formed by direct water vapor deposition on to an INP surface. Recently, laboratory studies demonstrated that ice formation at thermodynamic conditions relevant to classical deposition IN on porous material at cirrus temperature (below -38 °C) might actually be initiated by homogeneous freezing of liquid water held within the cavities below water saturation due to inverse Kelvin effect (Marcolli, 2014; David et al., 2019; David et al., 2020). This pathway by which porous material might form ice below water saturation below -38 °C is referred to as pore condensation and freezing (PCF, Marcolli, 2014).

Black carbon (BC) particles from aircraft emissions may be an important direct source of anthropogenic INPs to the tropopause (Petzold et al., 1998; Seinfeld, 1998; Popovicheva et al., 2004; Burkhardt and Kärcher, 2011; Kärcher, 2018). Global mass-based aviation BC emission rates are estimated to range between 2-20 Gg year$^{-1}$ (Bond et al., 2004; Lee et al., 2010; Bond et al., 2013; Zhang et al., 2019a), while the number-based aviation BC emission rate is estimated to be equivalent to ~1.3 % of total ground anthropogenic BC emissions (Zhang et al., 2019a). Aviation fuel usage is projected to increase 2-4 fold in the next few decades (Lee et al., 2009; Lee et al., 2010), simultaneously increasing aircraft-induced cloudiness (Petzold et al., 1998; Seinfeld, 1998; Popovicheva et al., 2004; Burkhardt and Kärcher, 2011; Kärcher, 2018). However, the role of BC aerosol-cloud-climate interactions in cirrus formation remains highly uncertain (IPCC, 2013).

Laboratory experiments have been carried out to study the effects of isolated processes on BC IN ability in detail. Both well-characterized commercially available BC (e.g., DeMott et al., 1999; Fornea et al., 2009; Brooks et al., 2014; Mahrt et al., 2018; Nichman et al., 2019) and soot particles from combustion sources (e.g., Diehl and Mitra, 1998; Möhler et al., 2005b; Dymarska et al., 2006; Kanji and Abbatt, 2006; Koehler et al., 2009; Crawford et al., 2011; Friedman et al., 2011; Kanji et al., 2011; Kulkarni et al., 2016; Mahrt et al., 2018; Nichman et al., 2019) have been used to investigate the IN ability of BC particles, with a particular focus on IN below -38 °C. According to previous studies (e.g., Koehler et al., 2009; Friedman et

al., 2011; Kulkarni et al., 2016; Mahrt et al., 2018; Nichman et al., 2019), the following physicochemical properties of particles may play vital roles in determining BC IN activity: a) mobility diameter ($d_m$), b) morphology, c) surface oxidation state, and d) organic material coating. It is widely acknowledged that larger particles act as more efficient INPs (e.g., Pruppacher and Klett, 2010), which is also confirmed by recent heterogeneous IN experiments with various INPs (e.g., Welti et al., 2009; Lüönd et al., 2010; Marcolli, 2014; Mason et al., 2016; Mahrt et al., 2018; Nichman et al., 2019). Although the mechanism remains uncertain, one common theory is that IN ability and rate are positively correlated to particle surface active sites density (Fletcher, 1960, 1969), an empirical parameter that is relevant to particle size (e.g., Connolly et al., 2009; Kiselev et al., 2017). Similarly, the probability of a BC aggregate to contain a pore with the right properties (e.g., pore size and surface hydrophilicity) increases with increasing aggregate diameter, which would favor PCF for larger particles (Mahrt et al., 2018). The IN ability of monodisperse BC particles with the size range of 100-800 nm has previously been characterized (Koehler et al., 2009; Friedman et al., 2011; Kulkarni et al., 2016; Mahrt et al., 2018; Nichman et al., 2019). The lower size limit at which BC particles act as active INPs below -38 °C varied between 100 nm and 400 nm. However, the size threshold below which BC cannot nucleate ice at thermodynamic conditions relevant to classical deposition mode at cirrus temperature and the underlying mechanism is still uncertain.

Laboratory experiments and field observations confirmed that BC morphology and surface chemistry may change significantly during atmospheric aging, leading to changes in particle surface area, shape, and chemical composition (e.g., Slowik et al., 2007; Khalizov et al., 2009; Tritscher et al., 2011; Fu et al., 2012; China et al., 2015b; Li et al., 2016; Moffet et al., 2016; Li et al., 2017; Wang et al., 2017; Bhandari et al., 2019). Commonly-used BC morphology characteristics are those derived from 2-D projected electron microscopy images, including fractal dimension ($D_f$), roundness, aspect ratio ($AR$), and convexity (e.g., Ramachandran and Reist, 1995; Lee and Kramer, 2004; China et al., 2013; China et al., 2014; China et al., 2015b; Kulkarni et al., 2016). Effective density and surface area have also been utilized to reflect BC morphology and mixing state (Tritscher et al., 2011; Kulkarni et al., 2016; Mahrt et al., 2018; Nichman et al., 2019).

Freshly emitted BC particles are typically hydrophobic, fractal, nanoscale (<200 nm) aggregates with a branched or chain-like structure (e.g., Kinsey et al., 2010; Beyersdorf et al., 2014; Liati et al., 2014; Vander Wal et al., 2014; Lobo et al., 2015; Moore et al., 2017). BC aggregate surface area is determined by primary particle sizes, number of primary particles, and the way primary particles are connected (Kittelson, 1998). Nichman et al. (2019) reported a generally positive correlation between BC particle surface area and IN activity for particles with same size. For smaller particles, Mahrt et al. (2018) presented a complex dependence of BC IN activity on particle size, surface area, and BC surface hydrophilicity. They attributed BC IN activity to PCF mechanism (Marcolli, 2014; David et al., 2019; David et al., 2020), in which IN of BC is considered as homogeneous freezing of liquid water taken up in mesopores (2-50 nm) due to capillary effect (Fisher et al., 1981).

The surface chemistry of the emitted particles is governed by the source and the host environment in which the particles evolve. Nascent BC particles can interact with volatile species such as sulfates and unburnt hydrocarbons in the aircraft cooling

exhaust plume and grow (e.g., Lefebvre, 1998; Onasch et al., 2009; Anderson et al., 2011; Kärcher, 2018). These particles can remain suspended in the atmosphere for days to weeks (Cape et al., 2012; Lund et al., 2018), or even months in the tropopause (Pusechel et al., 1992; Yu et al., 2019), during which the exposure to atmospheric biogenic and anthropogenic emissions, as well as oxidation, can lead to complex secondary organic aerosol (SOA) coatings (Jacobson, 2001; Zhang et al., 2008; China et al., 2015b; Kulkarni et al., 2016; Zhang et al., 2018b). Numerous experiments have been conducted to investigate the effects of surface coating on BC IN ability. Hygroscopic BC particles (Koehler et al., 2009), or BC particles coated by hygroscopic materials, such as sulfuric acid (DeMott et al., 1999; Möhler et al., 2005b; Crawford et al., 2011), water-soluble organic acids (Friedman et al., 2011; Nichman et al., 2019) and SOA (Kulkarni et al., 2016), tended to enhance BC water uptake ability and form aqueous solutions on BC surface, moving IN onset $SS_i$ towards the homogeneous freezing threshold. Hydrophobic organic coatings tended to impede surface interaction between BC and water molecules. Möhler et al. (2005b), Crawford et al. (2011), and Mahrt et al. (2018) reported a transition from heterogeneous to homogeneous freezing mode for combustion BC with increasing OC content. Ozone (Friedman et al., 2011) and hydroxyl (OH) radical (Chou et al., 2013; Kulkarni et al., 2016) oxidation can change surface functional groups of BC particles and enhance hydrophilicity, but no distinguishable BC IN activity change has been observed. Despite these previous efforts, the influence of particle morphology, chemistry, and aging, as well as the microphysical mechanism behind BC IN ability, remains ambiguous.

In this work, we examine the effects of particle mobility diameter, morphology, and SOA coating on the IN ability of several aerosolized BC proxies as a function of $SS_i$ in a cirrus relevant temperature regime (from -46 °C to -38 °C). Representative species of anthropogenic (toluene and *n*-dodecane) and biogenic (*β*-caryophyllene) volatile organic compounds were chosen to simulate potential photochemical atmospheric aging processes of BC. Different aging durations in equivalent atmospheric times were simulated by controlling the OH radical exposure. Our results help to clarify the effects of physicochemical properties and SOA formation on BC IN ability and cirrus formation in the upper troposphere.

## 2 Experimental: materials and methods

### 2.1 Materials

#### 2.1.1 Black carbon samples

Three types of commercially available BC particles (Raven 2500 Ultra, hereafter R2500U, Birla Carbon U.S.A., Inc.; REGAL 330R, hereafter R330R, Cabot Corporation; and CAB-O-JET 300, hereafter COJ300, Cabot Corporation Inkjet Colorants Division), corresponding to different surface chemistry and morphology regimes were studied as proxies of atmospheric BC. Table 1 summarizes the characteristics of these BC proxies. R2500U and R330R are carbonaceous black pigment powder generated by incomplete combustion (Joyce and Henry, 2006; Cabot). COJ300 is a highly dispersible ink due to the 4-carboxyphenyl-modified surface (Johnson, 1999). COJ300 is selected for its high degree of oxidation, which is confirmed by the Particle Analysis by Laser Mass Spectrometry (PALMS) chemical analysis (see Fig. A1), classifying it as

the most oxidized BC proxy in this study. R2500U and R330R are unoxidized but differ in morphology, which was confirmed by morphology characterization and PALMS analysis (see Sect. 3.1). R2500U, R330R, and COJ300 were chosen as proxies of freshly emitted BC, atmospheric compacted BC, and atmospheric oxidized BC, respectively. IN properties of 800 nm R330R and R2500U particles were previously studied (Nichman et al., 2019). This work addresses the remaining questions raised in the previous study and focuses on the impact of particle size, morphology, and surface oxidation.

**2.1.2 SOA coating materials**

      Three organic species, toluene, *n*-dodecane, and *β*-caryophyllene, were selected to represent atmospheric SOA precursors from anthropogenic and biogenic sources (Table S1). Toluene and *n*-dodecane are often selected as surrogate jet fuel components to investigate combustion and emission characteristics because they have been proven well-suited to represent tens of hundreds of components found in mainstream jet fuels (e.g., Dooley et al., 2010; Dooley et al., 2012; Zhang et al.,
2016; Zhao et al., 2017). Field aircraft emission studies also confirm the presence of these unburnt aliphatic and aromatic organic compounds in particles of aircraft engine exhaust (e.g., Pison and Menut, 2004; Kinsey et al., 2011; Beyersdorf et al., 2012; Timko et al., 2014). These organic compounds may coat BC particles, forming BC-containing aerosols in engine plume. Moreover, toluene is considered a dominant aromatic SOA precursor due to anthropogenic activities (e.g., Pandis et al., 1992) and serves as a proxy for other light aromatic species (such as xylenes, alkylbenzenes, naphthalene, etc.) in atmospheric
aromatic-seeded SOA formation models (e.g., Hildebrandt Ruiz et al., 2015). *n*-dodecane is one of the most studied long-chain aliphatic SOA precursor (e.g., Presto et al., 2010; Yee et al., 2013; Loza et al., 2014) representing less volatile aliphatic species. Modelling and field studies suggest that such less volatile organic species might be significant anthropogenic SOA precursors in highly populated area (Hodzic et al., 2010; Tsimpidi et al., 2010; Lee-Taylor et al., 2011; Hodzic et al., 2016), among which *n*-dodecane has relatively higher emission rate (Lee-Taylor et al., 2011). Terpenes are biogenic organics emitted by plants,
among which *β*-caryophyllene has been found to be one of the most atmospherically abundant sesquiterpenes originating from agricultural plants and pine trees, as well as other sources (Arey et al., 1991; Ciccioli et al., 1999; Helmig et al., 2006; Sakulyanontvittaya et al., 2008; Guenther et al., 2012; Henrot et al., 2017). Even though the atmospheric abundance of *β*-caryophyllene is not as significant as other biogenic organics, such as isoprene and α-pinene, its high reactivity towards ozone and hydroxyl radical to form oxidized products with low volatility makes *β*-caryophyllene an appreciable biogenic SOA source
in the atmosphere (Shu and Atkinson, 1995; Calogirou et al., 1997; Hoffmann et al., 1997; Griffin et al., 1999; Helmig et al., 2006; Lee et al., 2006; Jaoui et al., 2007). An up to 7 ng m$^{-3}$ atmospheric concentration of *β*-caryophyllene tracer during summer time was reported (Jaoui et al., 2007).

## 2.2 BC particle generation and characterization

### 2.2.1 BC particle generation

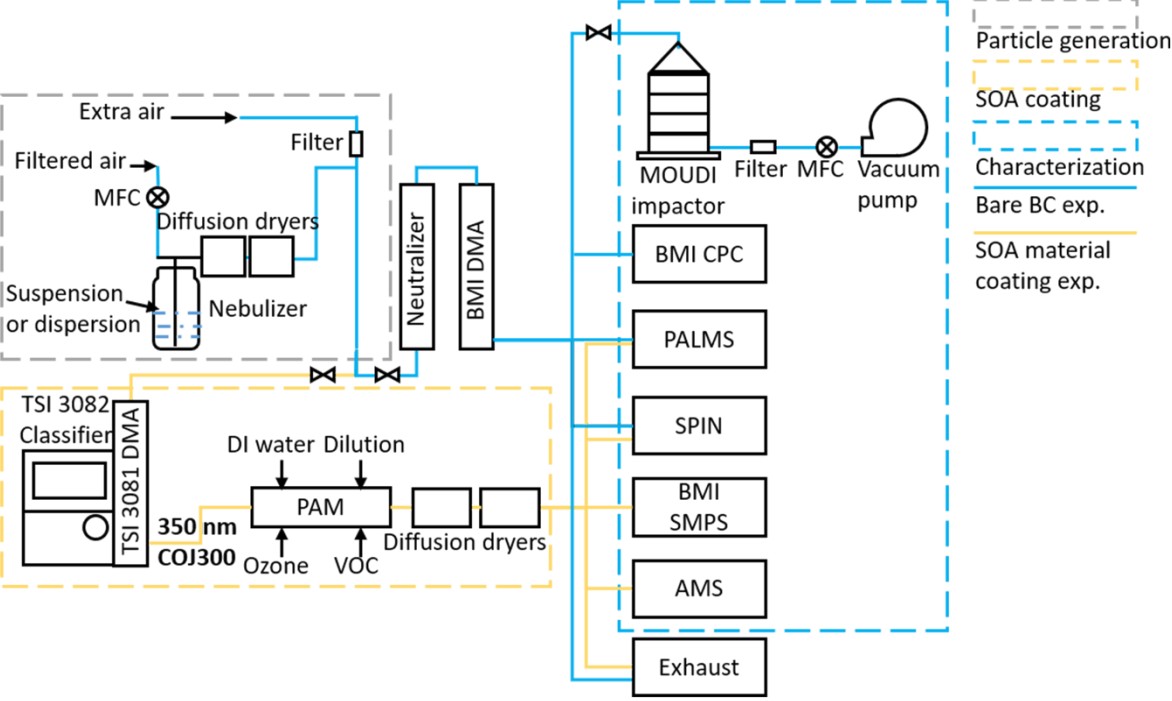


**Figure 1. Schematic diagram of the experimental apparatus for bare BC particles (blue lines) and organic SOA coating experiments (yellow lines). The grey dashed box encloses the particle generation section, which is used for both bare BC and organic SOA coating experiments. The yellow dashed box denotes the SOA coating section. The blue dashed box is the aerosol characterization and test section.**

Figure 1 shows a schematic diagram of the experimental apparatus used in this study. The particle generation setup is enclosed in the grey dashed box. Suspensions of R2500U and R330R, as well as a diluted COJ300 dispersion (dilution ratio 1:30) were atomized with a 3-jet collision nebulizer (CH Technologies (USA), Inc.), and bare BC experiments are marked by blue lines in Fig. 1. Suspensions of BC powder (R2500U and R330R) were prepared by mixing 1 g BC powder with 100 mL de-ionized (DI) water. The mixture was then sonicated for 10 minutes to make the suspension more uniform. The flow rates

through the nebulizer was 1.5 SLPM (standard liters per minute) and 2.2 SLPM for bare BC experiments and BC-SOA mixing experiments, respectively, which is controlled by a mass flow controller (MFC, Model MC-2SLPM-D; ALICAT Scientific). The atomized BC particles were dried by passing them through two consecutive 43 cm silica gel diffusion dryers (DDU 570/H, Topas). All samples were then neutralized and size selected. For bare BC experiments, a BMI differential mobility analyzer (BMI DMA, Model 2002; Brechtel Manufacturing Inc.) was used to size-select particles, with a 500 nm cut-off size impactor

installed at the DMA inlet to get rid of large particles. For BC-SOA mixing experiments, a TSI DMA (Model 3081, Classifier, Model 3082; TSI Inc.) was utilized to select 350 nm COJ300 particles with no impactor applied. The sheath-to-sample ratios

for bare BC and BC-SOA mixing experiments were respectively ~6:1 and 4:1. Compared with the widely used sheath-to-sample ratio (10:1, Karlsson and Martinsson, 2003), the lower ratios in our experiments might broaden the BC particle size distribution, yet still offering satisfactory number concentration at the target particle size (Fig. 2) because of the large particle
size being selected (Karlsson and Martinsson, 2003). The relative humidity ($RH$) of the aerosol stream entering the DMA measured by the BMI built-in $RH$ sensor was ~16 %. During the bare BC IN experiments, the size-resolved particle number concentration was monitored with a BMI condensation particle counter (BMI CPC, Model 1700; Brechtel Manufacturing Inc.).

### 2.2.2 Characterization of BC morphology

The 200 nm, 300 nm and 400 nm R2500U, and 400 nm R330R and COJ300 BC particles were collected on 300-mesh
carbon film copper grids (Ted Pella, Inc.) with a Micro-Orifice Uniform Deposit Impactor (MOUDI, Model M135-10; TSI Inc.) for offline morphology analysis. The flow rate through the impactor was controlled by a MFC (Model MC-5SLPM-D; ALICAT Scientific) at 2 SLPM so that the cut-off size of the impactor was 100 nm. The samples were analyzed offline in a Zeiss Merlin High-resolution Scanning Electron Microscopy (HRSEM; Carl Zeiss Microscopy GmbH).

**Table 1.** Characteristics of selected BC proxies in this study. $a_{BET-N_2}$ is the BET specific surface area based on $N_2$ adsorption
isotherms; $d_m$ is the particle mobility diameter; $\overline{d_a}$ denotes the mean 2-D projected area-equivalent aggregate diameter derived from SEM images; mean aspect ratio ($\overline{AR}$), roundness ($\overline{Roundness}$) and circularity ($\overline{Circularity}$) are the geometric mean morphology parameters derived from several aggregates and are defined in Sect. 2.2.2; $\overline{d_{pp}}$ denotes the mean geometric diameter of primary particles measured from SEM images, and $N$ the number of primary particles analyzed for each BC type and size; $D_f$ denotes the 3-D fractal dimension derived from 2-D SEM images; $d_{va}$ is the particle vacuum aerodynamic diameter
measured by the PALMS; values in parentheses are the corresponding one standard deviations; values in brackets are the 95% confidence intervals of $D_f$.

| BC type | R2500U | COJ300 | R330R |
|---|---|---|---|
| Composition | Furnace black | (4-carboxyphenyl)-modified carbon black[a] | Furnace black |
| CAS No. | 1333-86-4 | 1106787-35-2 | 1333-86-4 |
| Specific gravity (20 °C) | 1.7-1.9[a] | 1.07 (dispersion)[a] | 1.7-1.9[a] |
| Bulk density (g/cm³) | 20-380 | - | 20-380 |
| pH | 7.0[b]; 4-11[c] | 7.0-8.6[a] | 6.9[b]; 2-11[c] |
| Solubility | Insoluble | Insoluble but dispersible | Insoluble |
| $a_{BET-N_2}$ (m²/g)[a,d] | 270 | 200 | 90 |

| | | | | | |
|---|---|---|---|---|---|
| $d_m$ (nm) | 200 | 300 | 400 | 400 | 400 |
| $\overline{d_a}$ (nm) | 316.9 | 403.5 | 343.5 | 629.4 | 816.6 |
| ($\pm 1\sigma$) | (109.3) | (82.5) | (106.3) | (308.3) | (355.3) |
| $\overline{AR}$ | 1.22 | 1.36 | 1.44 | 1.19 | 1.33 |
| ($\pm 1\sigma$) | (0.16) | (0.27) | (0.29) | (0.17) | (0.28) |
| $\overline{Roundness}$ | 0.81 | 0.77 | 0.73 | 0.84 | 0.75 |
| ($\pm 1\sigma$) | (0.06) | (0.08) | (0.09) | (0.08) | (0.10) |
| $\overline{Circularity}$ | 0.78 | 0.64 | 0.61 | 0.72 | 0.53 |
| ($\pm 1\sigma$) | (0.18) | (0.14) | (0.15) | (0.20) | (0.16) |
| $\overline{d_{pp}}$ (nm) | 41.9 | 35.5 | 34.5 | 34.2 | 45.4 |
| ($\pm 1\sigma$) | (12.4) | (9.9) | (11.4) | (9.9) | (13.6) |
| $N$ | 242 | 256 | 343 | 139 | 251 |
| $D_f$ | 2.02 | 1.92 | 1.92 | 2.34 | 2.31 |
| [95% confidence interval] | [1.85, 2.18] | [1.30, 2.53] | [1.68, 2.16] | [2.12, 2.56] | [2.01, 2.61] |
| Median $d_{va}$[e] | - | - | 608.7 | 610.6 | - |
| Effective density (g/cm$^3$)[f] | - | - | 1.52 | 1.44 | - |
| Spectra percentage exhibiting m/z = 16 signal | - | - | 2.8 | 21.0 | - |
| Mean O:C ratio[g] | - | - | 0.008 | 0.024 | - |
| ($\pm 1\sigma$) | | | (0.024) | (0.036) | |

[a]Information offered by manufacturer datasheet. [b]Measured by Nichman et al. (2019) using VWR pH meter. [c]Measured by manufacturer in compliance with ASTM 1512. [d]BET specific surface area measured by manufacturers using $N_2$ adsorption in compliance with ASTM D-4820. [e]Converted from the measured time of flight. [f]Calculated from dividing median $d_{va}$ by $d_m$ (400 nm in this study) and times the reference density 1 g/m$^3$ (Cziczo et al., 2006). [g]Calculated from PALMS spectra area.


Table 1 summarizes the morphological characteristics, including the projected area-equivalent diameter ($d_a$), aspect ratio ($AR$), roundness, circularity, and 3-D fractal dimension ($D_f$), for different BC types and sizes derived from high resolution SEM images ($\times 30,000$ to $\times 150,000$). Primary particle diameter ($\overline{d_{pp}}$) is the geometric average of the length and width of a clear primary particle (see Fig. A3 and A4 as well as the text in appendix A for more details). $d_a = \sqrt{4A_a/\pi}$ is the diameter of a spherical aggregate that has the same projected area ($A_a$) as the BC aggregate (China et al., 2014). $AR = L_{max}/W_{max}$ is the ratio between the longest dimension ($L_{max}$) of an aggregate periphery to the perpendicular maximum width ($W_{max}$, Fig. A2). $Roundness = \sqrt{4A_a/\pi L_{max}{}^2}$ is used as a BC aggregate shape descriptor (e.g., China et al., 2013; China et al., 2015b; Kulkarni et al., 2016). Both $AR$ and roundness are used to represent shape deviation from a circle, whose $AR$ and roundness equal 1. $Circularity = 4\pi A_a/p^2$ is a parameter used to describe the rugged level of an aggregate periphery, with rugged irregular periphery causing circularity smaller than 1. $D_f$ depends on primary particle number ($N$) and radius of gyration ($R_g$) of the aggregate (Mandelbrot, 1982). By using an ensemble approach, $N$ is found to be scaled with $(A_a/A_p)^{1.09}$, where $A_a$ and $A_p$ are projected area of aggregate and primary particles, respectively (Samson et al., 1987; Köylü et al., 1995; Oh and Sorensen, 1997; China et al., 2014). The approximate relation $L_{max}/2R_g = 1.50 \pm 0.05$ iss used to substitute $R_g$, (Brasil et al., 1999), and yield $k\,(L_{max}/\overline{d_p})^{D_f} = (A_a/A_p)^{1.09}$. $D_f$ can then be derived by a power law fit of scattered points between $L_{max}/\overline{d_p}$ and $(A_a/A_p)^{1.09}$ for each aggregate (Fig. A4). 400 nm R2500U is more fractal than COJ300 and R330R with the same $d_m$, as well as 300 nm R2500U, which is indicated by $D_f$. Project area, as well as the derived $d_a$, are significantly affected by the degree of fractal, since highly fractal particles can have voids affecting project area. COJ300 and R330R, as well as 300 nm R2500U particles, are more spherical and compact than the fractal 400 nm R2500U, leading to larger $d_a$ in comparison with 400 nm R2500U.

**2.2.3 Chemical composition characterization of single BC particle**

Qualitative chemical composition of size-selected BC particles was determined by PALMS. The detailed description of PALMS can be found elsewhere in literature (Cziczo et al., 2006; Zawadowicz et al., 2015). PALMS is an online single particle mass spectrometer in which inlet particles are first aligned by an aerodynamic lens. Two Nd:YAG green (532 nm) laser beams separated by 33.6 mm are arranged at the bottom of the inlet, measuring particle velocity based on time gap between the scattering signals. The velocity can be converted into vacuum aerodynamic diameter ($d_{va}$) from the measured time of flight (Cziczo et al., 2006, Fig. A5). A 193 nm ultraviolet (UV) excimer laser is then triggered, ablating and ionizing the particle. The ions of both refractory and volatile particle components are classified based on their mass to charge ($m/z$) ratio. PALMS provides either positive or negative polarity spectra for each particle. Particle ionization is often not quantitative. However, average ion ratios across many spectra allows a qualitative compositional comparison between two similar aerosol populations. Hundreds of spectra were collected for each soot sample to account for ionization difference caused by particle orientation difference (Murphy et al., 1998).

Chemical composition of the SOA-coated BC particle stream was analyzed online by PALMS and a High-Resolution Time-of-Flight Aerosol Mass Spectrometry (HR-ToF-AMS; Aerodyne Research Inc.). More details about the AMS can be found in literatures (DeCarlo et al., 2006; Onasch et al., 2012), here a brief introduction will be given. The AMS offers quantitative average mass spectrum of an ensemble of aerosols. Particles entering AMS first go through an aerodynamic lens inlet to form a particle beam. A mechanical chopper is used downstream the inlet to control sampling particle or particle free period. The AMS employs a heated 600 °C tungsten surface to vaporize nonrefractory aerosols. Ionization is achieved using a universal 70 eV electron ionization technique. Ionized species are detected by time of flight mass spectrometry.

## 2.3 SOA material coating on BC particles

The 350 nm COJ300 BC was chosen to be the seed particle in all SOA coating experiments because of its effective IN activity as well as its higher particle concentration (~$1 \times 10^6$ # $L^{-1}$) at the selected size in comparison with other BCs (1-3 × $10^4$ # $L^{-1}$).

Particle generation during the SOA coating experiments was identical to bare BC experiments. The SOA coating experimental setup section is enclosed in the yellow dashed box of Fig. 1. As stated above, COJ300 BC particles were nebulized in an air flow of 2.2 SLPM and dried in two consecutive 43 cm silica gel diffusion dryers, and then 350 nm BC particles were size-selected by a TSI 3081 DMA with no impactor applied, and were directed to a potential aerosol mass (PAM) oxidation flow chamber (Kang et al., 2007; Lambe et al., 2011a; Liu et al., 2018). In the PAM reactor, gas phase volatile organic compound (VOC) reacts with OH radical and/or $O_3$ (Lambe et al., 2011a; Zhang et al., 2018a), and subsequently form SOA-coated BC particles. All flow rates were controlled by MFCs. The PAM chamber was operated at 4.4 SLPM total flow rate, including 2.2 SLPM BC aerosol flow, 1.0 SLPM $O_3$ carrier flow, 0.7 SLPM VOC carrier flow and 0.5 SLPM humidified air. The residence time of particles in PAM under such flow condition was approximately 260 s. $O_3$ was generated by irradiating 1.0 SLPM dry air through an external mercury lamp ($\lambda$= 185 nm, AnaLamp low pressure Hg lamp; BHK Inc.) with a concentration of 110 ppm inside the PAM chamber in our study (Lambe et al., 2011b). The typical $O_3$ mixing ratio in tropopause ranges between 0.1 to 1 ppm (Fioletov, 2008; Gettelman et al., 2011). The ozone concentration in this study was higher than ambient concentration to expediate the reaction given the short residence time of SOA within the PAM reactor (Lambe et al., 2011b; Zhang et al., 2015). 0.5 SLPM humidified air was introduced into the chamber to react with the oxygen radical and produce OH radicals, with four mercury lamps ($\lambda = 254$ nm; BHK Inc.) mounted in Teflon-coated quartz cylinders inside the chamber to irradiate $O_3$ and produce oxygen radical (O($^1$D)) via the UV photolysis reaction of $O_3$ first: $O_3 + h\nu \rightarrow O_2 + $ O($^1$D), O($^1$D) + $H_2O \rightarrow 2OH$. The OH radical concentration can be varied by changing the four lamps' voltage. Two voltage levels, i.e. 10 V and 3V, were tested in this study (indicated in Table 2 as suffixes *-10* and *-3*), corresponding to different OH exposure levels and atmospheric aging time, ~10-15 days and ~70-90 days based on previous calculation (Lambe et al., 2011a). The equivalent atmospheric aging time $t_{eq}$ were calculated using the OH concentration of the PAM reactor $c_{OH\_PAM}$, the residence time of the particles within the PAM, $t$, and the ambient OH concentration $c_0 = 1.0 \times 10^6$ $cm^{-3}$ (Li et al., 2018) following the

equation $t = c_{OH,PAM} \times t/c_0$. $c_{OH\_PAM}$ was calculated based on $O_3$ concentration and UV intensity, and RH measured real-time from the PAM reactor (Lambe et al., 2011a; Zhang et al., 2018a). The VOC was injected into a heated bulb by a syringe pump and mixed with 0.7 SLPM dry air. The particle size distributions downstream of the PAM were measured by a BMI scanning mobility particle sizer (BMI SMPS, comprising a Model 2002 DMA and a Model 1700 CPC; Brechtel Manufacturing Inc.). The injection rate was controlled so that the modal size of the particles shifted from 350 nm bare BC particles to 400 nm SOA-coated BC particles, as illustrated in Fig. 2. The 400 nm SOA-coated BC particles were then dried to ~16% *RH* and kept below 25% by passing through two consecutive 43 cm silica gel diffusion dryers (DDU 570/H, Topas). The PAM chamber was cleaned by flushing 10 SLPM clean air overnight after each experiment. In order to confirm the cleanliness of the chamber, particle concentration was measured before and after each experiment. The particle concentrations measured each day before experiments were below 70 # cc$^{-1}$.

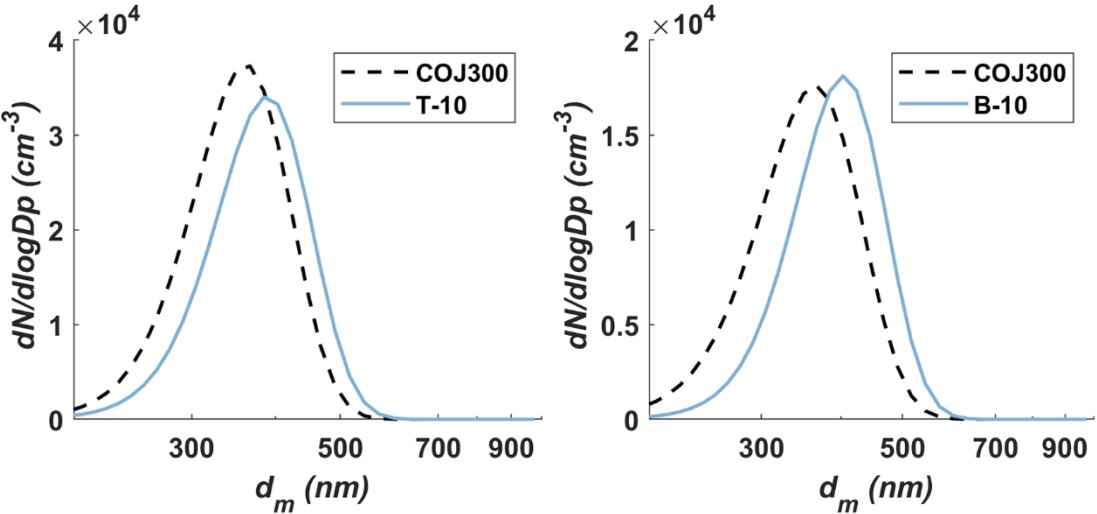

**Figure 2. The modal diameter shift of 350 nm COJ300 BC particles after toluene (left panel) and *β*-caryophyllene (right panel) SOA coating. The dashed and solid lines are fitted curves for bare uncoated and coated particles, respectively.**

Table 2 summarizes all the SOA mixing IN experiments and the operating conditions. A peak shift from 350 nm to 400 nm and an increase of the 400 nm particle concentrations was observed for all experiments (Fig. 2 and Fig. B1), implying SOA coating on BC particles. The name prefixes *BG, T, D, B* in Table 2 stand for background test, toluene SOA coating experiments, *n*-dodecane SOA coating experiments, and *β*-caryophyllene SOA coating experiments, respectively. The name suffixes *BC, 0, 3, 10*, and *s* denote seed BC only, $O_3$ oxidation only, low OH exposure level (3 V), high OH exposure level (10 V) and SOA self-nucleation experiments, respectively. All three organic species were exposed to both low and high OH concentrations to investigate the effect of oxidation level on SOA formation and IN activity. An extra $O_3$ oxidation experiment (*B-0*) was performed for *β*-caryophyllene because it is highly reactive towards $O_3$ and may form SOA absent of OH. Self-nucleation IN

experiments (-*s*) were performed for pure SOAs generated from each organic species to exclude the effect of nucleated pure SOAs mixing with SOA-coated BC particles.

**Table 2.** Experiment conditions of BC and SOA coating experiments. The name prefixes *BG, T, D, B* stand for background test, toluene SOA coating experiments, *n*-dodecane SOA coating experiments, and *β*-caryophyllene SOA coating experiments, respectively. The name suffixes *BC, 0, 3, 10*, and *s* denote seed BC only, O₃ oxidation only, low OH exposure level, high OH
exposure level and SOA self-nucleation experiments, respectively.

| Exp. Name | $O_3$ (ppm) | OH UV Lamp Voltage (V) | Equivalent Atmospheric Exposure (days) | BC Seed | VOC concentration (ppb)[a] | |
|---|---|---|---|---|---|---|
| BG-BC | 0 | 0 | 0 | Y | - | - |
| BG-0 | 110 | 0 | 0 | Y | - | - |
| BG-10 | 110 | 10 | 70-90 | Y | - | - |
| T-10 | 110 | 10 | 70-90 | Y | toluene | 6000 |
| T-3 | 110 | 3 | 10-15 | Y | toluene | 2000 |
| T-s[b] | 110 | 10 | 70-90 | N | toluene | 4000 |
| D-10 | 110 | 10 | 70-90 | Y | *n*-dodecane | 2000 |
| D-3 | 110 | 3 | 10-15 | Y | *n*-dodecane | 500 |
| D-s[b] | 110 | 10 | 70-90 | N | *n*-dodecane | 2000 |
| B-10 | 110 | 10 | 70-90 | Y | *β*-caryophyllene | 5000 |
| B-3 | 110 | 3 | 10-15 | Y | *β*-caryophyllene | 2300 |
| B-0 | 110 | 0 | 0 | Y | *β*-caryophyllene | 5000 |
| B-s[b] | 110 | 10 | 70-90 | N | *β*-caryophyllene | 5000 |

[a]Estimated base on VOC volume injection rate. [b]SOA self-nucleation experiments kept the same OH exposure level and SOA size distribution as corresponding SOA coating experiments

**2.4 Ice nucleation measurement**

BC IN properties, including thermodynamic conditions at IN onset and activation fraction (*AF*) as a function of $SS_i$ and
temperature, were measured with the SPectrometer for Ice Nuclei (SPIN, Droplet Measurement Technologies). The theory,

dimension and operating principles of SPIN can be found in previous studies (Garimella et al., 2016), and a brief description is given here.

SPIN is a continuous flow diffusion chamber style instrument comprising two flat parallel stainless-steel walls whose temperatures are controlled independently. The sampling flow rate of SPIN is 1.0 SLPM. Particles fed into SPIN are constrained by a ~9.0 SLPM sheath gas within a lamina near the centerline of SPIN chamber. Turbulent mixing at the injection point causes some particles to spread outside of the aerosol lamina centerline. Since particles experience lower $RH$ as they spread outside of the lamina, correction factors ranging from ~1.9 to 8.0 were considered in previous studies (Garimella et al., 2017; Nichman et al., 2019; Wolf et al., 2019). Both walls are coated with ~1 mm ice prior to experiments. At the beginning of each experiment, a linear temperature gradient and water vapor partial pressure field are established between the warm and cold walls. Supersaturation with respect to ice is achieved because of the exponential relationship between temperature and saturation vapor pressure (Borgnakke and Sonntag, 2013; Steane, 2016). For all the experiments in this study, SPIN was operated in a $SS_i$ scanning mode (1.0 to 1.6) while keeping the lamina temperature (-46 to -38 ℃) constant for each scan. The $SS_i$ increased from 1.0 at a rate of 0.03 per minute by increasing temperature gradient between the walls above homogeneous IN threshold and then lowered to ice saturation.

An optical particle counter (OPC) collects scattering signals for number counting and sizing, and a forward scattering depolarized signal for phase discrimination at SPIN chamber outlet. The size detection range of the OPC is 0.5 to 15 μm. A machine learning algorithm using the OPC scattering and laser depolarization signal (Garimella et al., 2016) was used to classify each particle as an inactivated aerosol or ice crystal over the course of an experiment.

We define the IN onset as 1% of particles activating, i.e. $AF$ = 1%, for a period of 10 seconds as activation. To account for aerosol spreading outside of the lamina where $SS_i$ is the highest (Garimella et al., 2016),  correction factors of 3.4 and 2.2 were applied for R2500U and R330R (Wolf et al., 2019). The correction factor was determined by taking the effect of morphology on particle behavior within SPIN lamina into consideration. Here the $AF$ is defined as the number concentration of ice crystals identified by the machine learning algorithm divided by the total particle number concentration entering SPIN. For the size-selected bare BC experiments, the total particle number concentration was measured by a CPC operating simultaneously with SPIN, while for the SOA coating experiments, the total particle number concentration was integrated from the SMPS measurement.

# 3 Results and discussion

## 3.1 Ice nucleation on bare BC particles

Figure 3 summarizes IN onset temperature versus $SS_i$ for 100-400 nm (a) R2500U, (b) COJ300 and (c) R330R BC
particles, respectively. Representative error bars in black lines show one standard deviation of variability for SPIN lamina
temperature and $SS_i$ derived from experimental data separately for each panel (Kulkarni and Kok, 2012).

As shown in Table 1 and Fig. 3, the three test BC types exhibit different particle morphology. 400 nm R2500U has the
smallest $D_f$ (~1.92), and COJ300 and R330R have larger $D_f$ (~2.34 and 2.31, respectively); R2500U is the most fractal BC
while COJ300 and R330R are more spherical and compact. Meanwhile, R2500U and COJ300 have similar $\overline{d_{pp}}$ (34-35 nm),
and R330R has larger (~45 nm) primary particles. The larger $\overline{d_{pp}}$ of 200 nm R2500U (Table 1) might result from the blurring
of primary particles under high magnification. Single particle surface area can be inferred by combining fractal level and $\overline{d_{pp}}$
together, and the decreasing order of single particle surface area is R2500U > COJ300 > R330R, which is in agreement with
BET specific surface area data. Negative polarity mass spectra collected for 400 nm BC particles with PALMS are presented
in Fig. A2. The spectra of all three BC types exhibit typical consecutive carbon peaks ($m/z$ = 12, 24, 36, etc.). The spectra of
COJ300 shows presence of oxidized ions, such as O⁻ ($m/z$ = 16), OH⁻ ($m/z$ = 17), and COOH⁻ ($m/z$ = 45), which are highlighted
in red in Fig. A2(b). The frequency of $m/z$ = 16 signal (Table 1) and PALMS O:C ratio result (Fig. A1) confirms that COJ300
is more oxidized than R2500U.

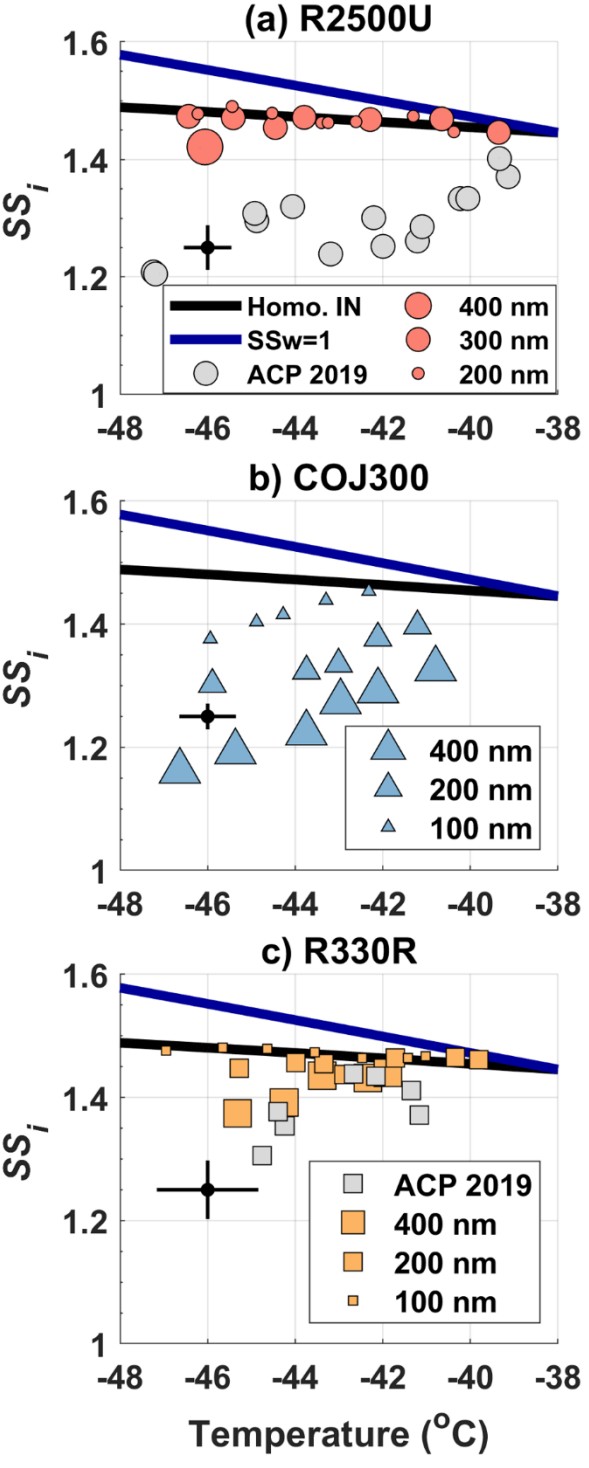

**Figure 3. IN onset $SS_i$ ($AF$ = 1%) phase diagram of bare (a) R2500U (pink circles ●), (b) COJ300 (blue triangles ▲), and (c) R330R (yellow squares ■), respectively. IN onset data for 800 nm R2500U and R330R from previous study is included and denoted in grey markers (Nichman et al., 2019). Different marker sizes in this study corresponds to different $d_m$. Solid blue lines are the water saturation lines, and black lines are homogeneous freezing lines of 200 nm aqueous droplets (Koop et al., 2000). A representative error bar of SPIN lamina temperature and $SS_i$ is given on the left for each panel.**

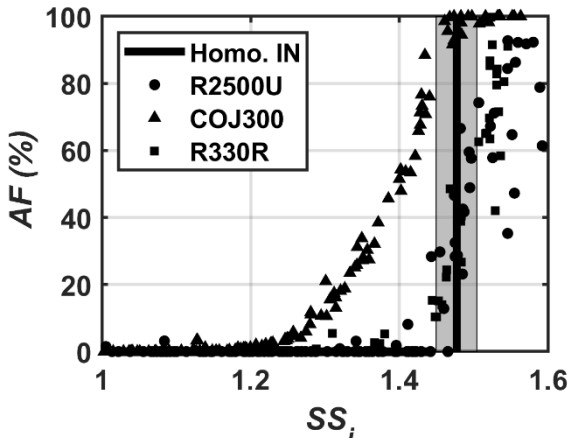

**Figure 4. -45 °C $SS_i$ scan of 400 nm bare BC particles, showing $AF$ as a function of $SS_i$. The black line is the homogeneous freezing threshold for 200 nm aqueous droplets at -45 °C (Koop et al., 2000). The grey shading indicates one standard deviation of variability for SPIN lamina $SS_i$.**

The results in Fig. 3(a) demonstrate that the particle size is relevant to the particle IN ability, consistent with the BC heterogeneous IN ability enhancement triggered by increasing particle size in previous studies (Mahrt et al., 2018; Nichman et al., 2019). The 400 nm R2500U and R330R BC particles were able to nucleate ice below the homogeneous freezing threshold within the representative uncertainty range of SPIN lamina $SS_i$ in the temperature range of -46 to -38 °C. The most spherical and oxidized COJ300 BC particles exhibited IN activity below homogeneous freezing threshold regardless of particle size and temperature in this study. The IN onset $SS_i$ of all depositional active BC particles increases with increasing temperature. The trend is in agreement with previous studies on in cirrus temperature regime (DeMott et al., 1999; Möhler et al., 2005a; Koehler et al., 2009; Chou et al., 2013; Kulkarni et al., 2016; Mahrt et al., 2018; Nichman et al., 2019). The IN onset $SS_i$ of 200 and 300 nm R2500U, as well as 100 and 200 nm R330R, falls into the homogeneous freezing regime. The sharp $AF$ increase of 400 nm R2500U and R330R along the expected homogeneous freezing threshold in Fig. 4 confirm that these two BC types nucleate ice via homogeneous freezing. We conclude that the lower size threshold where the BC particles exhibit IN activity below homogeneous freezing threshold at thermodynamic conditions relevant to cirrus may well lie between 300-400 nm and 200-400 nm for R2500U and R330R around -46 °C, respectively. The IN ability of different size R330R particles at higher temperature (above -45 °C) shows little difference, indicating that the lower size threshold for R330R is likely between 400-800 nm for temperature between -44 to -40 °C (Nichman et al., 2019). The COJ300 BC is more IN active compared with R2500U and R330R. The COJ300 particles show deposition IN ability below homogeneous freezing threshold down to 100

nm within the temperature range in this study; the lower size threshold for COJ300 is below 100 nm. This finding agrees with
the lower size limit between 100 nm and 200 nm for BC particles to act as an active INP reported by Mahrt et al. (2018).

The IN onset results show no clear dependence on particle fractal level and surface area. Even though the more fractal
and branching feature of R2500U BC particles with larger surface area do not clearly exhibit superior IN activity over R330R.
Koehler et al. (2009) showed that IN was favored for oxidized hydrophilic BC, but too many hydrophilic active sites may bond
water molecules, impeding ice embryo formation and thus impair IN (Pruppacher and Klett, 2010). The surface-modified,
highly dispersible and spherical COJ300 with smaller $\overline{d_{pp}}$ shows better IN efficiency than fractal BC, which is consistent with
the results of Mahrt et al. (2018) and Nichman et al. (2019) based on PCF mechanism. The physio-chemical properties of
COJ300 particles, including oxidized surface, appropriate $\overline{d_{pp}}$, and compacted spherical morphology, may result in higher
probability to have cavities with appropriate size and hydrophilicity on particle surfaces (Mahrt et al., 2020). Such cavities can
accommodate liquid water below bulk water saturation and initiate homogeneous freezing of liquid water via PCF pathway
(Marcolli, 2014; David et al., 2019; David et al., 2020).

**3.2 Ice nucleation on BC coated with SOA material**

Figure 5 shows the IN onset $SS_i$ at which 1% of 400 nm SOA-coated COJ300 particles nucleate ice within the temperature
range of -46 to -38 °C. The IN onset data of the bare 350 nm COJ300 particles (marked as + symbol) are also included to
highlight the effect of SOA coating. IN onset $SS_i$ of pure SOA particles are shown as an asterisk separately to rule out the
possible ice formation below homogeneous freezing threshold induced by pure SOA.

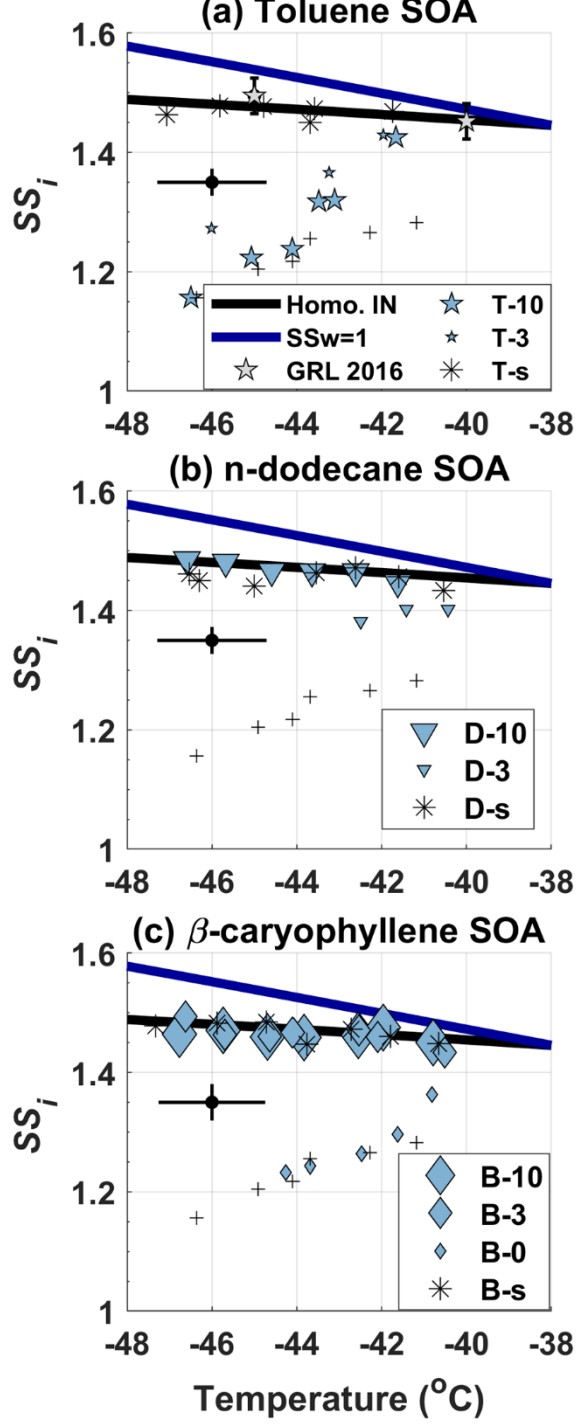

**Figure 5.** IN onset $SS_i$ phase diagram of 350 nm COJ300 BC particles coated with (a) toluene SOA; (b) *n*-dodecane SOA; (c) *β*-caryophyllene SOA. Different symbol sizes denote different OH exposure level. IN onset $SS_i$ of 350 nm bare COJ300 is shown in

black plus (+) symbol for comparison. Pure SOA IN onset $SS_i$ are presented as an asterisk (✳) symbol for each organic species, respectively. The solid blue and black lines are water saturation lines and homogeneous lines for 200 nm aqueous droplets (Koop et al., 2000), respectively. A representative SPIN lamina temperature and $SS_i$ error bar is given on the left side for each panel. The toluene-SOA coated diesel BC IN data from Kulkarni et al. (2016) is also included for comparison.

There exists no distinguishable difference between bare COJ300 and BC coated with highly oxidized toluene SOA (*T-10* in Table 2) from -46 to -44 °C. The toluene SOA mass spectrum in Fig. 6(a) exhibits higher $m/z = 44$ (COO$^-$) and lower $m/z = 43$ (C$_3$H$_7^-$) fraction signal, indicating more oxidized organic species were generated during *T-10* and *T-3* experiments (Lambe et al., 2011b), agreeing with the previous study on toluene SOA (Liu et al., 2018).

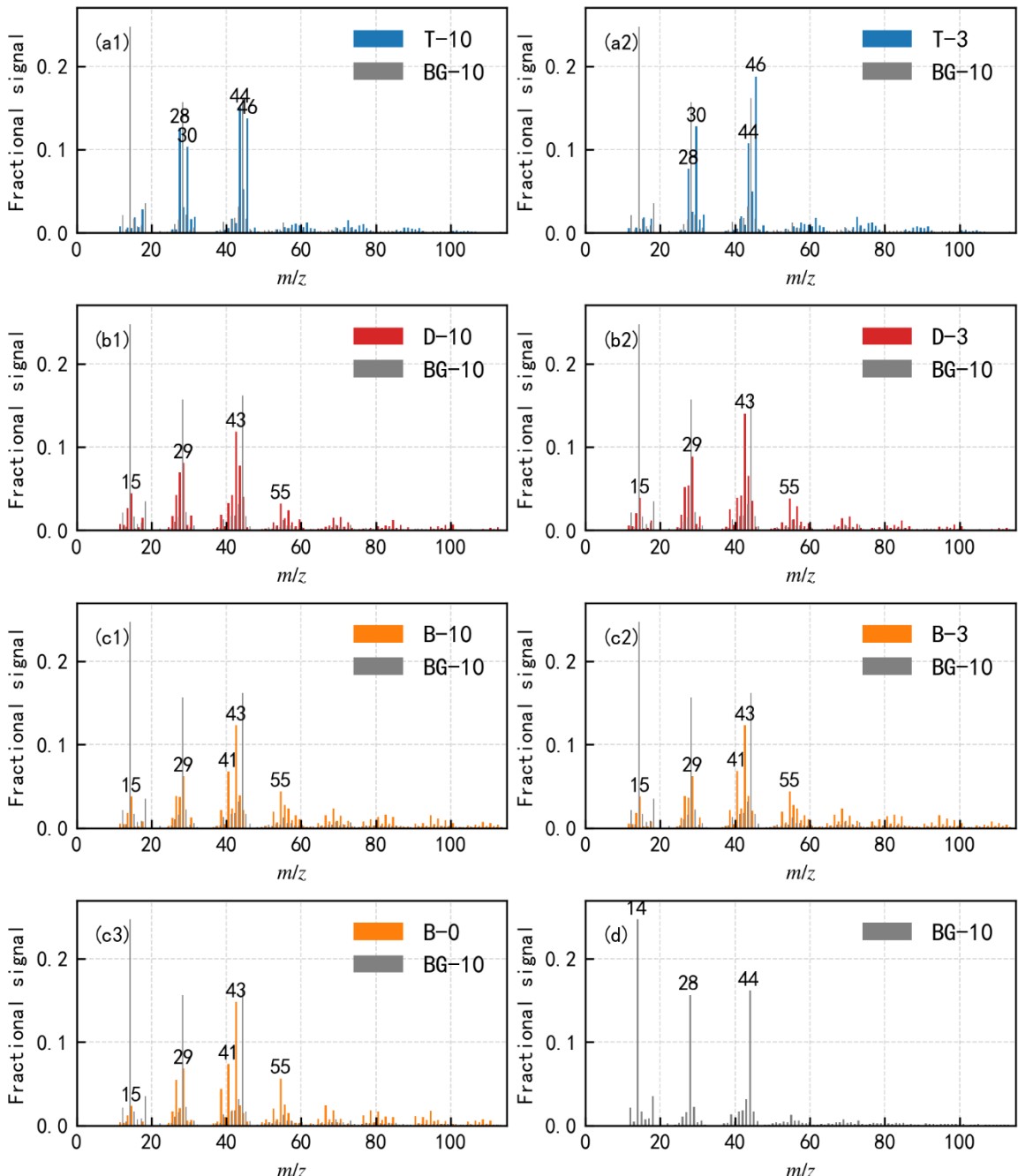

**Figure 6. Normalized AMS mass spectra of COJ300 BC particles coated with (a) toluene SOA; (b) *n*-dodecane SOA; (c) *β*-caryophyllene SOA; and (d) bare COJ300 BC particles. More oxidized SOA is generated when toluene act as precursor, while less oxidized SOAs are generated when *n*-dodecane and *β*-caryophyllene act as precursors in this study, as indicated by the different fractions of *m/z* = 43 and 44, respectively (Lambe et al., 2011b; Ng et al., 2011; Canagaratna et al., 2015). The absolute organic mass**

**loading present in the bare COJ300 BC experiment is less than 1% of the organic mass loading from the other three types of SOA coating experiments.**

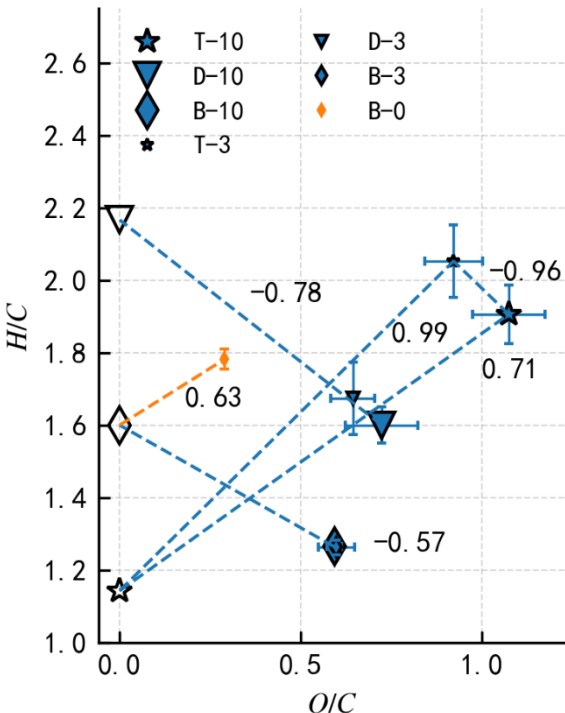

 **Figure 7. Elemental H/C ratio as a function of O/C ratio for three pure organic precursors (hollow symbols) and corresponding COJ300 BC particles seeded SOA inside the PAM reactor (filled symbols). Different symbol sizes denote different OH exposure level. The negative slopes SOA coating experiments are consistent with simultaneous carboxylic acid group addition and C-C single bond breakage (Heald et al., 2010; Lambe et al., 2011b).**

The higher O/C ratio (Fig. 7) of toluene-derived SOA may enhance the hygroscopicity of the particle (Lambe et al.,

2011b; Zhao et al., 2016; Liu et al., 2018) with a potential to form aqueous film on BC surface and reduce the IN ability of BC particles. On the other hand, Hinks et al. (2018) showed that toluene-derived SOA contained a significant amount of oligomers under dry laboratory conditions similar to what we conducted in the PAM chamber in this study. Volkamer et al. (2001) proved that glyoxal, which could facilitate the oligomerization as described by Hinks et al. (2018), can be produced from toluene reaction with OH as highly oxidized products. The slopes of T-10 and T-3 in Fig. 7 lie between 0-1 (0.71 and 0.99 for T-10

and T-3, respectively), suggests that oxygen and hydrogen atom addition accompanied by carbon-carbon double bond as well as benzene ring breakage might have happened during the toluene photooxidation experiments. These large oligomers could potentially reduce the hygroscopicity and alter the phase state of toluene SOA to be semi-solid or solid within the temperature range we investigated in this work (DeRieux et al., 2018; Zhang et al., 2018c; Li et al., 2020), under which the SOA can still nucleate ice (Murray et al., 2010; Berkemeier et al., 2014; Zhang et al., 2019c). The toluene SOA in *T-10* with O:C ratio over

1 (Fig. 7) has most likely already transited into solid or semi-solid glassy state at the temperature range we investigated before

entering SPIN (DeRieux et al., 2018). Therefore, it is very likely that BC particles are mostly sticked with or embedded in these glassy SOA with some bare BC part exposure. Ice crystals may therefore form on the carbonaceous part of partially coated particles, whose IN onset $SS_i$ should be the same as bare COJ300. At temperatures above -43 °C, toluene SOA-coated BC particles nucleate ice at $SS_i$ ~0.1 to 0.15 above bare 350 nm COJ300, but still ~0.15 below the homogeneous freezing

threshold. This might be due to the hygroscopicity enhancement of the toluene SOA-coated BC in $T$-10. BC coated by ~10-15 equivalent days atmospherically oxidized toluene SOA ($T$-3 in Table 2) is less IN active than those coated by highly oxidized toluene SOA ($T$-10) at around -46 °C and -43 °C. This might also be attributed the oxidation level and the corresponding phase state difference between the SOA generated from $T$-10 and $T$-3 experiments, which is beyond the scope of this study and requires further detailed phase transition study for toluene SOA. Overall, two competing effects, i.e. the hygroscopicity

enhancement deactivating BC IN ability, together with toluene SOA glass phase transition producing sticked or partly coated BC, may make our toluene SOA coating IN onset move towards but not fully in the homogeneous freezing regime. The toluene SOA coated diesel combustion BC (Kulkarni et al., 2016), however, nucleate ice near the homogeneous threshold, as indicated in Fig. 3(a). This might be due to the much thicker (90 nm) organic coating in their study compared to the 25 nm coating of this work, leading to a complete coverage of BC surface.

The IN onset $SS_i$ of highly oxidized $n$-dodecane SOA-coated COJ300 particles ($D$-10 in Table 2) in Fig. 5(b) show that these particles nucleate ice homogeneously between -46 and -42 °C. BC coated by slightly oxidized $n$-dodecane SOA ($D$-3 in Table 2) nucleates ice nominally lower than homogeneous freezing threshold between -43 and -40 °C. As shown in Fig. 5(c), the IN onset $SS_i$ of OH-oxidized $\beta$-caryophyllene SOA-coated COJ300 particles ($B$-10 and $B$-3 in Table 2) is in the homogeneous freezing regime. The mass spectra in Fig. 6(b) and Fig. 6(c) exhibit large fraction of signals at $m/z$ = 15 ($CH_3^-$),

29 ($C_2H_5^-$), 43 ($C_3H_7^-$), and 55 ($C_4H_7^-$) for $n$-dodecane and $\beta$-caryophyllene SOA coating experiments in this study, implying formation of less oxidized aliphatic fragments during these experiments (Lambe et al., 2011b). The H/C and O/C values of $n$-dodecane and $\beta$-caryophyllene SOA coating in Fig. 7 are smaller than that of toluene SOA, which are in agreement with previous studies (Simonen et al., 2017; Li et al., 2019; Pereira et al., 2019). The slopes of H/C and O/C values of these two types of SOA and their respective two precursors (Fig. B2) are in the range between -1 and 0, which is consistent with the

simultaneous formation of carboxylic acid functional groups and C-C bond breakage (Heald et al., 2010; Lambe et al., 2011b). Addition of carboxylic acid group may enhance the hygroscopicity of $n$-dodecane and $\beta$-caryophyllene SOA, and the hygroscopicity is further enhanced with more OH exposure (Frosch et al., 2013; Yee et al., 2013; Schilling et al., 2015; Bé et al., 2017). We conclude that BC with OH oxidized $n$-dodecane and $\beta$-caryophyllene SOA coatings, regardless of oxidation level, may condense on BC surface and forms organic films, leading to nucleation in the homogeneous regime. However,

COJ300 BC coated with $O_3$ oxidized $\beta$-caryophyllene SOA ($B$-0 in Table 1) shows no significantly alternation of IN ability, as shown in Fig. 5(c). Unlike OH oxidation of $\beta$-caryophyllene where fragmentation happens, $O_3$-addition is very likely to happen first on the carbon-carbon double bond of $\beta$-caryophyllene in $B$-0, leading to formation of semi-solid or solid SOA (Nguyen et al., 2009; Winterhalter et al., 2009), as illustrated in Fig. B2 with a slope between 0-1. As with the case of $T$-10,

such semi-solid or solid SOA might collide and stick with BC particles, leaving some bare carbonaceous surface that can nucleate ice following the IN pattern of COJ300 BC.

The experimental results are attributed to two factors: organic coating and SOA phase state. Previous studies controlling the combustion fuel-air-ratio produced BC particles occupying different organic content fractions, with higher organic content resulting in amorphous organic surfaces (Möhler et al., 2005b; Crawford et al., 2011; Mahrt et al., 2018). In these studies, shifts from heterogeneous to homogeneous freezing with increasing organic content have been observed. Kulkarni et al. (2016) reported that an 80 nm $\alpha$-pinene SOA coating can suppressed the ice nucleation ability of 120 nm diesel BC particles. However, studies show that as the phase state of the organic coating changes below certain threshold, especially near glass transition temperature, these organic coatings might be able to heterogeneously nucleate ice (Murray et al., 2010; Berkemeier et al., 2014; Zhang et al., 2019c). The suppression of BC IN ability by organic coating was attributed to coverage of surface-active sites and filling of pores on BC surface when the volatility of the organic coating is relatively high and might present in liquid phase. Certain SOA coatings in this study are less oxidized and thus may similarly impair BC IN ability due to their relatively high volatility, as Docherty et al. (2018) and Hildebrandt Ruiz et al. (2015) showed an inverse correlation between the volatility and oxidation state. Our results suggest that less oxidized SOA ($n$-dodecane and $\beta$-caryophyllene derived SOA from photooxidation), with 200 to 4000 folds of typical tropospheric SOA (Tsigaridis and Kanakidou, 2003; Heald et al., 2008; Hodzic et al., 2016) mass loading in PAM chamber (~2000 to 4000 µg m$^{-3}$), are more likely to condense on seed particle and form fully coated BC particles, moving IN onset $SS_i$ to the homogeneous regime, while $\beta$-caryophyllene SOA oxidized by O$_3$ did not alter the $SS_i$ of the soot particles. In addition, more oxidized SOA (toluene derived SOA from photooxidation) with potentially more oligomer formation, moving IN onset $SS_i$ towards, but still below, homogeneous freezing.

## 4 Atmospheric implications

BC particles emitted from combustion sources (such as aero-engines) are carbonaceous nanoscale fractal aggregates with primary particle diameter of 20-50 nm (Bockhorn et al., 2009; Vander Wal et al., 2014). These BC particles can remain suspended in the atmosphere for days, and might undergo compaction and atmospheric aging, such as oxidation and mixing with atmospheric organic species. This study focuses on the impact of morphology, particle size and mixing state on the IN ability of BC-containing aerosols. Three BC proxies were chosen to represent freshly emitted (in other words, unoxidized and more fractal) BC (R2500U), unoxidized compacted BC (R330R), and atmospheric chemically aged BC (COJ300). The morphological characteristics are within the value range of typical BC emitted from combustion sources, and those collected in field observations (e.g., Lapuerta et al., 2007; China et al., 2013; China et al., 2014; Vander Wal et al., 2014; China et al., 2015b; Zhang et al., 2019b). BC primary particle size range in this study lies between 10 to 70 nm with a modal size around 25 to 40 nm, being consistent with previous primary particle studies on combustion BC (e.g, Smekens et al., 2005; Liati et al., 2016; Joo et al., 2018). Previous field observations of transportation emissions and biomass burning reported that ambient BC

occupied $d_a$, circularity, and roundness in the range of 130 to 940 nm, 0.19 to 0.55, and 0.32 to 0.6, respectively (China et al., 2013; China et al., 2014; China et al., 2015a; China et al., 2015b), overlapping with the range in this work. R2500U is similar to the fresh BC emitted from B737 at medium power burning conventional jet fuel in terms of morphology characteristics (Vander Wal et al., 2014). The primary particle size is consistent with BC emitted from prevalent gas turbine engines (Huang and Vander Wal, 2013). Findings in this study can be relevant to airborne aircraft emissions and ground emissions carried by updrafts to tropopause.

The IN results for bare BC particles show dependence on particle size and surface chemistry, but the role of fractal level seems to be of limited importance. The lower size limit of bare BC to exhibit IN activity is between 300-400 nm for R2500U at -46 °C. This is important for freshly emitted BC from aircraft engines and ground transportation, which are generally fractal and smaller than 200 nm with modal size ranging from 20 to 100 nm (Kittelson, 1998; Wey et al., 2006; Anderson et al., 2011; Wang et al., 2016; Moore et al., 2017; Raza et al., 2018; Awad et al., 2020). It is unlikely that small, freshly emitted BC will activate as INP in aircraft plumes below the homogeneous freezing threshold if they possess similar physicochemical properties as R2500U. The smallest size for compacted BC (R330R) to activate as INPs lies between 200-400 nm at -46 °C. However, IN ability of small BC particles may be enhanced after cloud cycles, during which fractal BC geometries may collapse, forming PCF favoring morphology (Mahrt et al., 2020). Apart from the most spherical morphology, the smaller $d_{pp}$ of COJ300 may also offer higher probability to form smaller mesopores with appropriate size to accommodate ice crystal formation below water saturation, with particles down to 100 nm can act as efficient INP. Pores formed by BC with larger $d_{pp}$ might be too wide to accommodate liquid water at our experimental conditions. Besides, the COJ300 IN results imply that ice crystal formation may favor oxidized hydrophilic surfaces, confirming the importance of surface hydrophilicity for pore filling in PCF mechanism (David et al., 2019; David et al., 2020). This suggests that long-lived atmospheric BC particles, after being oxidized and compacted, may act as efficient INP.

To simulate atmospheric aging, toluene, $n$-dodecane and $\beta$-caryophyllene were chosen to represent anthropogenic and biogenic SOA precursors (Atkinson and Arey, 2003; Hu et al., 2008; Ding et al., 2014). Toluene-derived SOA coatings impede BC heterogeneous IN activity slightly while $n$-dodecane and $\beta$-caryophyllene-derived SOA coatings caused BC particles to nucleate ice homogeneously. BC emitted from aircraft and vehicles are likely to be coated by toluene and $n$-dodecane derived SOA (e.g., Beyersdorf et al., 2012; Beyersdorf et al., 2014; Timko et al., 2014). According to our experimental results, even though such coating can facilitate particle growth, SOA-coated particles are more likely to nucleate ice near the homogeneous freezing threshold.

The conclusions drawn here for BC proxies may deviate from genuine BC emitted from combustion sources. Nonetheless, BC surrogates are often used in research to mimic aircraft emitted BC for their similarity and availability (e.g., Persiantseva et al., 2004). Additional IN studies, over a wider temperature range would also be required for the proxies to firmly verify the

PCF mechanism; the question whether the studied IN is depositional or in fact homogeneous IN of liquid water in pores and cavities, remains to be answered due to the limited temperature range investigated in this study.

## 5 Summary

The IN ability of size-selected (100-400 nm) BC particles with different morphologies and surface chemistry and BC particles coated with toluene, *n*-dodecane, and *β*-caryophyllene-derived SOA has been systematically investigated in the cirrus temperature regime (-46 to -38 °C). Three aerosolized BC proxies were selected to represent particle morphology at different atmospheric aging stages, i.e. freshly emitted (R2500U), atmospheric compacted (R330R), and atmospheric compacted and oxidized (COJ300). The IN activity was investigated in relation to particle size, morphology, surface chemistry, SOA precursor type and OH exposure level.

The results show the lower size limit for BC particles to exhibit IN activity varies between BC type. 400 nm freshly emitted and compacted BC particles nucleate ice near the homogeneous freezing threshold. Ice crystals form on most spherical, surface-modified hydrophilic BC at $SS_i$ as low as 1.15. The onset of some IN below the homogeneous freezing threshold, as opposed to purely homogeneous freezing, occurs for some BC types between 100-200 nm, in some cases below 100 nm. We conclude that BC IN favors larger, spherical particles with oxidized hydrophilic surfaces. The highly fractal BC particles did not necessarily act as superior INP over more spherical ones as would normally be anticipated from surface area. This could be attributed to PCF occurring in the pores and cavities with appropriate size offered by compacted BC particles.

Toluene-derived SOA coatings increase bare BC IN onset $SS_i$ by 0.1-0.15, but still below the homogeneous freezing threshold. Slightly oxidized toluene SOA coatings seem to have a stronger deactivation effect on BC IN ability than highly oxidized toluene SOA, which might be caused by oligomer formation and phase state transition of toluene SOA material under different oxidation levels. The larger molar masses of OH oxidized *n*-dodecane and *β*-caryophyllene SOA enhances the coating thickness and further elevates the IN onset $SS_i$ into the homogeneous freezing regime. This might be due to SOA material filling the pores on BC surfaces and leading to IN near the homogeneous regime. $O_3$ oxidized *β*-caryophyllene SOA does not seem to affect BC IN activity. OH exposure levels of *n*-dodecane and *β*-caryophyllene SOA coating experiments, from an equivalent atmospheric 10 to 90 days, shows no significant difference. Our study broadens aging processes of atmospheric BC particles and may offer the basis to better predict their IN activity and contribution to cirrus cloud formation. We suggest future studies should focus on IN activity of realistic combustion particles (aircraft, vehicles, and biomass burning, etc.) and advanced single particle characterization for validation of the PCF mechanism.

## Appendix A: BC physio-chemical properties characterization

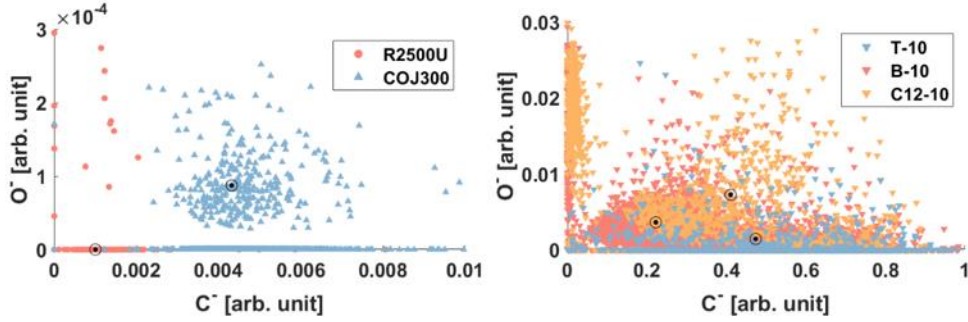

Figure A1. Negative polarity oxygen and carbon peak areas from PALMS for (left panel) 400 nm R2500U and COJ300 BC; (right panel) SOA-coated BC particles. Cluster centroid denoted as ⊙. Generally, the probability of O⁻ signal presence in COJ300 is an order of magnitude higher than R2500U.

**Figure A2. Representative negative-ion PALMS mass spectra of bare size selected (a) R2500U, (b) COJ300, and (c) R330R BC particles with a modal size around 400 nm. The ions indicative of oxidized material (*m/z* = 16, 17, and 45) are highlighted in red.**

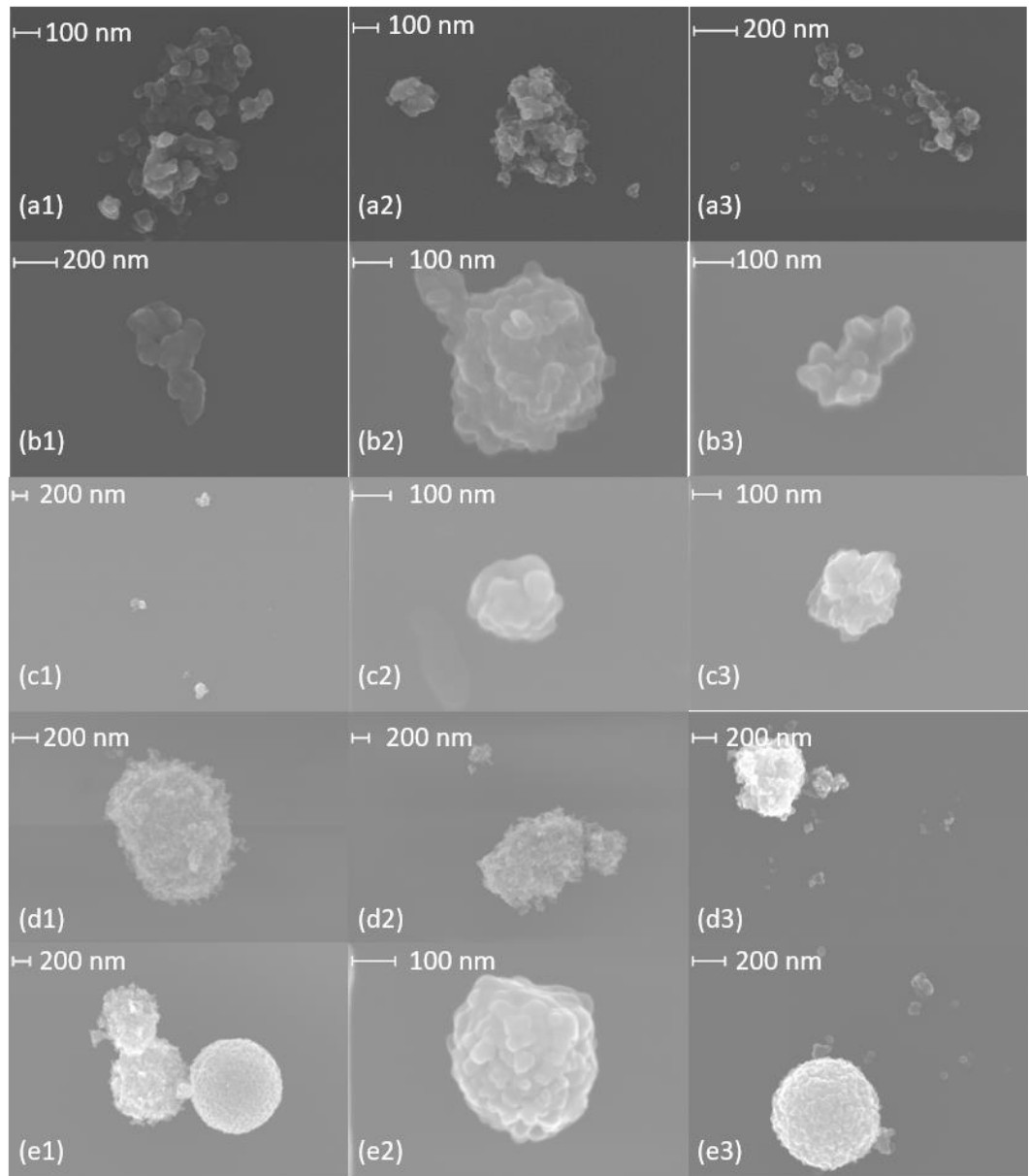

**Figure A3. SEM images of (a) 400 nm R2500U, (b) 300 nm R2500U, (c) 200 nm R2500U, (d) 400 nm R330R, and (e) 400 nm COJ300.**

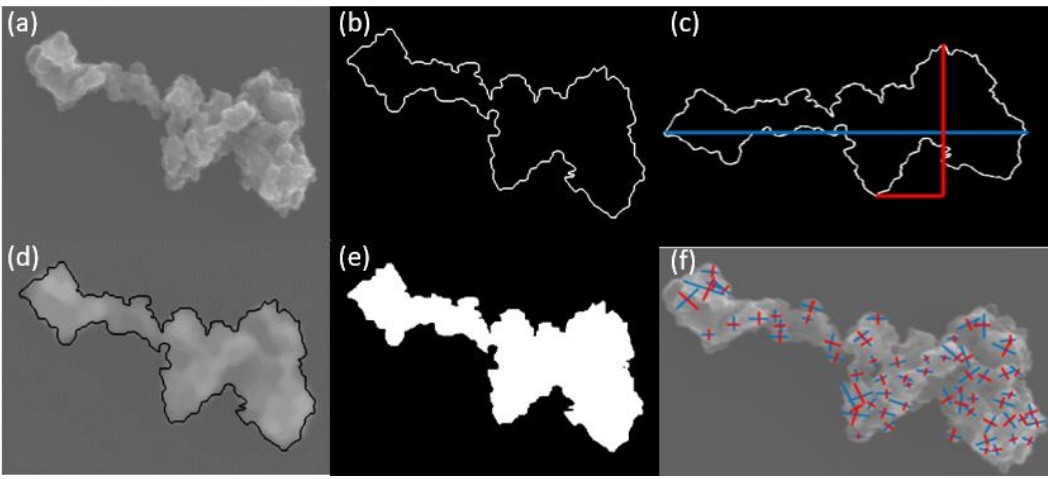

**Figure A4. Example of processing of SEM images. (a) original image; (b) manually draw an approximate aggregate outline; (c) obtain the longest dimension ($L_{max}$) of an aggregate periphery to the perpendicular maximum width ($W_{max}$); (d) validation of the periphery; (e) use binary figure to obtain project aggregate area ($A_a$); (f) measurement of primary particle diameter ($d_{pp}$).**

BC aggregate parameters, including $L_{max}$ and $W_{max}$, perimeter, and project area, are determined by manually drawing the periphery of the aggregate, as shown in Fig. A4 (a-e). The primary particle diameter is determined by identifying the boundary of BC primary particle in the SEM image. After manual selection of the start and end points of a primary particle's length and width respectively, the distance between these points are calculated automatically and recorded. The primary particle diameter equals to the geometric mean of the length and width. Primary particle size distribution of a specific BC type and size can then be obtained by categorizing primary particle diameters into different size bins and count the frequency, as shown in Fig. A5. BC primary particle size distribution obeys normal distribution. The coefficients and goodness of normal distribution fittings are shown in Table A1.

**Table A1.** Coefficients and goodness for the normal primary particle size distribution fitting. 95% confidence intervals are enclosed in brackets.

| BC type and size | Geometric mean diameter (nm) | Geometric standard deviation (nm) | $R^2$ |
|---|---|---|---|
| 200 nm R2500U | 36.85 [35.97, 37.73] | 14.30 [13.05, 15.55] | 0.98 |
| 300 nm R2500U | 30.50 [29.43, 31.56] | 12.70 [11.20, 14.21] | 0.96 |
| 400 nm R2500U | 29.06 [28.25, 29.97] | 12.70 [11.55, 13.85] | 0.98 |
| 400 nm COJ300 | 30.41 [29.54, 31.29] | 12.11 [10.88, 13.35] | 0.97 |
| 400 nm R330R | 40.39 [33.44, 47.34] | 16.32 [15.06, 17.58] | 0.95 |

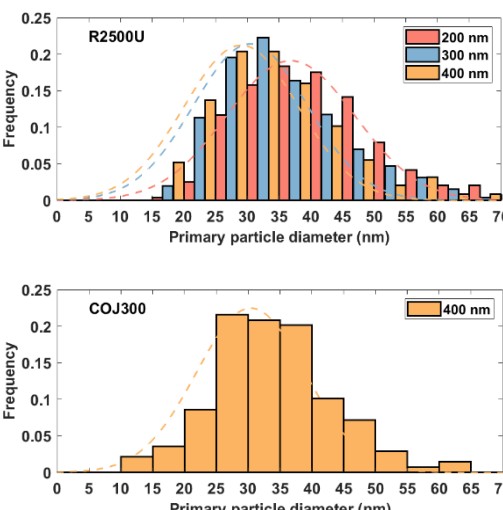

**Figure A5. Primary particle size distributions for select BC particle types. The dashed lines are fitted normal distribution curves.**

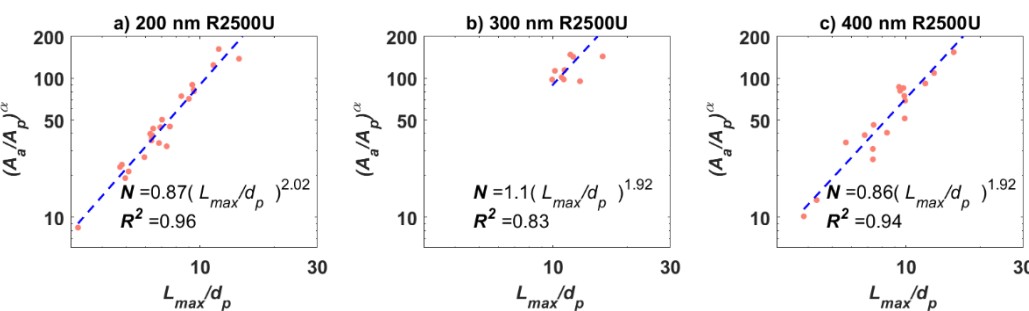

**Figure A6. Power law fit to obtain 3-D fractal dimensions of (a) 200 nm ($N$=25), (b) 300 nm ($N$=12), (c) 400 nm ($N$=21) R2500U BC particles. More than 10 aggregates were analyzed for each size.**

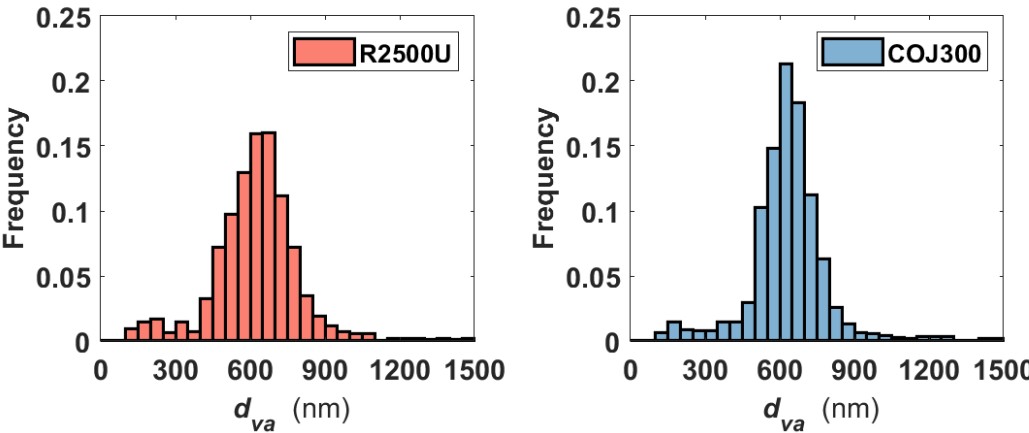

**Figure A7. Vacuum aerodynamic diameter ($d_{va}$) derived from PALMS for 400 nm R2500U and COJ300 (Cziczo et al., 2006).**

## Appendix B: SOA coating experiments

**Table B1.** Organic compounds engaged in this study. The parameters are taken from room temperature data.

| Compound | Structure | Formula (m/z) | SOA mass yields (%)[a] | Rate constants $\times 10^{12}$ [cm$^3$ molecule$^{-1}$ s$^{-1}$] | |
|---|---|---|---|---|---|
| Toluene | | $C_7H_8$ (92) | 8 - 49 (Hildebrandt et al., 2009) | $k_{OH}$ | 6.36 (Tully et al., 1981) |
| n-dodecane | | $C_{12}H_{26}$ (170) | 9 (Presto et al., 2010) | $k_{OH}$ | 13.3 (Lamkaddam et al., 2019) |
| β-caryophyllene | | $C_{15}H_{24}$ (205) | 17 - 63 (Griffin et al., 1999) | $k_{OH}$ | 200 (Shu and Atkinson, 1995) |
| | | | 53[b] (Jaoui et al., 2013) | $k_{O3}$ | 1.16 × 10$^{-2}$ (Shu and Atkinson, 1994) |

[a]Measured at organic particle concentration of 10 µg/m$^3$; [b]Measured at organic particle concentration of 26 µg/m$^3$.

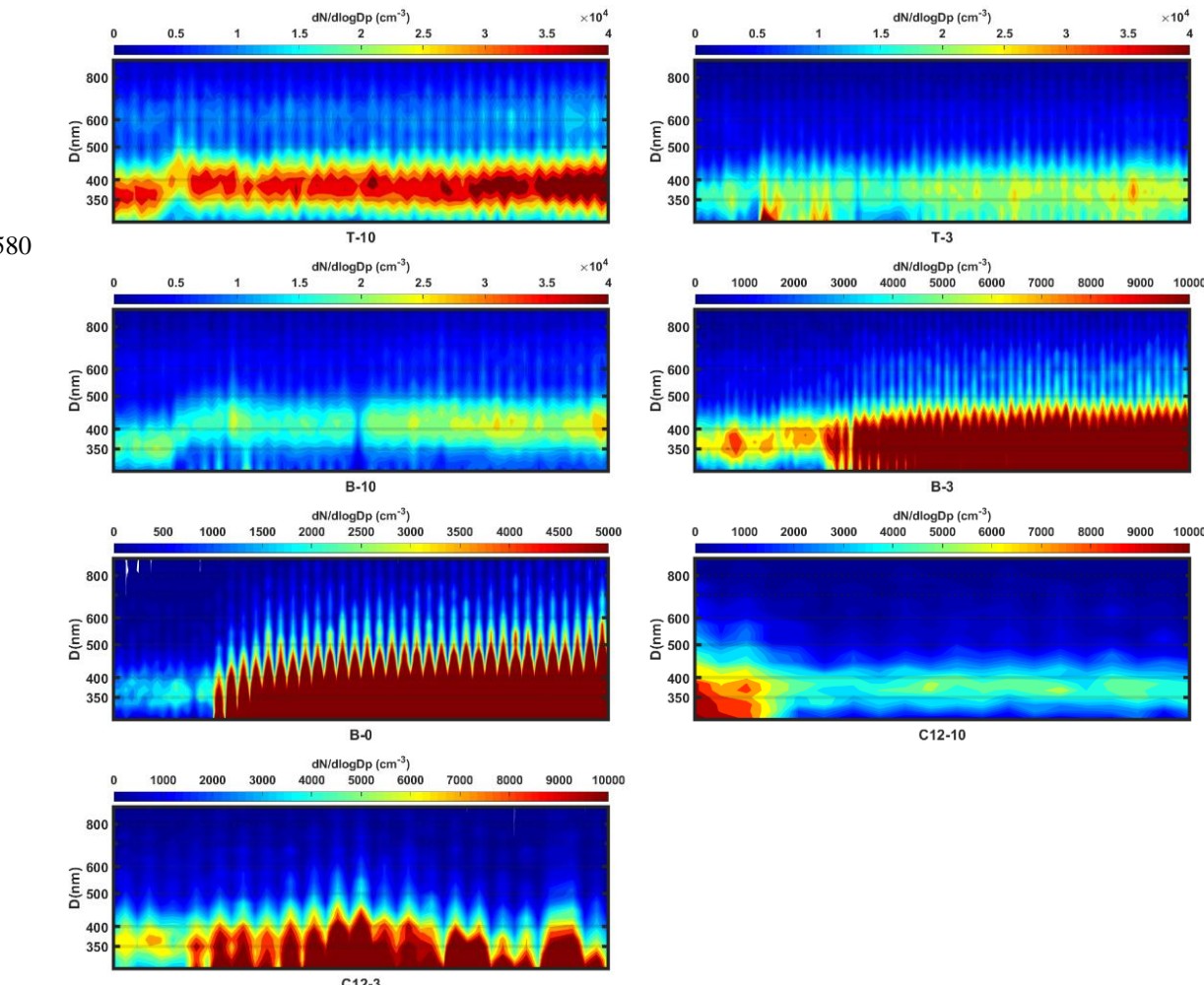

**Figure B1. Temporal particle size evolution for different BC and SOA mixing experiments. The x axis is the number of scans, equivalent to experiment time. Together with the color map, y axis shows the size distribution for a certain time. A size shift from 350 nm to 400 nm can be observed for each experiment.**

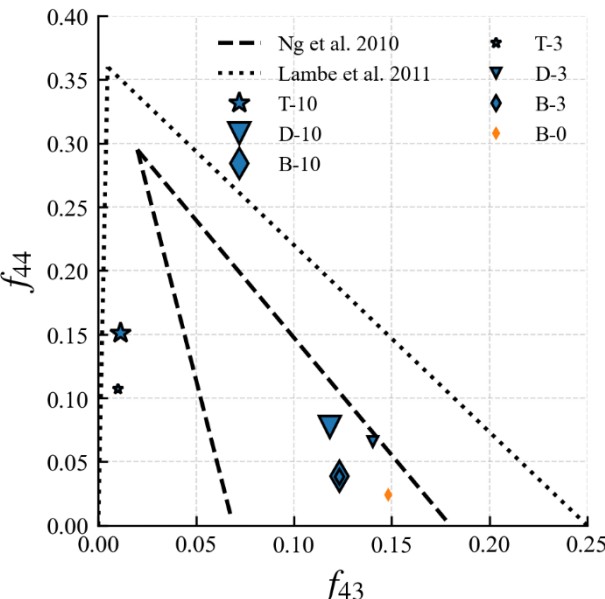

**Figure B2 The measured fraction of AMS signals at** *m/z* **= 43 (*f₄₃*) and** *m/z* **= 44 (*f₄₄*). SOA generated from *n*-dodecane and *β*-caryophyllene in this study are within the ambient SOA *f₄₄* and *f₄₃* range measured by Ng et al. (2010). Toluene-derived SOA in this study exhibits similar *f₄₄* and *f₄₃* signal range to the laboratory measurement of glyoxal-derived SOA (Lambe et al., 2011b).**

**Data availability**

Data inquires can be directed to the corresponding author (Longfei Chen, chenlongfei@buaa.edu.cn).

**Author Contributions**

CZ, YZ, MJW, LN, TBO and DJC designed the experiments and methodology. CZ collected black carbon samples and performed morphology characterization. CZ, YZ, MJW and CS performed chemical analyses, and measured ice nucleation activity. CZ, YZ, MJW, LN, LC, and DJC prepared manuscript with input from all coauthors.

**Acknowledgments**

The authors declare no competing interests. We thank our peer reviewers for their valuable comments and suggestions to make 600 our paper scientifically improved and more concise. We thank Andrew Lambe and other colleagues at Aerodyne Research Inc. for their help with the PAM reactor and SOA coating experiment. This work was supported by National Natural Science Foundation of China (Grant No. 51922019) and Chinese Government Scholarship (Grant No. 201806020052). YZ was supported by the NSF Postdoctoral Fellowship under AGS Grant No. 1524731 and the National Institutes of Health (NIH) Grant No. T32ES007018.

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
