# Peer review of "The effects of morphology, mobility size and SOA material coating on the ice nucleation activity of black carbon in the cirrus regime"

_Atmospheric Chemistry and Physics, 2020_

## Referee Comment (RC1) · Anonymous Referee #2 · 25 Sep 2020

**Review to "The effects of morphology, mobility size and SOA material coating on the ice nucleation activity of black carbon in the cirrus regime" by Zhang et al.**

The work by Zhang et al. presents laboratory data of the ice nucleation ability of different types of black carbon, along with a characterization of the physical and chemical particle properties. The study extends previous laboratory measurements on the ice nucleation ability of soot. This topic fits well within the scope of ACP, and the results are novel and of great interest for the ice nucleation community.
Overall, I find the manuscript is well written and structured and the results are presented in a clear way and put into context of previous literature. However, from my point of view there are some aspects, in particular in the discussion of the ice nucleation results, that should be revised. Once the points outlined below are sufficiently addressed, I support publication of the manuscript in ACP.

**Overall comments and general concerns:**

- The authors do not seem to provide a consistent view of the ice nucleation mechanism of the investigated soot types, which I find in part hard to follow. At various instances throughout the manuscript the authors describe the observed ice nucleation activity to result from deposition nucleation (e.g. L25, 27, 68, 90, 284). At other instances, the authors describe how earlier studies by Nichman et al. (2019; including authors from the present study) and Mahrt et al. 2018 (L80-84) have identified soot to form ice via the pore condensation and freezing (PCF) mechanism at cirrus cloud temperatures, the temperature range studied here. The PCF mechanism is then suggested to explain the observed ice nucleation results (e.g. L228-332) and also used to argue how the coating of the soot particles with secondary organic aerosol (SOA) leads to filling of the pore, which renders these pores unable to contribute to ice formation via PCF (e.g. L396-397). This seems to be in stark contradiction to the suggested ice formation via deposition nucleation for some of the bare soot particles. In their discussion (L438-439 and L460) the authors then vaguely talk about "verifying the PCF" mechanism in future studies but it does not become fully clear if the ice nucleation observed in the present study is attributed to PCF or to "classical" deposition nucleation.
  I assume that the authors interpret the observed ice formation as PCF and whenever the authors refer to "deposition nucleation", they refer to ice formation at cirrus temperatures below water saturation (and below homogeneous freezing conditions, to be specific). However, this terminology is confusing for the reader, and suggests a flip-flop between both mechanisms, i.e. classical deposition nucleation and PCF. I would appreciate if the authors could provide a clearer description of the ice nucleation mechanism of the observed ice formation on the investigated soot particles. A relatively simple solution to reduce the in part confusing back and forth would be to use just "IN activity" rather than "deposition IN activity" to describe the ice formation below homogeneous freezing conditions and then provide a comprehensive discussion of both "classical" deposition nucleation and PCF in the results section.
- The title of the presented manuscript suggests that an effect of particle size, morphology and coating on the ice nucleation ability of BC is investigated in this study. While I appreciate the efforts in comprehensively and carefully characterizing the physical and chemical properties of the investigated soot types the associated uncertainties are not sufficiently discussed, which I find problematic. For example, the goodness of the size selection and hence the difference between a 300 nm and a 400 nm R2500U particle are not discussed. Upon revision of the manuscript, the uncertainties of the various measurements of these variables on the dependent variable (ice nucleation activity), should be more clearly emphasized and discussed. As another example, I am not clear whether the O:C ratio of the COJ300 and the R2500U samples are really different.

**Specific comments:**

- L22: Delete "aerosolized"
- L25: Delete "(-24 to -46°C)"
- L25,27: "Deposition IN"; please see my general comment above.
- L31: "We attribute the inhibition of IN ability to the filling of pores on the BC surface by the SOA material coating." In other words, the IN ability of the uncoated particles results from the pores, i.e. takes place by PCF. This is in contradiction to the "deposition IN" mentioned above and at various other instances in the manuscript. Please see my general comment above.

- L33: Delete first "days"
- L37: Delete "terrestrial"
- L39,41: I suggest replacing the given references by primary reference, e.g. the classical textbook by Pruppacher and Klett (1997).
- L43: Why are you only introducing "deposition IN" here and not also PCF?
- L45: Rephrase as: "BC from aircraft emissions may be an …"
- L53: Delete "simulate the atmospheric environment"
- L59: Hoose and Möhler (2012) is not a primary reference and I suggest deleting it here.
- L63-64: Similarly, the probability of a BC aggregate to contain a pore with the right properties (e.g. pore size and contact angle) increases with increasing aggregate diameter, which would favor PCF for larger particles (Mahrt et al., 2018). This could be added.
- L68: "deposition mode"; see my general comment above.
- L70: Rephrase to "…during atmospheric aging, leading…"
- L71: Add Bhandari et al. (2019) and Khalizov et al. (2009)
- L80: Delete Kumfer ad Kennedy (2009)
- L80-81: This statement is not true and needs to be revised. E.g. see Table 1 and Fig. 3 in Mahrt et al. (2018). The investigated "FS" soot has the second highest surface area, but "LB_RC" shows a similar or higher IN activity.
- L82: Delete Koop (2017) and Marcolli (2017). Add David et al. (2020).
- L83: Rephrase to: "…in which IN of BC is considered as homogeneous freezing of…"
- L84: Replace references by Fisher et al. (1981).
- L90, 100: "deposition nucleation"; please see my general comment above
- L102: "-38°C"
- L128: Add: "… in particles of aircraft engine exhaust…"
- L131: Please specify "light" aromatic species
- L133: How representative is β-caryophyllene for biogenic SOA? How does it compare to other major biogenic SOA types such as α-pinene SOA? For the Sect. 2.1.2. more discussion on the atmospheric abundance and relevance of the chosen SOA precursors should be given.
- L146: Can the authors comment on how the wet dispersion in the atomizer affects the BC particle morphology?
- L150: The flow reported here (1.5 SLPM) and on L215 (2.2 SLPM) are different. How does this change your flow rate through the DMA and possibly affect your size selection. Please comment in the main text.
- L152-153: Have the TSI and BMI DMA been compared? Can the authors provide a size distribution of the soot types investigated that was size-selected with both DMAs to ensure that there is no difference in the uncoated and coated particles due to the instrumentation used? Please also add the aerosol-to-sheath flow ratios used in the DMAs for aerosol size selection.
- L155: An RH value of 16% seems extremely high. Please comment how this affects your IN measurements that are performed using this air stream (conserving specific humidity) but exposing it to T < -40 °C.
- L160: Please specify "MOUDI" acronym
- L170: Rephrase as "… by the PALMS."
- Table 1:
  - da of R2500: Why is the projected area diameter for dm = 300 nm larger than for dm = 400 nm?
  - da: Please comment in the text why the da values for dm = 400 nm particles for COJ300 and R330R are significantly larger than for R2500U.
  - Roundness: The roundness values for all soot types and mobility sizes are very similar and comparable. Can this be a result of the wet dispersion? Do the authors have data for dry aerosolized BC particles for comparison?
  - Please add the standard deviation values for the reported fractal dimension.
  - Please add the standard deviation for the reported O:C. Why is the median used here and not the mean? Is the difference between 0 (400 nm R2500U) and 0.02 (400 nm COJ300) significant?
- L178-179: Form Figs. A2 and A3 it remains ambiguous how dpp was determined. More information should be added.
- L196: Fig. A5 shows a very broad size distribution, even though dm = 400 nm have been sampled. How does dva and dm relate? Please discuss in the manuscript how the broad, not monodisperse size distribution affect your conclusions for the IN measurements. A similar question arises when looking at Fig. 2.

- o Please also specify what size threshold was used for the analysis of the SPIN OPC data and discuss this in context of the shown BC size distributions. E.g. Fig. A5 suggest presence of supermicron particles, which could have been miscounted as ice crystals.
- L203: Please add references
- L211: Are these number concentrations after size selection? They seem very high.
- L222: O3 concentrations of 110 ppm seem very high. Please put this in context of typical tropospheric O3 concentrations. Can this be used to then report estimates for "Equivalent atmospheric exposure days" of O3 in your Table 2?
- L225: Replace "pyrolysis" by "photolysis"
- L228: Please specify how the reported "atmospheric aging time" of 10-15 days was calculated.
- L232: What causes the increase in RH from 16% to 25%?
- L236: Please add what the CPC number concentration was after flushing the PAM.
- L259: Delete references to Nichman et al. (2019) and Wolf et al. (2019).
- L263: Specify "centerline" of what?
- L276: Specify OPC size threshold used for AF analysis.
- L277: Please elaborate why different correction factors were applied for R2500U and R330 R and comment in the manuscript.
- Section 3.1: Please add the fraction of multiple charged particles in your population of size-selected particles to your Table 1 and discuss a potential impact on your reported IN onsets, using AF = 1% as threshold.
- L284: "deposition IN"; please see my general comment above.
- L286-287: Are the error bars based on theoretical calculations presented in the cited reference or based on measurements; please specify.
- L288: I suggest tuning down "substantially" here. Looking at your morphological parameters in Table 1 (AR, roundness, circularity), the values of the different soot types are within uncertainty of each other. Please see also my general comment above.
- L289: Please use the same precision for the given Df values as in Table 1, i.e. Df = 1.92.
- L292: Can this also simply result from uncertainty in the dpp measurements? Please see Anderson et al. (2017).
- Fig. 3:
    - o Panel A: Why does the temperature uncertainty change between the subpanels (a), (b) and (c)?
    - o Panel B: Pleas specify if the scale bar is valid for all images. The SEM image of COJ300 suggest that this soot type is the most spherical. However, the roundness values reported in Table 1 are similar to e.g. dm = 200 nm R2500U. Can the authors provide more SEM images for the Appendix to evaluate the representativeness of the shown images?
    - o Panel B: Considering the values of the roundness of the various soot types and dm = 400 nm, suggest that the morphology is the same within experimental uncertainty. In this context, the conclusion drawn that no clear dependence of IN on the particle morphology was found (e.g. L324, 451) is not surprising. However, the COJ300 soot type seems clearly more spherical on the (typical) images depicted in Fig. 3. Interestingly, at the same time this soot type reveals the largest ice nucleation activity. This trend is in-line with recent results of Mahrt et al. (2020), showing that more compacted, round soot aggregates are better ice nucleation particles (INPs) via PCF compared to more fractal soot particles. This provides further evidence that the observed ice nucleation activity in the present study is caused by PCF and might be worthwhile to consider for the discussion of the results. Do the authors have an explanation why the COJ300 aggregates look almost spherical?
    - o Panel C: It looks like there is also some O (m/z = 16) for R2500U, also this is hard to see from the scale. This figure could be improved by moving the PALMS spectra to the SI (they are only mentioned once in the manuscript) and increase the size of the panels showing the IN results, which is the major focus of this work.
- L308: Add: "… relevant to the particle..:"
- L309: "Representative uncertainty" of what?
- L310: "deposition IN"; please see my general comment above.
- L315: Add: "… R330R along the expected homogeneous freezing threshold in Fig…"
- L316: "…where the IN mode transitions from heterogeneous IN ability to homogeneous freezing…" Please consider revising this statement. During PCF, the formation of ice takes place by homogeneous freezing, i.e. PCF should not be viewed as heterogeneous ice nucleation process (Marcolli, 2020).

- L318: "higher" not "warmer" temperature
- L324-326: You might want to revisit these lines: The number of surface defects and pores for soot particles to form ice via PCF does not necessarily scale with the degree of branching of a soot aggregate. For instance, Mahrt et al. (2020) reported that cloud processes, more compacted (less fractal, even though associated with a higher fractal dimension) soot aggregates were more potent INPs to form ice via PCF. In these more compacted, round soot aggregates the probability of having a pore suitable for PCF is higher. In fact, your results are in-line with these previous findings and further support these. Your Fig. 3 reveals that COJ300 soot is most ice active and also most round based on the SEM image, with the latter being further supported by the roundness and fractal dimension values reported for COJ300.
- L331-332: Please expand your explanation on this aspect. What you show in Fig. C1 is not the saturation ratio required for pore filling (or onset of PCF), but the negative pressure resulting from keeping water within the cavity; this needs to be revised. Please also comment on how relevant the pore geometries investigated in Marcolli (2020), i.e. cylindrical and wedge-shaped pores are for the BC agglomerates in your study.
- Figs. 3 (and 5): Does each dot correspond to a single RH-scan?
- Sect. 3.2: Can you determine the coating thickness from your measurements. Adding an estimate of the coating thickness to this section would be helpful, in order to estimate for what coating thickness ice nucleation by PCF becomes inhibited.
- L337: "deposition IN"; please see my general comment above
- L346-347: It looks like the onset of toluene SOA-coated BC particles and bare BC particles is still within uncertainty.
- L349: 10-15 days is longer than the average tropospheric lifetime of soot particles. This should be reflected in your discussion.
- Fig. 6: Showing difference spectra between each panel and the bare soot shown in Fig. 6d would facilitate showing the differences between the coatings.
- L369: Rephrase to: "The IN onset…" … "… in Fig. 5b show that these particles nucleate ice …"
- L373: Can you provide a difference AMS spectra of the OH and the O3 oxidized β-caryophyllene coated BC particles?
- L384: "form" instead of "forms"
- Fig. 7:
  - o It would be helpful to add the H/C and the O/C values of the O3 oxidized β-caryophyllene soot here as well.
  - o Please verify that the suggested slope for the toluene coated SOA is correct. From your legend it looks like you should connect the white start in the lower left. corner to the blue star in the middle right. If that is the case, you need to revise your statement in L378-380.
- L392: Add: "In these studies shifts from…"
- L394-396: I do not think that it is the volatility that changes and allows a SOA particle to become glassy, but the phase state (here viscosity). Please reformulate.
- L396-398: If the suppression of the ice nucleation results from filling of the pores, the ice nucleation mechanism of the bare/uncoated particles would likely be best described by PCF, not? Please see my general comment above.
- L401: Please give details about the mass loadings used in the PAM and briefly comment on typical tropospheric SOA mass loadings.
- L412-415: This discussion should be expanded and more specific examples are needed to make the claim that the characteristics of the investigated BC types is similar to ambient soot.
- L419: The reported threshold of "dm < 200 nm" seems rather high and is misleading. For instance, Moore et al. (2017) report mode sizes of around 30 nm. While Kittelson (1998) report soot aggregates up to 200 nm (see their Fig. 11) a more profound literature search should be done here to support the given threshold.
- L223: Change to: "However, IN ability of small BC particles… may collapse forming PCF favoring…"
- L224-225: Why do you mention "surfaces" here and then talk about the "mesopores" in the next sentence.
- L225: The connection between the primary particle size and the mesopores is unclear and should be further elaborated.
- L226: I suggest tuning down: "This suggest that long-lived…INPs."
- L433: Specify as: "… growth, SOA-coated soot particles…"
- L441: Delete "monodisperse", your Fig. 2 indicates a more polydisperse sample.

- L444: Why is the R330R sample "atmospheric compacted"? The COJ300 is the most round according to your Fig. 3B and Table 1.
- L449: "deposition nucleation"; please see my general comment above. I think that in particular in the summary section you need to be careful on how to describe the ice nucleation mechanism, as in the same section (L456 and L461), you imply some ice formation via PCF.
- L451-452: See my earlier comment. The study by Mahrt et al. (2020) found more compacted soot particles to be better INP (via the PCF mechanism) compared to less compacted soot.

Anderson, P. M., Guo, H. Q. and Sunderland, P. B.: Repeatability and reproducibility of TEM soot primary particle size measurements and comparison of automated methods, J. Aerosol Sci., 114, 317–326, doi:10.1016/j.jaerosci.2017.10.002, 2017.

Bhandari, J., China, S., Chandrakar, K. K., Kinney, G., Cantrell, W., Shaw, R. A., Mazzoleni, L. R., Girotto, G., Sharma, N., Gorkowski, K., Gilardoni, S., Decesari, S., Facchini, M. C., Zanca, N., Pavese, G., Esposito, F., Dubey, M. K., Aiken, A. C., Chakrabarty, R. K., Moosmüller, H., Onasch, T. B., Zaveri, R. A., Scarnato, B. V., Fialho, P. and Mazzoleni, C.: Extensive Soot Compaction by Cloud Processing from Laboratory and Field Observations, Sci. Rep., 9(1), 11824, doi:10.1038/s41598-019-48143-y, 2019.

David, R. O., Fahrni, J., Marcolli, C., Mahrt, F., Brühwiler, D. and Kanji, Z. A.: The role of contact angle and pore width on pore condensation and freezing, Atmospheric Chem. Phys., 20(15), 9419–9440, doi:https://doi.org/10.5194/acp-20-9419-2020, 2020.

Fisher, L. R., Gamble, R. A. and Middlehurst, J.: The Kelvin equation and the capillary condensation of water, Nature, 290(5807), 575–576, doi:10.1038/290575a0, 1981.

Hoose, C. and Möhler, O.: Heterogeneous ice nucleation on atmospheric aerosols: a review of results from laboratory experiments, Atmospheric Chem. Phys., 12(20), 9817–9854, doi:10.5194/acp-12-9817-2012, 2012.

Khalizov, A. F., Zhang, R., Zhang, D., Xue, H., Pagels, J. and McMurry, P. H.: Formation of highly hygroscopic soot aerosols upon internal mixing with sulfuric acid vapor, , 114(D5), doi:10.1029/2008JD010595, 2009.

Mahrt, F., Marcolli, C., David, R. O., Grönquist, P., Barthazy Meier, E. J., Lohmann, U. and Kanji, Z. A.: Ice nucleation abilities of soot particles determined with the Horizontal Ice Nucleation Chamber, Atmos Chem Phys, 18(18), 13363–13392, doi:10.5194/acp-18-13363-2018, 2018.

Mahrt, F., Kilchhofer, K., Marcolli, C., Grönquist, P., David, R. O., Rösch, M., Lohmann, U. and Kanji, Z. A.: The Impact of Cloud Processing on the Ice Nucleation Abilities of Soot Particles at Cirrus Temperatures, J. Geophys. Res. Atmospheres, 125(3), e2019JD030922, doi:10.1029/2019JD030922, 2020.

Marcolli, C.: Technical note: Fundamental aspects of ice nucleation via pore condensation and freezing including Laplace pressure and growth into macroscopic ice, Atmospheric Chem. Phys., 20(5), 3209–3230, doi:https://doi.org/10.5194/acp-20-3209-2020, 2020.

Pruppacher, H. R. and Klett, D. J.: Microphysics of Clouds and Precipitation, 2nd edition., Kluwer Academic Publishers, Dordrecht, The Netherlands., 1997.

---

## Referee Comment (RC2) · Anonymous Referee #1 · 4 Oct 2020

**Review of "The effects of morphology, mobility size and SOA material coating on the ice nucleation activity of black carbon in the cirrus regime" by Zhang et al.**

General Comment:

This study reports the ice nucleation abilities of three different types of black carbon (BC) particles at thermodynamic conditions relevant to cirrus clouds. The importance of different physicochemical properties such as morphology, size, and SOA coating on the ice nucleation of bare BC particles was evaluated. This is a very interesting topic that nicely fits with the ACP scope. The experiments were carefully performed and the manuscript is well written. The manuscript can be accepted for its publication after the following points are properly addressed.

Major comments:

1. From the data shown in Figure 5a, it is concluded that the highly oxidized toluene SOA-coated BC (T-10) reports similar onsets as the bare BC particles. However, it is shown that the slightly oxidized toluene SOA-coated BC (T-3) deactivate the BC particles reporting higher $SS_i$ values. A clear explanation about this is missing and needs to be provided.

2. In Figure 5 important data is not reported. This information is needed to support the conclusions. a) missing bare BC data above -43°C; b) missing bare BC data at -40°C, missing D-10 data at -40°C, missing D-3 data below -43°C; and c) missing bare BC data at -40°C, and missing B-0 data below -44°C.

Minor comments:

Line 31: Add "coated BC" after "SOA".

Lines 32-33: "OH exposure levels of all SOA coating experiments, from an equivalent atmospheric 10 days to 90 days, did not render significant differences in IN potential". This is not true for Toluene SOA.

Line 62: I suggest to add some key studies in this topic instead of P&K.

Line 63: "ice nucleation".

Line 64: I suggest to add some key studies in this topic instead of the cited review paper.

Line 88: Add a reference after "weeks".

Line 89: Can the authors add an older study in addition to Kulkarni et al. (2016).

Line 102: It should be "-38°C".

Line 114: Add a reference after "combustion".

Lines 133-134: "β-caryophyllene has been found to be one of the most atmospherically abundant sesquiterpenes". Does it also apply to the upper troposphere (the focus of the present study)? What is its typical concentration at such high altitudes?

Line 145: experimental?

Lines 152-154: "All samples were then neutralized and size selected by a BMI differential mobility analyzer (BMI DMA, Model 2002; Brechtel Manufacturing Inc.) or TSI DMA (Model 3081, Classifier, Model 3082; TSI Inc.) for bare BC and BC-SOA mixing experiments, respectively". Are there any differences in the size selected particles when using the BMI or TSI DMA?

Line 160: Define "MOUDI".

Line 191: "monodisperse BC particles". Based on the data shown in Figures 2 and A3, the particles are not really monodisperse.

Lines 202-203: Please clarify the type of the AMS used.

Line 269: Add a reference after "pressure".

Lines 274-275: "signal (Garimella et al., 2016) was used to classify each particle as an inactivated aerosol or ice crystal over the course of an experiment". Could have liquid droplets been present? How was this evaluated?

Lines 279-281: "SPIN. For the size-selected bare BC experiments, the total particle number concentration was measured by a CPC operating simultaneously with SPIN, while for the SOA coating experiments, the total particle number concentration was integrated from the SMPS measurement." Are there any particle losses during the SMPS analysis?

Lines 285-286: As mentioned above the particles were not truly monodisperse.

Line 291: Add "(Table 1)" after R2500U.

Line 308: "The results in Fig. 3A demonstrate that the particle size is relevant to particle IN ability". What has been reported in the literature?

Lines 346-347: The 350 nm bare BC particles data at temperatures above -43°C on *panel a* is not shown.

Lines 347-348: The discussed data from the present study was not pure Toluene SOA. It was a mixture of BC/SOA, and the ice nucleation should be attributed to the BC and not the SOA as shown by the pure toluene SOA data. Also, do the authors claim here that all aromatic SOA should behave in the same way?

Line 348: "aromatic SOA". Indicate the SOA precursor.

Lines 350-351: I don't get the point of this sentence.

Lines 348-361: I do not fully understand the reasoning here. This discussion is useless to understand this (Major comment) as it compares the toluene-SOA with the other two SOA but does not discuss the differences between the slightly and highly oxidized toluene SOA.

Line 370: "between -46 and -42 °C". How about at -40°C?

Line 371: "between -43 and -40 °C". How about below -43°C?

Line 403: "while β-caryophyllene SOA oxidized by O3 did not alter the $SS_i$ of the soot particles". Why not? What do the authors think happened here?

Lines 403-405: this deserved to be deeper discussed.

Line 410: "aerosols". How about gasses?

Line 441: are they really monodisperse?

Figure 2. Why the 350 nm bare BC selected particles did not show a narrow peak at 350 nm?

Figures 3 and 5. I am wondering why the authors did not include literature data in both figures. This will help the readers to directly compare the present results with previous studies.

Figure 7: I think this figure add very little to the main text, and therefore, it can go to the supplementary material.

Figure A3. Why are the size distribution so polydisperse after size selection?

Figure B1. This figure is not easy to follow. Is it time in the x-axis?

Figure B2. It is not mentioned/discussed in the text.

---

## Author Comment (AC1) · 30 Oct 2020

Dear editors,

We would like to thank the referees for their careful reading and highly valuable comments that substantially help to raise the quality of our paper. We have made every effort to address their comments and made necessary revisions to the manuscript. Please see below our point-by-point response to referees' comments.

For clarity, referee comments are shown in *italic font*; the response to the comments are shown in blue color, the notes and explanations for the changes in the manuscript are shown in blue with underlines; the revised contents in the manuscript are shown in green color. Our line numbers refer to the updated manuscript, not the original. Detailed corrections can be found in this file after our responses.

**Referee #1**

*General Comment:*

*This study reports the ice nucleation abilities of three different types of black carbon (BC) particles at thermodynamic conditions relevant to cirrus clouds. The importance of different physicochemical properties such as morphology, size, and SOA coating on the ice nucleation of bare BC particles was evaluated. This is a very interesting topic that nicely fits with the ACP scope. The experiments were carefully performed and the manuscript is well written. The manuscript can be accepted for its publication after the following points are properly addressed.*

We thank this referee for accurately summarising and affirming our work. Below, the referee outlines major concerns on our toluene SOA coating experiment data interpretation and SOA coating experiments data representation. We have made substantial revisions which we believe properly address the referee's concerns.

*Major comments:*

*1. From the data shown in Figure 5a, it is concluded that the highly oxidized toluene SOA-coated BC (T-10) reports similar onsets as the bare BC particles. However, it is shown that the slightly oxidized toluene SOA-coated BC (T-3) deactivate the BC particles reporting higher SSi values. A clear explanation about this is missing and needs to be provided.*

[Response]: Inferred from the AMS spectra in Fig. 6 and the O/C-H/C in Fig. 7, both T-10 and T-3 coating experiments generated highly oxidized SOA, among which a significant fraction of oligomers might present and induce IN below homogeneous freezing threshold, as stated between L406-431. However, the fraction of oligomers and their phase state cannot be determined from current experiments and data. We can just infer potential SOA formation pathway from the mass spectra and O/C-H/C diagrams. This could be an interesting topic to investigate in future studies since toluene is one of the dominant anthropogenic aromatic emission species. Therefore, we added "BC coated by ~10-15 equivalent days atmospherically

oxidized toluene SOA (*T-3* in Table 2) is less IN active than those coated by highly oxidized toluene SOA (*T-10*) at around -46 °C and -43 °C. This might also be attributed the oxidation level and the corresponding phase state difference between the SOA generated from *T-10* and *T-3* experiments, which is beyond the scope of this study and requires further detailed phase transition study for toluene SOA." between L422-426. Besides, we tune down the temporary conclusion drawn at the end of this paragraph. More detailed information can be found in the following point-by-point response. The text in Sect. 4 and 5, as well in abstract has been rephrased accordingly.

L406-429: "The higher O/C ratio (Fig. 7) of toluene-derived SOA may enhance the hygroscopicity of the particle (Lambe et al., 2011b; Zhao et al., 2016; Liu et al., 2018) with a potential to form aqueous film on BC surface and reduce the IN ability of BC particles. On the other hand, Hinks et al. (2018) showed that toluene-derived SOA contained a significant amount of oligomers under dry laboratory conditions similar to what we conducted in the PAM chamber in this study. Volkamer et al. (2001) proved that glyoxal, which could facilitate the oligomerization as described by Hinks et al. (2018), can be produced from toluene reaction with OH as highly oxidized products. The slopes of T-10 and T-3 in Fig. 7 lie between 0-1 (0.71 and 0.99 for T-10 and T-3, respectively), suggests that oxygen and hydrogen atom addition accompanied by carbon-carbon double bond as well as benzene ring breakage might have happened during the toluene photooxidation experiments. These large oligomers could potentially reduce the hygroscopicity and alter the phase state of toluene SOA to be semi-solid or solid within the temperature range we investigated in this work (DeRieux et al., 2018; Zhang et al., 2018c; Li et al., 2020), under which the SOA can still nucleate ice (Murray et al., 2010; Berkemeier et al., 2014; Zhang et al., 2019b). The toluene SOA in *T-10* with O:C ratio over 1 (Fig. 7) has most likely already transited into solid or semi-solid glassy state at the temperature range we investigated before entering SPIN (DeRieux et al., 2018). Therefore, it is very likely that BC particles are mostly sticked with or embedded in these glassy SOA with some bare BC part exposure. Ice crystals may therefore form on the carbonaceous part of partially coated particles, whose IN onset $SS_i$ should be the same as bare COJ300. At temperatures above -43 °C, toluene SOA-coated BC particles nucleate ice at $SS_i$ ~0.1 to 0.15 above bare 350 nm COJ300, but still ~0.15 below the homogeneous freezing threshold. This might be due to the hygroscopicity enhancement of the toluene SOA-coated BC in *T*-10. BC coated by ~10-15 equivalent days atmospherically oxidized toluene SOA (*T-3* in Table 2) is less IN active than those coated by highly oxidized toluene SOA (*T-10*) at around -46 °C and -43 °C. This might also be attributed the oxidation level and the corresponding phase state difference between the SOA generated from *T-10* and *T-3* experiments, which is beyond the scope of this study and requires further detailed phase transition study for toluene SOA. Overall, two competing effects, i.e. the hygroscopicity enhancement deactivating BC IN ability, together with toluene SOA glass phase transition producing sticked or partly coated BC, may make our toluene SOA coating IN onset move towards but not fully in the homogeneous freezing regime. The toluene SOA coated diesel combustion BC (Kulkarni et al., 2016), however, nucleate ice near the homogeneous threshold, as indicated in Fig. 3(a). This might be due to the much thicker (90 nm) organic coating in their study compared to the 25 nm coating of this work, leading to a complete coverage of BC surface."

*2. In Figure 5 important data is not reported. This information is needed to support the conclusions. a) missing bare BC data above -43°C; b) missing bare BC data at -40°C, missing D-10 data at -40°C, missing D-3 data below -43°C; and c) missing bare BC data at -40°C, and missing B-0 data below -44°C.*

[Response]: The missing bare BC data in Fig. 5(a) is blocked by the legend. Fig. 5(a) has been modified and the bare BC data above -43°C is now visible.

For consistency, the IN onset was determined from consecutive RH scans. The missing SOA coating experiment data in Fig. 5(b) and 5(c) were excluded on purpose due to instrument issues, such as SPIN inlet clog at the end of the day, during these experiments.

[Figure]

**Figure R1 1 Modified Fig. 5**

*Minor comments:*

*Line 31: Add "coated BC" after "SOA".*

[Response]: Added as suggested.

*Lines 32-33: "OH exposure levels of all SOA coating experiments, from an equivalent atmospheric 10 days to 90 days, did not render significant differences in IN potential". This is not true for Toluene SOA.*

[Response]: The word 'all' has been replaced by "*n*-dodecane and *β*-caryophyllene". Besides, we added "Slightly oxidized toluene SOA coating seemed to have a stronger deactivation effect on BC IN ability than highly oxidized toluene SOA, which might be caused by oligomer formation and phase transition of toluene SOA under different oxidation levels." between L31-32.

L35-36: "OH exposure levels of *n*-dodecane and *β*-caryophyllene SOA coating experiments, from an equivalent atmospheric 10 to 90 days, did not render significant differences in IN potential."

*Line 62: I suggest to add some key studies in this topic instead of P&K.*

[Response]: Reference modified. Added "which is also confirmed by recent heterogeneous IN experiments with various INPs (e.g., Welti et al., 2009; Lüönd et al., 2010; Marcolli, 2014; Mason et al., 2016; Mahrt et al., 2018; Nichman et al., 2019)" after the P&K reference.

IN studies (e.g., Welti et al., 2009; Lüönd et al., 2010; Mason et al., 2016; Mahrt et al., 2018; Nichman et al., 2019) focusing on size effect of ice nucleating particle (INP) have been added.

L69-71: "It is widely acknowledged that larger particles act as more efficient INPs (e.g., Pruppacher and Klett, 2010), which is also confirmed by recent heterogeneous IN experiments with various INPs (e.g., Welti et al., 2009; Lüönd et al., 2010; Marcolli, 2014; Mason et al., 2016; Mahrt et al., 2018; Nichman et al., 2019)."

*Line 63: "ice nucleation".*

[Response]: Added "ice" before "nucleation".

We meant to refer to a wider range of nucleation studies apart from ice nucleation because the surface area and active sites theory is a widely accepted nucleation theory in phase transition field with various working fluids other than water here. Since we are talking about ice nucleation in this paper, it is more appropriate and concise to add ice.

*Line 64: I suggest to add some key studies in this topic instead of the cited review paper.*

[Response]: Reference modified. Rephrased to "Although the mechanism remains uncertain, one common theory is that ice nucleation probability and rate are positively correlated to particle surface area active sites density (Fletcher, 1960, 1969); therefore, an empirical parameter that is relevant to particle size (e.g., Connolly et al., 2009; Kiselev et al., 2017)".

The classical ice nucleation active sites papers (Fletcher, 1960, 1969) and the surface active sites-particle size correlation papers (e.g., Connolly et al., 2009; Kiselev et al., 2017) have been added.

L71-74: "Although the mechanism remains uncertain, one common theory is that IN ability and rate are positively correlated to particle surface active sites density (Fletcher, 1960, 1969), an empirical parameter that is relevant to particle size (e.g., Connolly et al., 2009; Kiselev et al., 2017)."

*Line 88: Add a reference after "weeks".*

[Response]: Added (Cape et al., 2012; Lund et al., 2018). Also added "or even months in the tropopause (Pusechel et al., 1992; Yu et al., 2019)".

L99-101: "These particles can remain suspended in the atmosphere for days to weeks (Cape et al., 2012; Lund et al., 2018), or even months in the tropopause (Pusechel et al., 1992; Yu et al., 2019)…"

*Line 89: Can the authors add an older study in addition to Kulkarni et al. (2016).*

[Response]: Added (Jacobson, 2001; Zhang et al., 2008; China et al., 2015b; Kulkarni et al., 2016; Zhang et al., 2018b).

L101-103: "…as well as oxidation, can lead to complex secondary organic aerosol (SOA) coatings (Jacobson, 2001; Zhang et al., 2008; China et al., 2015b; Kulkarni et al., 2016; Zhang et al., 2018b)."

*Line 102: It should be "-38°C".*

[Response]: Corrected.

*Line 114: Add a reference after "combustion".*

 [Response]: Added reference for R2500U (Joyce and Henry, 2006) and Cabot safety datasheet for R330R (Cabot, 2020)().

L126-127: "R2500U and R330R are carbonaceous black pigment powder generated by incomplete combustion (Joyce and Henry, 2006; Cabot)."

*Lines 133-134: "β-caryophyllene has been found to be one of the most atmospherically abundant sesquiterpenes". Does it also apply to the upper troposphere (the focus of the present study)? What is its typical concentration at such high altitudes?*

[Response]: There is no direct $\beta$-caryophyllene observation data available in the tropopause to the authors' knowledge. However, there is one field observation of $\beta$-caryophyllene SOA presence at 1534 m, with a terpene organic carbon concentration of about 0.6 µg m$^{-3}$ (Fu et al., 2009). Besides, previous field studies found that a considerable fraction of ambient organic aerosol is composed of $\beta$-caryophyllene SOA (Hu et al., 2008), and the concentrations are highly temperature sensitive (Arey et al., 1991; Helmig et al., 2006; Sakulyanontvittaya et al., 2008),

i.e., tropical areas with stronger convection also have higher $\beta$-caryophyllene emission. Therefore, it is reasonable to infer that the strong updrafts in tropical areas or during summer times might bring a significant amount of $\beta$-caryophyllene to the upper troposphere or lower stratosphere. Moreover, the $\beta$-caryophyllene concentration used in our experiments (2300/5000 PPB) is in line with that used in previous studies (190-200000 ppb) (Grosjean et al., 1993; Jaoui et al., 2007; Jaoui et al., 2013).

L149-157: "Terpenes are biogenic organics emitted by plants, among which $\beta$-caryophyllene has been found to be one of the most atmospherically abundant sesquiterpenes originating from agricultural plants and pine trees, as well as other sources (Arey et al., 1991; Ciccioli et al., 1999; Helmig et al., 2006; Sakulyanontvittaya et al., 2008; Guenther et al., 2012; Henrot et al., 2017). Even though the atmospheric abundance of $\beta$-caryophyllene is not as significant as other biogenic organics, such as isoprene and α-pinene, its high reactivity towards ozone and hydroxyl radical to form oxidized products with low volatility makes $\beta$-caryophyllene an appreciable biogenic SOA source in the atmosphere (Shu and Atkinson, 1995; Calogirou et al., 1997; Hoffmann et al., 1997; Griffin et al., 1999; Helmig et al., 2006; Lee et al., 2006; Jaoui et al., 2007). An up to 7 ng m$^{-3}$ atmospheric concentration of $\beta$-caryophyllene tracer during summer time was reported (Jaoui et al., 2007)."

*Line 145: experimental?*

[Response]: Changed to "experimental apparatus".

*Lines 152-154: "All samples were then neutralized and size selected by a BMI differential mobility analyzer (BMI DMA, Model 2002; Brechtel Manufacturing Inc.) or TSI DMA (Model 3081, Classifier, Model 3082; TSI Inc.) for bare BC and BC-SOA mixing experiments, respectively". Are there any differences in the size selected particles when using the BMI or TSI DMA?*

[Response]: The BMI DMA was used to select 100-400 nm BC particles (i.e., the results showed in Sect. 3.1), while the TSI DMA was used to select 350 nm COJ300 particles during SOA coating experiments (i.e., the results in Sect. 3.2). During the SOA coating experiments, the BMI SMPS was used to confirm the BC size distribution after TSI DMA selection. We consistently used the same experimental apparatus throughout a series of experiments, and therefore the IN onset results shown in Sect. 3.1 and 3.2 are unaffected. Therefore, the difference between size-selected particles is not our concern as long as the particles size distribution peaks around 350 nm.

However, we did calibrate the TSI SMPS and BMI SMPS with 350 nm COJ300 particles. The BMI and TSI SMPSs have been intercompared by scanning 350 nm COJ300. The sheath-to-sample ratio were set to 6:1 for size-selecting DMAs, and the sheath-to-sample ratio were set to 10:1 to perform standard size scans for the SMPSs during instrument intercomparison. There is minor difference between these two systems in terms of particle size selection. The plot below exhibits the instrument intercomparison result.

[Figure]

**Figure R1 2. Intercomparison of BMI and TSI SMPS by scanning the size distribution of bare 350 nm COJ300.**

*Line 160: Define "MOUDI".*

[Response]: Changed to "Micro-Orifice Uniform Deposit Impactor (MOUDI)".

L185: "…with a Micro-Orifice Uniform Deposit Impactor (MOUDI, Model M135-10; TSI Inc.) …"

*Line 191: "monodisperse BC particles". Based on the data shown in Figures 2 and A3, the particles are not really monodisperse.*

[Response]: Changed to "size-selected".

*Lines 202-203: Please clarify the type of the AMS used.*

[Response]: Changed to "High-Resolution Time-of-Flight Aerosol Mass Spectrometry (HR-ToF-AMS; Aerodyne Research Inc.)".

We used a SP-AMS for SOA coating experiment (Onasch et al., 2012). But only the AMS part was used, i.e. a standard HR-ToF-AMS (DeCarlo et al., 2006). The reference has been modified accordingly.

L232-234: "Chemical composition of the SOA-coated BC particle stream was analyzed online by PALMS and a High-Resolution Time-of-Flight Aerosol Mass Spectrometry (HR-ToF-AMS; Aerodyne Research Inc.). More details about the AMS can be found in literatures (DeCarlo et al., 2006; Onasch et al., 2012), here a brief introduction will be given."

*Line 269: Add a reference after "pressure".*

[Response]: Added (Borgnakke and Sonntag, 2013; Steane, 2016).

*Lines 274-275: "signal (Garimella et al., 2016) was used to classify each particle as an inactivated aerosol or ice crystal over the course of an experiment". Could have liquid droplets been present? How was this evaluated?*

[Response]: We have several lines of evidence that allow us to rule out droplet breakthrough during our experiments.

First of all, SPIN was operated in the deposition freezing regime, at temperatures below the threshold for homogeneous freezing and relative humidities below liquid water saturation. Aerosol would therefore be expected to freeze heterogeneously or homogeneously below liquid water saturation.

Secondly, SPIN has an evaporation section before ice crystals entering optical particle counter. The temperature of evaporation section is set so that the droplet will evaporate while ice crystals remain. Besides, the $SS_i$ at which liquid water saturates at our experiment condition is ~1.45-1.57, and liquid droplet breakthrough requires even higher supersaturation beyond the highest $SS_i$ in our experiments (Garimella et al., 2016).

Thirdly, both the laser scattering and polarity signals collected from SPIN OPC were used in this study to train a machine learning algorithm as described in Garimella et al. (2016), to robustly distinguish different phases, as stated between L304-306.

Therefore, we are confident that liquid droplets did not present during our experiments.

*Lines 279-281: "SPIN. For the size-selected bare BC experiments, the total particle number concentration was measured by a CPC operating simultaneously with SPIN, while for the SOA coating experiments, the total particle number concentration was integrated from the SMPS measurement." Are there any particle losses during the SMPS analysis?*

[Response]: The particle in bins smaller than 200 nm was corrected by using a log-normal fit. We conducted several COJ300 BC scans before injecting organics each day. The difference between integrated SMPS particle concentration during experiments and pure COJ300 BC concentration obtained prior experiments was within ±1%.

*Lines 285-286: As mentioned above the particles were not truly monodisperse.*

[Response]: Panel (b) and (c) of Fig. 3 have been moved to Appendix A and named as Fig. A2 and A3, respectively. Changed the caption of Fig. A3 to "…bare size-selected BC particles with a modal size around 400 nm."

L548-549: "Representative negative-ion PALMS mass spectra of bare size selected (a) R2500U, (b) COJ300, and (c) R330R BC particles with a modal size around 400 nm."

*Line 291: Add "(Table 1)" after R2500U.*

[Response]: Added as suggested.

*Line 308: "The results in Fig. 3A demonstrate that the particle size is relevant to particle IN ability". What has been reported in the literature?*

[Response]: Added "ability below homogeneous freezing threshold, consistent with the BC heterogeneous IN ability enhancement triggered by increasing particle size in previous studies (Mahrt et al., 2018; Nichman et al., 2019)".

The literature discussion was originally placed after the description of our results.

L348-350: "The results in Fig. 3(a) demonstrate that the particle size is relevant to the particle IN ability, consistent with the BC heterogeneous IN ability enhancement triggered by increasing particle size in previous studies (Mahrt et al., 2018; Nichman et al., 2019)."

*Lines 346-347: The 350 nm bare BC particles data at temperatures above -43°C on panel a is not shown.*

[Response]: Please see our response to the second major comment. The missing bare BC data in Fig. 5(a) was blocked by the legend. Fig. 5(a) has been modified and the bare BC data above -43°C is now visible.

*Lines 347-348: The discussed data from the present study was not pure Toluene SOA. It was a mixture of BC/SOA, and the ice nucleation should be attributed to the BC and not the SOA as shown by the pure toluene SOA data. Also, do the authors claim here that all aromatic SOA should behave in the same way?*

[Response]: We agree with the referee that our IN data originates from a mixture of bare BC, partially coated BC, fully coated BC, and pure toluene SOA.

The toluene SOA in *T-10* with O:C ratio over 1 (Fig. 7) has most likely already transited into solid or semi-solid glassy state at the temperature range we investigated before entering SPIN (DeRieux et al., 2018). Volkamer et al. (2001) proved that glyoxal, which could facilitate the oligomerization as described by Hinks et al. (2018), can be produced from toluene reaction with OH as highly oxidized products. The slopes of T-10 and T-3 in Fig. 7 lie between 0-1 (0.71 and 0.99 for T-10 and T-3, respectively), suggests that oxygen and hydrogen atom addition accompanied by carbon-carbon double bond as well as benzene ring breakage might have happened during the toluene photooxidation experiments. These large oligomers could potentially reduce the hygroscopicity and alter the phase state of toluene SOA to be semi-solid or solid within the temperature range we investigated in this work (DeRieux et al., 2018; Zhang et al., 2018c; Li et al., 2020)

Therefore, it is very likely that BC particles are mostly sticked with or embedded in these glassy SOA with some bare BC part exposure. We hence have modified the discussion statement and attributed the *T-10* IN results to the uncoated bare BC part.

However, slightly oxidized toluene SOA with smaller O:C ratio requires colder glass transition temperature (DeRieux et al., 2018). The toluene SOA generated in *T-3* experiment might still be liquid and entirely enclose BC particles, forming an organic liquid coating film and deactivate BC particles.

The comparison with pure aromatic SOA (Wang et al., 2012) has been modified. Please see the response to major comment #1.

L391-428: "The toluene SOA mass spectrum in Fig. 6(a) exhibits higher $m/z = 44$ (COO$^-$) and lower $m/z = 43$ (C$_3$H$_7^-$) fraction signal, indicating more oxidized organic species were generated during *T-10* and *T-3* experiments (Lambe et al., 2011b), agreeing with the previous study on

toluene SOA (Liu et al., 2018). The higher O/C ratio (Fig. 7) of toluene-derived SOA may enhance the hygroscopicity of the particle (Lambe et al., 2011b; Zhao et al., 2016; Liu et al., 2018) with a potential to form aqueous film on BC surface and reduce the IN ability of BC particles. On the other hand, Hinks et al. (2018) showed that toluene-derived SOA contained a significant amount of oligomers under dry laboratory conditions similar to what we conducted in the PAM chamber in this study. Volkamer et al. (2001) proved that glyoxal, which could facilitate the oligomerization as described by Hinks et al. (2018), can be produced from toluene reaction with OH as highly oxidized products. The slopes of T-10 and T-3 in Fig. 7 lie between 0-1 (0.71 and 0.99 for T-10 and T-3, respectively), suggests that oxygen and hydrogen atom addition accompanied by carbon-carbon double bond as well as benzene ring breakage might have happened during the toluene photooxidation experiments. These large oligomers could potentially reduce the hygroscopicity and alter the phase state of toluene SOA to be semi-solid or solid within the temperature range we investigated in this work (DeRieux et al., 2018; Zhang et al., 2018c; Li et al., 2020), under which the SOA can still nucleate ice (Murray et al., 2010; Berkemeier et al., 2014; Zhang et al., 2019b). The toluene SOA in *T-10* with O:C ratio over 1 (Fig. 7) has most likely already transited into solid or semi-solid glassy state at the temperature range we investigated before entering SPIN (DeRieux et al., 2018). Therefore, it is very likely that BC particles are mostly sticked with or embedded in these glassy SOA with some bare BC part exposure. Ice crystals may therefore form on the carbonaceous part of partially coated particles, whose IN onset $SS_i$ should be the same as bare COJ300. At temperatures above -43 °C, toluene SOA-coated BC particles nucleate ice at $SS_i$ ~0.1 to 0.15 above bare 350 nm COJ300, but still ~0.15 below the homogeneous freezing threshold. This might be due to the hygroscopicity enhancement of the toluene SOA-coated BC in *T*-10. BC coated by ~10-15 equivalent days atmospherically oxidized toluene SOA (*T-3* in Table 2) is less IN active than those coated by highly oxidized toluene SOA (*T-10*) at around -46 °C and -43 °C. This might also be attributed the oxidation level and the corresponding phase state difference between the SOA generated from *T-10* and *T-3* experiments, which is beyond the scope of this study and requires further detailed phase transition study for toluene SOA. Overall, two competing effects, i.e. the hygroscopicity enhancement deactivating BC IN ability, together with toluene SOA glass phase transition producing sticked or partly coated BC, may make our toluene SOA coating IN onset move towards but not fully in the homogeneous freezing regime. The toluene SOA coated diesel combustion BC (Kulkarni et al., 2016), however, nucleate ice near the homogeneous threshold, as indicated in Fig. 3(a). This might be due to the much thicker (90 nm) organic coating in their study compared to the 25 nm coating of this work, leading to a complete coverage of BC surface."

*Line 348: "aromatic SOA". Indicate the SOA precursor.*

[Response]: The precursor used in (Wang et al., 2012) was naphthalene. We rephrased this paragraph, please see the response to major comment #1 and last comment.

*Lines 350-351: I don't get the point of this sentence.*

[Response]: Deleted.

*Lines 348-361: I do not fully understand the reasoning here. This discussion is useless to understand this (Major comment) as it compares the toluene-SOA with the other two SOA but does not discuss the differences between the slightly and highly oxidized toluene SOA.*

[Response]: This whole paragraph has been reformulated. Please see the response to major comment #1 and the comment on L347-348.

*Line 370: "between -46 and -42 °C". How about at -40°C?*

[Response]: The lamina temperate in Fig. 5(b) was between -46.6 °C to -41.6 °C for *D-10*. We draw a rather conservative conclusion here. Please also see the response to major comment 2.

The data was excluded due to SPIN operating issues that day.

*Line 371: "between -43 and -40 °C". How about below -43°C?*

[Response]: Please see the response to major comment 2.

The data was excluded due to SPIN operating issues that day.

*Line 403: "while β-caryophyllene SOA oxidized by O3 did not alter the SSi of the soot particles". Why not? What do the authors think happened here?*

[Response]: $O_3$ addition on $\beta$-caryophyllene carbon-carbon double bond leads to formation of organics that have super-low volatility that could be semi-solid or solid (Nguyen et al., 2009; Winterhalter et al., 2009). As with the toluene SOA discussion above, BC particles might be sticked with or embedded in such semi-solid or solid SOA, leaving some bare carbonaceous surface that can nucleate ice in the way bare COJ300 BC particles do.

L465-457: "while $\beta$-caryophyllene SOA oxidized by $O_3$ did not alter the $SS_i$ of the soot particles. In addition, more oxidized SOA (toluene derived SOA from photooxidation) with potentially more oligomer formation, moving IN onset $SS_i$ towards, but still below, homogeneous freezing."

*Lines 403-405: this deserved to be deeper discussed.*

[Response]: We gratefully thank the referee to suggest us revisit these lines and make the discussion deeper here. The IN onset description has been moved to the end of this paragraph and discussion has been added.

Fig.7 (O/C and R/C plot) was modified by adding B-0 and connect pure $\beta$-caryophyllene with B-0. The slope between 0-1 is in compliance with carbon-carbon double bond breakage, and oxygen and hydrogen atoms addition.

L444-450: "However, COJ300 BC coated with $O_3$ oxidized $\beta$-caryophyllene SOA (*B-0* in Table 1) shows no significantly alternation of IN ability, as shown in Fig. 5(c). Unlike OH oxidation of $\beta$-caryophyllene where fragmentation happens, $O_3$-addition is very likely to happen first on the carbon-carbon double bond of $\beta$-caryophyllene in *B-0*, leading to formation of semi-sold or solid SOA (Nguyen et al., 2009; Winterhalter et al., 2009). As with the case of *T-10*, such semi-solid

or solid SOA might collide and stick with BC particles, leaving some bare carbonaceous surface that can nucleate ice following the IN pattern of COJ300 BC."

[Figure]

**Figure R1 3 Modified Fig.7.**

*Line 410: "aerosols". How about gasses?*

[Response]: Changed to "organic species" to be consistent.

*Line 441: are they really monodisperse?*

[Response]: Deleted.

To be more cautious and rigorous, the word "monodisperse" has been replaced by "size-selected" throughout the paper.

*Figure 2. Why the 350 nm bare BC selected particles did not show a narrow peak at 350 nm?*

[Response]: To highlight the modal size shift, the size range of x axis in Fig.2 is 300 to 500 nm, which stretches the figure and makes the size distribution seem very broad. The figure below shows a narrow peak and modal size shift for *T-10* and *B-10* experiment as examples.

Updated Fig. 2.

[Figure]

**Figure R1 4. Particle size distribution for T-10 and B-10 experiments.**

*Figures 3 and 5. I am wondering why the authors did not include literature data in both figures.*

*This will help the readers to directly compare the present results with previous studies.*

[Response]: Added the 800 nm R330R and R2500U IN data from (Nichman et al., 2019) in Fig. 3(a) and (b), and toluene SOA coated diesel BC IN data from (Kulkarni et al., 2016) in Fig. 5(a). To the author's knowledge, there is no data available for *n*-dodecane and *β*-caryophyllene SOA coated BC IN experiments.

*Figure 7: I think this figure add very little to the main text, and therefore, it can go to the supplementary material.*

[Response]: Please see our response to the comment on L403-405. Fig. 7 is crucial when inferring the atomic change during the experiments. Therefore, we modified Fig. 7 and keep the figure in main text.

*Figure A3. Why are the size distribution so polydisperse after size selection?*

[Response]: This figure is not the size distribution of BC particles. It is the primary particle (the basic carbonaceous spherules forming BC aggregates) size distribution of size-selected BC aggregates, corresponding to dpp and its standard deviation in Table 1. The primary particle size also obeys normal distribution (Huang and Vander Wal, 2013; Liati et al., 2014)

*Figure B1. This figure is not easy to follow. Is it time in the x-axis?*

[Response]: This figure is the temporal evolution of particle size distribution, corresponding to the top view of the plots below. Each column of x-axis in Fig. B1 corresponds to a complete BMI SMPS scan, which is equivalent to time. Combining the y-axis and colormap together we can see the size distribution for each scan.

Added "The x axis denotes time. Together with the color map, y axis shows the size distribution for a certain time." in the figure caption.

[Figure]

**Figure R1 5. BMI SMPS scans for *T-10* (left) and *B-10* (right) experiments exhibiting modal size shift from 350 nm to 400 nm as a function of scan number.**

*Figure B2. It is not mentioned/discussed in the text.*

[Response]: This figure is placed here to demonstrate that the AMS signals of our SOA results are in line with previous studies.

**Referee #2**

*The work by Zhang et al. presents laboratory data of the ice nucleation ability of different types of black carbon, along with a characterization of the physical and chemical particle properties. The study extends previous laboratory measurements on the ice nucleation ability of soot. This topic fits well within the scope of ACP, and the results are novel and of great interest for the ice nucleation community. Overall, I find the manuscript is well written and structured and the results are presented in a clear way and put into context of previous literature. However, from my point of view there are some aspects, in particular in the discussion of the ice nucleation results, that should be revised. Once the points outlined below are sufficiently addressed, I support publication of the manuscript in ACP.*

We thank this referee for recognizing the contribution of our work to the community. Below, the referee outlines major concerns regarding our usage of IN terms and uncertainty discussion. We highlight several substantial revisions which we believe address the referee's concerns properly and sufficiently.

*Overall comments and general concerns:*

- *The authors do not seem to provide a consistent view of the ice nucleation mechanism of the investigated soot types, which I find in part hard to follow. At various instances throughout the manuscript the authors describe the observed ice nucleation activity to result from deposition nucleation (e.g. L25, 27, 68, 90, 284). At other instances, the authors describe how earlier studies by Nichman et al. (2019; including authors from the present study) and Mahrt et al. 2018 (L80-84) have identified soot to form ice via the pore condensation and freezing (PCF) mechanism at cirrus cloud temperatures, the temperature range studied here. The PCF mechanism is then suggested to explain the observed ice nucleation results (e.g. L228-332) and also used to argue how the coating of the soot particles with secondary organic aerosol (SOA) leads to filling of the pore, which*

*renders these pores unable to contribute to ice formation via PCF (e.g. L396-397). This seems to be in stark contradiction to the suggested ice formation via deposition nucleation for some of the bare soot particles. In their discussion (L438-439 and L460) the authors then vaguely talk about "verifying the PCF" mechanism in future studies but it does not become fully clear if the ice nucleation observed in the present study is attributed to PCF or to "classical" deposition nucleation.*

*I assume that the authors interpret the observed ice formation as PCF and whenever the authors refer to "deposition nucleation", they refer to ice formation at cirrus temperatures below water saturation (and below homogeneous freezing conditions, to be specific). However, this terminology is confusing for the reader, and suggests a flip-flop between both mechanisms, i.e. classical deposition nucleation and PCF. I would appreciate if the authors could provide a clearer description of the ice nucleation mechanism of the observed ice formation on the investigated soot particles. A relatively simple solution to reduce the in part confusing back and forth would be to use just "IN activity" rather than "deposition IN activity" to describe the ice formation below homogeneous freezing conditions and then provide a comprehensive discussion of both "classical" deposition nucleation and PCF in the results section.*

[Response]: We agree with the referee that the mixing usage of deposition IN and PCF might be misleading and confusing. The terms have been checked and replaced throughout the paper. Discussion of IN mechanism has been added where applicable. Please see below the point-to-point response for more details.

- *The title of the presented manuscript suggests that an effect of particle size, morphology and coating on the ice nucleation ability of BC is investigated in this study. While I appreciate the efforts in comprehensively and carefully characterizing the physical and chemical properties of the investigated soot types the associated uncertainties are not sufficiently discussed, which I find problematic. For example, the goodness of the size selection and hence the difference between a 300 nm and a 400 nm R2500U particle are not discussed. Upon revision of the manuscript, the uncertainties of the various measurements of these variables on the dependent variable (ice nucleation activity), should be more clearly emphasized and discussed. As another example, I am not clear whether the O:C ratio of the COJ300 and the R2500U samples are really different.*

[Response]: Added uncertainty and confidence intervals where applicable. The uncertainty of size selection, morphology and chemical composition characterization has been added and discussed. Please see the point-to-point response to comment on Table 1 and the relevant lines for detailed information.

*Specific comments:*

- *L22: Delete "aerosolized"*

[Response]: Deleted as suggested.

- *L25: Delete "(-24 to -46°C)"*

[Response]: Deleted.

- *L25,27: "Deposition IN"; please see my general comment above.*

[Response]: Replaced "Deposition IN" by "Ice formation at cirrus temperatures below homogeneous freezing thresholds" in L25.

Deleted "deposition" in L27.

- *L31: "We attribute the inhibition of IN ability to the filling of pores on the BC surface by the SOA material coating." In other words, the IN ability of the uncoated particles results from the pores, i.e. takes place by PCF. This is in contradiction to the "deposition IN" mentioned above and at various other instances in the manuscript. Please see my general comment above.*

[Response]: We agree with the referee. Changes have been made accordingly.

L37-39: "Our study of selected BC types and sizes suggests that increase in diameter, compactness, and/or surface oxidation of BC particles lead to more efficient IN via pore condensation freezing (PCF) pathway, and that the organic coating materials can inhibit the formation of ice."

- *L33: Delete first "days"*

[Response]: Deleted as suggested.

- *L37: Delete "terrestrial"*

[Response]: Deleted.

- *L39,41: I suggest replacing the given references by primary reference, e.g. the classical textbook by Pruppacher and Klett (1997).*

[Response]: References are replaced by (Pruppacher and Klett, 2010)

- *L43: Why are you only introducing "deposition IN" here and not also PCF?*

[Response]: Added a brief introduction of PCF.

PCF was originally introduced in L83 after description of the experiment results that doubt the classical deposition IN mode.

L47-52: "Deposition IN is one classical heterogeneous IN mode, in which solid ice is formed by direct water vapor deposition on to an INP surface. Recently, laboratory studies demonstrated that ice formation at thermodynamic conditions relevant to classical deposition IN on porous material at cirrus temperature might actually be initiated by homogeneous freezing of liquid water held within the cavities below water saturation due to inverse Kelvin effect (Marcolli, 2014; David et al., 2019; David et al., 2020). This pathway by which porous material might form ice below water saturation below -38 °C is referred to as pore condensation and freezing (PCF, Marcolli, 2014)."

- *L45: Rephrase as: "BC from aircraft emissions may be an ..."*

[Response]: modified as suggested.

- *L53: Delete "simulate the atmospheric environment"*

[Response]: Deleted.

- *L59: Hoose and Möhler (2012) is not a primary reference and I suggest deleting it here.*

[Response]: Deleted as suggested.

- *L63-64: Similarly, the probability of a BC aggregate to contain a pore with the right properties (e.g. pore size and contact angle) increases with increasing aggregate*

*diameter, which would favor PCF for larger particles (Mahrt et al., 2018). This could be added.*

[Response]: Added as suggested. Since contact angle is a bulk quantity, we replaced the word "contact angle" by "surface hydrophilicity" here to be more rigorous and consistent.

L74-75: "Similarly, the probability of a BC aggregate to contain a pore with the right properties (e.g., pore size and surface hydrophilicity) increases with increasing aggregate diameter, which would favor PCF for larger particles (Mahrt et al., 2018)."

- *L68: "deposition mode"; see my general comment above.*

[Response]: Deleted "deposition" in L77. Replaced "…in deposition…" in L79 by "…at thermodynamic conditions relevant to classical deposition mode at cirrus temperature".

L78-80: "However, the size threshold below which BC cannot nucleate ice at thermodynamic conditions relevant to classical deposition mode at cirrus temperature and the underlying mechanism is still uncertain."

- *L70: Rephrase to "…during atmospheric aging, leading…"*

[Response]: Reworded as suggested.

- *L71: Add Bhandari et al. (2019) and Khalizov et al. (2009)*

[Response]: Added.

- *L80: Delete Kumfer ad Kennedy (2009)*

[Response]: Deleted.

- *L80-81: This statement is not true and needs to be revised. E.g. see Table 1 and Fig. 3 in Mahrt et al. (2018). The investigated "FS" soot has the second highest surface area, but "LB_RC" shows a similar or higher IN activity.*

[Response]: Removed Mahrt et al. (2018). Added "For smaller particles, Mahrt et al. (2018) presented a complex dependence of BC IN activity on particle size, surface area, and BC surface hydrophilicity." after this sentence.

L92-96: "Nichman et al. (2019) reported a generally positive correlation between BC particle surface area and IN activity for particles with same size. For smaller particles, Mahrt et al. (2018) presented a complex dependence of BC IN activity on particle size, surface area, and BC surface hydrophilicity. They attributed BC IN activity to PCF mechanism (Marcolli, 2014; David et al., 2019; David et al., 2020), in which IN of BC is considered as homogeneous freezing of liquid water taken up in mesopores (2-50 nm) due to capillary effect (Fisher et al., 1981)."

- *L82: Delete Koop (2017) and Marcolli (2017). Add David et al. (2020).*

[Response]: References modified as suggested.

- *L83: Rephrase to: "…in which IN of BC is considered as homogeneous freezing of…"*

[Response]: Rephrased as suggested.

- *L84: Replace references by Fisher et al. (1981).*

[Response]: Modified.

- *L90, 100: "deposition nucleation"; please see my general comment above*

[Response]: Deleted "deposition".

- *L102: "-38°C"*

[Response]: Corrected.

- *L128: Add: "… in particles of aircraft engine exhaust…"*

[Response]: Added.

- *L131: Please specify "light" aromatic species*

[Response]: Added "such as xylenes, alkylbenzenes, naphthalene, etc.)".

L144: "…a proxy for other light aromatic species (such as xylenes, alkylbenzenes, naphthalene, etc.) in atmospheric aromatic-seeded SOA formation models"

- *L133: How representative is β-caryophyllene for biogenic SOA? How does it compare to other major biogenic SOA types such as α-pinene SOA? For the Sect. 2.1.2. more discussion on the atmospheric abundance and relevance of the chosen SOA precursors should be given.*

[Response]: There is no direct $\beta$-caryophyllene observation data available in the tropopause to the authors' knowledge. However, there is one field observation of $\beta$-caryophyllene SOA presence at 1534 m, with a terpene organic carbon concentration of about 0.6 µg m$^{-3}$ (Fu et al., 2009).

In ambient environment, the $\alpha$-pinene concentration is generally higher than $\beta$-caryophyllene. However, due to the high yield of $\beta$-caryophyllene, its SOA could still be quite significant (Yee et al., 2018). Previous field studies found that $\beta$-caryophyllene SOA comprise a significant fraction of ambient organic aerosol (Hu et al., 2008), and the concentrations are highly temperature sensitive (Arey et al., 1991; Helmig et al., 2006; Sakulyanontvittaya et al., 2008), i.e., tropical areas with stronger convection also have higher $\beta$-caryophyllene emission. The deposition ice nucleation of $\alpha$-pinene SOA has been studied before and the results varies as some studies show α-pinene SOA are not efficient INPs (Ladino et al., 2014; Wagner et al., 2017) while others show they can be relatively good INPs with pre-cooling (Ignatius et al., 2016). This work mainly focuses on $\beta$-caryophyllene due to its relative importance in the ambient.

Therefore, it is reasonable to infer that the strong updrafts in tropical areas or during summer times might bring a significant amount of $\beta$-caryophyllene to the upper troposphere or lower stratosphere. Moreover, the $\beta$-caryophyllene concentration used in our experiments (2300/5000 PPB) lies in the range of that has been used in previous studies (190-200000 ppb) (Grosjean et al., 1993; Jaoui et al., 2007; Jaoui et al., 2013).

Added several lines to address the importance and abundance of n-dodecane and $\beta$-caryophyllene in the atmosphere.

L147-157: "Modelling and field studies suggest that such less volatile organic species might be significant anthropogenic SOA precursors in highly populated area (Hodzic et al., 2010; Tsimpidi et al., 2010; Lee-Taylor et al., 2011; Hodzic et al., 2016), among which *n*-dodecane has

relatively higher emission rate (Lee-Taylor et al., 2011). Terpenes are representative biogenic organics emitted by plants , among which $\beta$-caryophyllene has been found to be one of the most atmospherically abundant sesquiterpenes originating from agricultural plants and pine trees, as well as other sources (Arey et al., 1991; Ciccioli et al., 1999; Helmig et al., 2006; Sakulyanontvittaya et al., 2008; Guenther et al., 2012; Henrot et al., 2017). Even though the atmospheric abundance of $\beta$-caryophyllene is not as significant as other biogenic organics, such as isoprene and α-pinene, its high reactivity towards ozone and hydroxyl radical to form oxidized products with low volatility makes $\beta$-caryophyllene an appreciable biogenic SOA source in the atmosphere (Shu and Atkinson, 1995; Calogirou et al., 1997; Hoffmann et al., 1997; Griffin et al., 1999; Helmig et al., 2006; Lee et al., 2006; Jaoui et al., 2007). An up to 7 ng m$^{-3}$ atmospheric concentration of $\beta$-caryophyllene tracer during summer time was reported (Jaoui et al., 2007)."

- *L146: Can the authors comment on how the wet dispersion in the atomizer affects the BC particle morphology?*

[Response]: We acknowledge that generation method may affect BC morphology because of the surface tension of water. However, based on the SEM observation and shape descriptor derivation in this study, the difference between wet and dry generation BC particles are minor, as showed below. Moreover, a previous study conducted by Nichman et al. (2019) demonstrated that the IN onset for similar BC types was not very much affected by dry vs. wet generation (see their Fig. 5).

**Table R2 1.** Characteristics of wet and dry generation 400 nm R2500U BC particles

| 400 nm R2500U | Wet generation | Dry generation |
|---|---|---|
| $\overline{d_a}$ **(nm)** | 343.5±106.3 | 465.0±282.2 |
| $\overline{AR}$ | 1.44±0.29 | 1.53±0.33 |
| $\overline{Roundness}$ | 0.73±0.09 | 0.72±0.09 |
| $\overline{Circularity}$ | 0.61±0.15 | 0.62±0.25 |
| $\overline{d_{pp}}$ **(nm)** | 34.5±11.4 | 35.9±8.0 |
| **N** | 343 | 105 |
| **$D_f$** | 1.92 [1.68, 2.16] | 2.04 [1.81, 2.28] |

- *L150: The flow reported here (1.5 SLPM) and on L215 (2.2 SLPM) are different. How does this change your flow rate through the DMA and possibly affect your size selection. Please comment in the main text.*

[Response]: The statement here might be a little misleading. For clarity, we changed the description in L168-170. We used different flow rate to ensure sufficient BC concentration. Based on the DMA transfer function study by Karlsson and Martinsson (2003), the degradation due to imperfect DMA operation condition becomes less than 5% when the particle size exceeds 100 nm. Please see our response to the next comment for more discussion on DMA difference.

We also added the sheath-to-sample flow rates used in bare BC and BC-SOA mixing experiments and discussed the potential impact of the flow condition on size-selection between L176-180.

L169-171: "The flow rates through the nebulizer was 1.5 SLPM (standard liters per minute) and 2.2 SLPM for bare BC experiments and BC-SOA mixing experiments, respectively, which is controlled by a mass flow controller (MFC, Model MC-2SLPM-D; ALICAT Scientific)."

L176-180: "The sheath-to-sample ratios for bare BC and BC-SOA mixing experiments were respectively ~6:1 and 4:1. Compared with the widely used sheath-to-sample ratio (10:1, Karlsson and Martinsson, 2003), the lower ratios in our experiments might broaden the BC particle size distribution, yet still offering satisfactory number concentration at the target particle size (Fig. 2) because of the large particle size being selected (Karlsson and Martinsson, 2003)."

- *L152-153: Have the TSI and BMI DMA been compared? Can the authors provide a size distribution of the soot types investigated that was size-selected with both DMAs to ensure that there is no difference in the uncoated and coated particles due to the instrumentation used? Please also add the aerosol-to-sheath flow ratios used in the DMAs for aerosol size selection.*

[Response]: The BMI and TSI SMPSs have been intercompared by scanning 350 nm COJ300. The sheath-to-sample ratio were set to 6:1 for size-selecting DMAs, and the sheath-to-sample ratio were set to 10:1 to perform standard size scans for the SMPSs during instrument intercomparison. There is minor difference between these two systems in terms of particle size selection. The plot below exhibits the instrument intercomparison result. To minimize the influence of instrument difference on measured particle number concentration, the BMI DMA and CPC were used in bare BC experiments, and the BMI SMPS (DMA+CPC) was used to measure real-time particle size distribution in SOA coating experiments. Please also see our response to the comment on section 3.1 for more information on doubly-charged BC discussion.

[Figure]

**Figure R2 1. Intercomparison of BMI and TSI SMPS by scanning the size distribution of bare 350 nm COJ300. The grey line and black dashed lines represent first-order Gaussian fitting of the scanned particle size distribution.**

**Table R2 2.** Fitting parameters of the scanned particle size distribution with different instrument.

| Instrument | Peak | Maximum particle concentration (#/cc) | $d_m \pm \sigma_{d_m}$ | $R^2$ | Particle Number (#) |
|---|---|---|---|---|---|
| BMI SMPS | 1st | 3.74E4 | $360.81 \pm 84.78$ | 0.96 | 5790 |
| | 2nd | 7.96E3 | $600.94 \pm 129.72$ | 0.89 | 595 |
| TSI SMPS | 1st | 3.35E4 | $353.99 \pm 57.18$ | 0.95 | 3757 |

There might be some misunderstanding for the results of bare and coated BC here. To clarify: the BMI DMA was only used to select 100-400 nm BC particles (i.e., the results represented in Sect. 3.1) and the particle concentration was monitored with a BMI CPC. The TSI DMA was only used to select 350 nm COJ300 particles during SOA coating experiments (i.e., the results in Sect. 3.2), with the BMI SMPS used to monitor particle size distribution.

Since the same experimental apparatus was used consistently and separately throughout the experiments in Sect. 3.1 and 3.2, we did not mix and intercompare these IN onset results. The 350 nm bare COJ300 results in Sect. 3.2 Fig. 5 was size-selected by TSI DMA and went through PAM chamber. Therefore, the difference between these two DMAs is not our major concern and does not affect our conclusion drawn here as long as the bare COJ300 particles size distribution obtained by BMI SMPS peaks around 350 nm.

Besides, we added "The sheath to sample ratios for bare BC and BC-SOA mixing experiments were respectively ~6:1 and 4:1." between L176-177.

- *L155: An RH value of 16% seems extremely high. Please comment how this affects your IN measurements that are performed using this air stream (conserving specific humidity) but exposing it to T < -40 °C.*

[Response]: Typical ambient *RH* is about 30-80% on the ground. The *RH* is generally above 20% in the tropopause (Ruzmaikin et al., 2014) with exceptions for desert regions around N30° and S30° due to the dry subtropical high. Tropical tropopause layer occupies even higher *RH* at around 13 km. The *RH* in our experiments is at least in-line with, if not lower than, the ambient *RH*.

Besides, when the humidified air stream encounters the sharp temperature gradient at SPIN inlet, the water immediately condense and freeze in the inlet, so the high RH in the air stream should have minor effect on the ice formation inside the chamber.

- *L160: Please specify "MOUDI" acronym*

[Response]: Changed to "Micro-Orifice Uniform Deposit Impactor (MOUDI, Model M135-10; TSI Inc.)".

- *L170: Rephrase as "… by the PALMS."*

[Response]: Modified as suggested.

- *Table 1:*

    o   *da of R2500: Why is the projected area diameter for dm = 300 nm larger than for dm = 400 nm?*

[Response]: This could be possible, because 1) R2500U particles are branched fractal aggregates. The smaller (e.g. 200 and 300 nm) R2500U particles are generally more spherical and compact than the size-selected 400 nm particles, leading to larger projection area without voids; 2) there could be random errors due to manually selection of aggregates to analyse; 3) The particle orientation could affect the projection area. The round aggregates are less affected by particle orientation than the branched ones. Please also see the SEM images in Fig. A3.

Added "400 nm R2500U is more fractal than COJ300 and R330R with the same $d_m$, as with 300 nm R2500U, which is indicated by $D_f$. Project area, as well as the derived $d_a$, are significantly affected by the degree of fractal, since highly fractal particles can have voids affecting project area. COJ300 and R330R, as well as 300 nm R2500U particles, are more spherical and compact than the fractal 400 nm R2500U, leading to larger $d_a$ in comparison with 400 nm R2500U." between L215-219.

L215-219: "400 nm R2500U is more fractal than COJ300 and R330R with the same $d_m$, as well as 300 nm R2500U, which is indicated by $D_f$. Project area, as well as the derived $d_a$, are significantly affected by the degree of fractal, since highly fractal particles can have voids affecting project area. COJ300 and R330R, as well as 300 nm R2500U particles, are more spherical and compact than the fractal 400 nm R2500U, leading to larger $d_a$ in comparison with 400 nm R2500U."

    o   *da: Please comment in the text why the da values for dm = 400 nm particles for COJ300 and R330R are significantly larger than for R2500U.*

[Response]: Please see the response to last comment.

    o   *Roundness: The roundness values for all soot types and mobility sizes are very similar and comparable. Can this be a result of the wet dispersion? Do the authors have data for dry aerosolized BC particles for comparison?*

[Response]: Please see our response to the comment on L146. We did compare the morphology characteristics of both wet and dry generation R2500U BC particles. It shows that generation method has minor impact on particle morphology.

    o   *Please add the standard deviation values for the reported fractal dimension.*

[Response]: Added the 95% confidence intervals of the fitted $D_f$, embraced in brackets.

    o   *Please add the standard deviation for the reported O:C. Why is the median used here and not the mean? Is the difference between 0 (400 nm R2500U) and 0.02 (400 nm COJ300) significant?*

[Response]: Changed to mean O:C ratio and added the standard deviation.

As stated in L221, PALMS can provide qualitative chemical composition information and is operated in a single particle mode. About 21.0% COJ300 negative spectra showed significant $m/z = 16$ signal compared to merely 2.8% for R2500U in this work. The frequency of $m/z = 16$ signal appearance can reflect the probability and abundance of oxygen content in a BC type. The consistent appearance of $m/z = 16$ signals for COJ300 negative spectra is a clear indication of

higher oxygen content in comparison with R2500U. The sparse oxygen signal of R2500U could be either contamination or ion re-combination in PALMS. Therefore, the "*Spectra percentage exhibiting m/z = -16 signal*" row is added in Table 1.

Besides, the caption of Fig. A1 has been rephrased.

L527-528: "Generally, the probability of O⁻ signal presence in COJ300 is an order of magnitude higher than R2500U."

"**Table 1.** Characteristics of selected BC proxies in this study. $a_{BET-N_2}$ is the BET specific surface area based on N$_2$ adsorption isotherms; $d_m$ is the particle mobility diameter; $\overline{d_a}$ denotes the mean 2-D projected area-equivalent aggregate diameter derived from SEM images; mean aspect ratio ($\overline{AR}$), roundness ($\overline{Roundness}$) and circularity ($\overline{Circularity}$) are the geometric mean morphology parameters derived from several aggregates and are defined in Sect. 2.2.2; $\overline{d_{pp}}$ denotes the mean geometric diameter of primary particles measured from SEM images, and $N$ the number of primary particles analyzed for each BC type and size; $D_f$ denotes the 3-D fractal dimension derived from 2-D SEM images; $d_{va}$ is the particle vacuum aerodynamic diameter measured by the PALMS; values in parentheses are the corresponding one standard deviations; values in brackets are the 95% confidence intervals of $D_f$.

| BC type | R2500U | | | COJ300 | R330R |
|---|---|---|---|---|---|
| **Composition** | Furnace black | | | (4-carboxyphenyl)-modified carbon black[a] | Furnace black |
| **CAS No.** | 1333-86-4 | | | 1106787-35-2 | 1333-86-4 |
| **Specific gravity (20 °C)** | 1.7-1.9[a] | | | 1.07 (dispersion)[a] | 1.7-1.9[a] |
| **Bulk density (g/cm³)** | 20-380 | | | - | 20-380 |
| **pH** | 7.0[b]; 4-11[c] | | | 7.0-8.6[a] | 6.9[b]; 2-11[c] |
| **Solubility** | Insoluble | | | Insoluble but dispersible | Insoluble |
| $a_{BET-N_2}$ **(m²/g)[a,d]** | 270 | | | 200 | 90 |
| $d_m$ **(nm)** | 200 | 300 | 400 | 400 | 400 |
| $\overline{d_a}$ **(nm)** | 316.9 | 403.5 | 343.5 | 629.4 | 816.6 |
| **(± 1σ)** | (109.3) | (82.5) | (106.3) | (308.3) | (355.3) |
| $\overline{AR}$ | 1.22 | 1.36 | 1.44 | 1.19 | 1.33 |
| **(± 1σ)** | (0.16) | (0.27) | (0.29) | (0.17) | (0.28) |
| $\overline{Roundness}$ | 0.81 | 0.77 | 0.73 | 0.84 | 0.75 |
| **(± 1σ)** | (0.06) | (0.08) | (0.09) | (0.08) | (0.10) |
| $\overline{Circularity}$ | 0.78 | 0.64 | 0.61 | 0.72 | 0.53 |
| **(± 1σ)** | (0.18) | (0.14) | (0.15) | (0.20) | (0.16) |

| $\overline{d_{pp}}$ (nm) | 41.9 | 35.5 | 34.5 | 34.2 | 45.4 |
|---|---|---|---|---|---|
| (± 1σ) | (12.4) | (9.9) | (11.4) | (9.9) | (13.6) |
| N | 242 | 256 | 343 | 139 | 251 |
| $D_f$ | 2.02 | 1.92 | 1.92 | 2.34 | 2.31 |
| [95% confidence interval] | [1.85, 2.18] | [1.30, 2.53] | [1.68, 2.16] | [2.12, 2.56] | [2.01, 2.61] |
| Median $d_{va}$[e] | - | - | 608.7 | 610.6 | - |
| Effective density (g/cm³)[f] | - | - | 1.52 | 1.44 | - |
| Spectra percentage exhibiting m/z = 16 signal | - | - | 2.8 | 21.0 | - |
| Mean O:C ratio[g] | | | 0.008 | 0.024 | |
| (± 1σ) | - | - | (0.024) | (0.036) | - |

[a]Information offered by manufacturer datasheet. [b]Measured by Nichman et al. (2019) using VWR pH meter. [c]Measured by manufacturer in compliance with ASTM 1512. [d]BET specific surface area measured by manufacturers using $N_2$ adsorption in compliance with ASTM D-4820. [e]Converted from the measured time of flight. [f]Calculated from dividing median $d_{va}$ by $d_m$ (400 nm in this study) and times the reference density 1 g/m³ (Cziczo et al., 2006). [g]Calculated from PALMS spectra area.”

- *L178-179: Form Figs. A2 and A3 it remains ambiguous how dpp was determined. More information should be added.*

[Response]: Added "manually" in the caption as well as a description paragraph after Fig. A4. A table of the coefficients and goodness of primary particle normal distribution fittings are also added (Table A1).

L553-560: "BC aggregate parameters, including $L_{max}$ and $W_{max}$, perimeter, and project area, are determined by manually drawing the periphery of the aggregate, as shown in Fig. A4 (a-e). The primary particle diameter is determined by artificially recognizing the boundary of BC primary particle in the SEM image. After manual selection of the start and end points of a primary particle's length and width respectively, the distance between these points are calculated automatically and recorded. The primary particle diameter equals to the geometric mean of the length and width. Primary particle size distribution of a specific BC type and size can then be obtained by categorizing primary particle diameters into different size bins and count the frequency, as shown in Fig. A5. BC primary particle size distribution obeys normal distribution. The coefficients and goodness of normal distribution fittings are shown in Table A1."

- *L196: Fig. A5 shows a very broad size distribution, even though dm = 400 nm have been sampled. How does dva and dm relate? Please discuss in the manuscript how the broad, not monodisperse size distribution affect your conclusions for the IN measurements. A similar question arises when looking at Fig. 2.*

[Response]: The $d_{va}$ was obtained by calculating particle velocity in PALMS. However, the emperical velocity-$d_{va}$ curve was obtained using PSL particles. For complex particles like BC,

the shape should be taken into account. Omitting shape factors and other factors may introduce bias and lead to wide particle distribution. Kulkarni et al. (2016) also reported larger modal size and broader $d_{va}$ distribution compared with $d_m$.

The discussion was added between L175-179, please see the response to the comment on L150-153.

> o *Please also specify what size threshold was used for the analysis of the SPIN OPC data and discuss this in context of the shown BC size distributions. E.g. Fig. A5 suggest presence of supermicron particles, which could have been miscounted as ice crystals.*

[Response]: We agree with the referee that OPC size threshold is a commonly used criteria to distinguish between ice crystals and aerosols. However, it cannot differentiate water droplets (if there is any) and ice crystals.

Instead, both the laser scattering and polarity signals collected from SPIN OPC were used in this study to train a machine learning algorithm as described in (Garimella et al., 2016), to distinguish different phases, as stated between L305-307.

- *L203: Please add references*

[Response]: Added "More details about the AMS can be found in literatures (DeCarlo et al., 2006; Onasch et al., 2012), here a brief introduction will be given." after the appearance of AMS, and deleted "More details about AMS can be found in literatures (Jayne et al., 2000; Onasch et al., 2012)" at the end of this paragraph.

L232-238: "Chemical composition of the SOA-coated BC particle stream was analyzed online by PALMS and a High-Resolution Time-of-Flight Aerosol Mass Spectrometry (HR-ToF-AMS; Aerodyne Research Inc.). More details about the AMS can be found in literatures (DeCarlo et al., 2006; Onasch et al., 2012), here a brief introduction will be given. The AMS offers quantitative average mass spectrum of an ensemble of aerosols. Particles entering AMS first go through an aerodynamic lens inlet to form a particle beam. A mechanical chopper is used downstream the inlet to control sampling particle or particle free period. The AMS employs a heated 600 °C tungsten surface to vaporize nonrefractory aerosols. Ionization is achieved using a universal 70 eV electron ionization technique. Ionized species are detected by time of flight mass spectrometry."

- *L211: Are these number concentrations after size selection? They seem very high.*

[Response]: These are the number concentrations obtained from bare BC experiments size-selected by BMI DMA. We used high flow rate and high concentration in BC-SOA mixing experiments because the particle concentration could drop to a low level and introduce uncertainties after consecutive size-selection by two DMAs when determining particle number and activation fraction.

- *L222: O3 concentrations of 110 ppm seem very high. Please put this in context of typical tropospheric O3 concentrations. Can this be used to then report estimates for "Equivalent atmospheric exposure days" of O3 in your Table 2?*

[Response]: The typical tropopause $O_3$ mixing ratio ranges between 0.1-1 ppm (Fioletov, 2008; Gettelman et al., 2011). However, the residence time of $\beta$-caryophyllene in the ambient environment is much longer than in the PAM reactor (5 min or less). Hence we used a higher concentration of ozone to expedite the reaction that would normally take longer to react in the ambient, which is common for flow tube or oxidation flow reactor related studies (Lambe et al., 2011b; Zhang et al., 2015).

The equivalent atmospheric exposure days are based on OH concentration, which was calculated combining $O_3$ concentration and UV intensity. The ambient OH concentration was assumed to be $1 \times 10^6$ cm$^{-3}$ (Li et al., 2018) and the OH exposure equivalent time was calculated by using the OH concentration from the PAM and the residence time of the aerosols within the PAM (Zhang et al., 2018a).

Added "The typical $O_3$ mixing ratio in tropopause ranges between 0.1 to 1 ppm (Fioletov, 2008; Gettelman et al., 2011). The ozone concentration in this study was higher than ambient concentration to expedite the reaction given the short residence time of SOA within the PAM reactor (Lambe et al., 2011b; Zhang et al., 2015)." between L252-255

- *L225: Replace "pyrolysis" by "photolysis"*

[Response]: Replaced.

- *L228: Please specify how the reported "atmospheric aging time" of 10-15 days was calculated.*

[Response]: The equivalent atmospheric aging time $t_{eq}$ are calculated using the OH concentration of the PAM reactor $c_{OH\_PAM}$, the residence time of the particles within the PAM, $t$, and the ambient OH concentration $c_0 = 1 \times 10^6$ cm$^{-3}$ (Li et al., 2018). $O_3$ concentration and UV intensity, where the OH concentration of the PAM reactor was calculated based on the $O_3$ concentration, relative humidty (RH), and UV light intensity measured real-time from the PAM reactor (Lambe et al., 2011a; Zhang et al., 2018a). The equation is:

$$t_{eq} = \frac{c_{OH\_PAM} \times t}{c_0}$$

L262-265: "The equivalent atmospheric aging time $t_{eq}$ were calculated using the OH concentration of the PAM reactor $c_{OH\_PAM}$, the residence time of the particles within the PAM, $t$, and the ambient OH concentration $c_0 = 1.0 \times 10^6$ cm$^{-3}$ (Li et al., 2018) following the equation $t = c_{OH,PAM} \times t / c_0$. $c_{OH\_PAM}$ was calculated based on $O_3$ concentration and UV intensity, and RH measured real-time from the PAM reactor (Lambe et al., 2011a; Zhang et al., 2018a)."

- *L232: What causes the increase in RH from 16% to 25%?*

[Response]: *RH* = 25% is the upper *RH* limit during our experiments. We report the limit to give the readers a better understanding of our experiment condition.

The *RH* increases gradually during IN experiments because the silica dryers are degraded by the water content in the atomized sample flow. At the end of experiments each day, the *RH* can reach up to 25% if the ambient humidity is high.

- *L236: Please add what the CPC number concentration was after flushing the PAM.*

[Response]: Added "The particle concentration measured each day before experiments were below 70 # cc$^{-1}$." between L265-266.

- *L259: Delete references to Nichman et al. (2019) and Wolf et al. (2019).*

[Response]: References removed.

- *L263: Specify "centerline" of what?*

[Response]: Added "of SPIN chamber".

L299-300: "Particles fed into SPIN are constrained by a ~9.0 SLPM sheath gas within a lamina near the centerline of SPIN chamber."

- *L276: Specify OPC size threshold used for AF analysis.*

[Response]: Please see the response to the last comment on Table 1.

- *L277: Please elaborate why different correction factors were applied for R2500U and R330 R and comment in the manuscript.*

[Response]: The correction factor to account for aerosol lamina spreading has been extensively characterized in both theory and measurement for the SPIN (Garimella et al., 2017; Wolf et al., 2020). Based on classical fluid dynamics, the turbulence spreading calculation used the spherical particle assumption for simplicity, whose behavior is easier to predict and capture. When it comes to fractal particles, the morphology, dynamic shape factor, and other factors has to be taken into consideration, and the particle behavior are harder to predict. Fractal particles may bear different drag force and not follow the idealized flow trajectory. Therefore, the upper range of correction factors at these SPIN experimental conditions was applied to the more fractal R2500U data, and the lower range of possible correction factors was applied to R330R.

Added "To account for aerosol spreading outside of the lamina where $SS_i$ is the highest (Garimella et al., 2016),… The correction factor was determined by taking the effect of morphology on particle behavior within SPIN lamina into consideration." between L314-317.

L314-317: "We define the IN onset as 1% of particles activating, i.e. $AF = 1\%$, for a period of 10 seconds as activation. To account for aerosol spreading outside of the lamina where $SS_i$ is the highest (Garimella et al., 2016), correction factor of 3.4 and 2.2 were applied for R2500U and R330R (Wolf et al., 2019). The correction factor was determined by taking the effect of morphology on particle behavior within SPIN lamina into consideration."

- *Section 3.1: Please add the fraction of multiple charged particles in your population of size- selected particles to your Table 1 and discuss a potential impact on your reported IN onsets, using AF = 1% as threshold.*

[Response]: We apologize for leaving this detail in our paper. We actually used a 500 nm impactor at the BMI DMA inlet in our bare BC IN experiments to avoid large doubly-charged particles. Theoretically, at least 50% doubly-charged particle should be removed in the bare 300 and 400 nm BC experiments with the impactor. Since we did not have another DMA or CPMA during bare BC IN experiments, we were not able to determine the doubly-charged population for bare BC.

[Figure]

**Figure R2 2. The overall particle size distribution of bare COJ300 BC, and toluene and *β*-caryophyllene SOA coated COJ300 BC. The figure also shows doubly-charged BC particle peaks with modal sizes of ~600 nm.**

However, based on the bare BC data in BC SOA coating experiments illustrated above, the number fraction of doubly-charged bare BC particles with a peak around 600 nm are ~13% to 16% of total bare BC particle number, respectively. Therefore, we assume the doubly-charged BC particles made up of 16% of total size-selected BC population, which is quite conservative since the sheath-to-sample ratio in BMI DMA is higher (~6:1) in bare BC experiments than that (~4:1) for the TSI DMA in BC-SOA mixing experiments.

As shown in Fig. 3 (a) and (c), R2500U and R330R did not exhibit IN activity below 300 nm, so we believe doubly-charged particles would affect the 100 and 200 nm COJ300 results most. If the doubly-charged particles who are more IN active did affect the 100 and 200 nm COJ300 results, the -40 °C $AF$-$SS_i$ activation curve of 100 and 200 nm COJ300 showed below should exhibit a gradual AF increase at $SS_i \approx 1.3$ and grow to about 15%, then with a sudden AF jump to ~100% once $SS_i$ exceeds homogeneous freezing threshold. However, there is no obvious AF growth exceeding 1% at $SS_i \approx 1.3$ in the plot below, and the $SS_i$ approaches homogeneous freezing threshold when the AF approaches 1%. Therefore, we can imply that doubly-charged particles have limited effect on our bare BC IN onset results.

[Figure]

**Figure R2 3. Activation curve as a function of $SS_i$ for 100 nm and 200 nm COJ300 particles at -40 °C**

For clarity, we modified the statement in Sect. 2.2.1 and added "For bare BC experiments, a BMI differential mobility analyzer (BMI DMA, Model 2002; Brechtel Manufacturing Inc.) was used to size-select particles, with a 500 nm cut-off size impactor installed at the DMA inlet to get rid of large particles." between L73-175.

- *L284: "deposition IN"; please see my general comment above.*

[Response]: Deleted "deposition".

- *L286-287: Are the error bars based on theoretical calculations presented in the cited reference or based on measurements; please specify.*

[Response]: The error bars are based on measured temperatures. Added "…derived from experimental data for each panel".

L325-326: "Representative error bars in black lines show one standard deviation of variability for SPIN lamina temperature and $SS_i$ derived from experimental data separately for each panel (Kulkarni and Kok, 2012)."

- *L288: I suggest tuning down "substantially" here. Looking at your morphological parameters in Table 1 (AR, roundness, circularity), the values of the different soot types are within uncertainty of each other. Please see also my general comment above.*

[Response]: Changed to "the three test BC types exhibit different particle morphology." Please also see the response to the comment on L146.

- *L289: Please use the same precision for the given Df values as in Table 1, i.e. Df = 1.92.*

[Response]: Modified as suggested.

L327-328: "400 nm R2500U has the smallest $D_f$ (~1.92), and COJ300 and R330R have larger $D_f$ (~2.34 and 2.31, respectively);"

- *L292: Can this also simply result from uncertainty in the dpp measurements? Please see Anderson et al. (2017).*

[Response]: As stated between L341-342 "The larger $\overline{d_{pp}}$ of 200 nm R2500U (Table 1) might result from the blurring of primary particles under high magnification."

[Figure]

**Figure R2 4. Boundary blurring process of BC primary particles under high magnification SEM electron beam when the observation time became longer. The scale bar applies to all panels.**

During our SEM observation, the BC particles showed clear boundary at first, but after a few seconds of exposure to electron beam, the boundary of primary particles blurred, resulting in larger primary particle diameter. Such blurring was more severe for 200 nm particles that required higher magnification, and thus stronger electron beam intensity, to see the details of BC primary particles clearly. One possible explanation is that the less conductive part of the BC particles gets surface charging under high intensity electron beam. The SEM detectors will get electrons from multiple sources when the charge dissipation rate is low. As a result, the image becomes less clear the longer we scan at the same spot.

Besides, different magnification level (corresponding to different scales and length of scale bars) can also induce uncertainty in post-processing. To find the optimum balance between magnification level and fusion, we tried our best to use the ×30,000 to ×105,000 SEM images for BC aggregate morphology analysis when possible.

Despite that we cannot exclude the uncertainty caused by manual selection of start and end points when determining $d_{pp}$ (please see our response to the comment on L178-179) apart from the magnification level error mentioned above, such uncertainty is different from the ones raised by automated image processing algorithm in (Anderson et al., 2017).

- *Fig. 3:*
  - *Panel A: Why does the temperature uncertainty change between the subpanels (a), (b) and (c)?*

[Response]: Each dot in Fig. 3(A) represents a separate RH scan ramp. To ensure consistency, the IN onsets of a certain BC type was determined from consecutive RH scan ramps in a day. The error bar in Fig. 3 represents the instrument condition during that day calculated following the methodology described in Kulkarni and Kok (2012).

However, the experiments for different BC types were performed on different days. The temperature uncertainty varied slightly between different days.

Besides, we should clarify that the representative uncertainty was calculated using the four coldest thermocouples. In other words, the overall temperature uncertainty is smaller if we take all sixteen thermocouples of SPIN into consideration.

  - *Panel B: Please specify if the scale bar is valid for all images. The SEM image of COJ300 suggest that this soot type is the most spherical. However, the roundness values reported in Table 1 are similar to e.g. dm = 200 nm R2500U. Can the authors provide more SEM images for the Appendix to evaluate the representativeness of the shown images?*

[Response]: Moved panel B to Appendix A, numbered as Fig. A2 and added more SEM images. Fig. A2 includes SEM images of (a) 400 nm R2500U, (b) 300 nm R2500U, (c) 200 nm R2500U, (d) 400 nm R330R, and (e) 400 nm COJ300 with separate scale bars for each image. Changed the figure number in Appendix A accordingly.

200 nm R2500U is quite spherical based on the calculated $D_f$ and SEM images. Therefore, there is no surprise that the shape descriptors such as roundness of 200 nm R2500U is close to that of 400 nm COJ300.

o *Panel B: Considering the values of the roundness of the various soot types and dm = 400 nm, suggest that the morphology is the same within experimental uncertainty. In this context, the conclusion drawn that no clear dependence of IN on the particle morphology was found (e.g. L324, 451) is not surprising. However, the COJ300 soot type seems clearly more spherical on the (typical) images depicted in Fig. 3. Interestingly, at the same time this soot type reveals the largest ice nucleation activity. This trend is in-line with recent results of Mahrt et al. (2020), showing that more compacted, round soot aggregates are better ice nucleation particles (INPs) via PCF compared to more fractal soot particles. This provides further evidence that the observed ice nucleation activity in the present study is caused by PCF and might be worthwhile to consider for the discussion of the results. Do the authors have an explanation why the COJ300 aggregates look almost spherical?*

[Response]: According to the technology pathway in the COJ300 patent (Johnson, 1999), the BC particle surface modification was achieved by acid and other solvents washing. The washing and stirring process might shake off BC chains and cause collapse, which may explain why the COJ300 looks so spherical. However, the manufacturer should be contacted for more details about the BC manufacture and modification technology, which should be their core IP.

o *Panel C: It looks like there is also some O (m/z = 16) for R2500U, also this is hard to see from the scale. This figure could be improved by moving the PALMS spectra to the SI (they are only mentioned once in the manuscript) and increase the size of the panels showing the IN results, which is the major focus of this work.*

[Response]: Moved panel B and C to Appendix A. Please see the response to the comment on the O:C ratio in Table 1 for PALMS spectra interpretation.

- *L308: Add: "… relevant to the particle..:"*
[Response]: Added.

- *L309: "Representative uncertainty" of what?*
[Response]: Added "range of SPIN lamina $SS_i$".

- *L310: "deposition IN"; please see my general comment above.*
[Response]: Deleted "deposition", added "below homogeneous freezing threshold".

L351-353: "The most spherical and oxidized COJ300 BC particles exhibited IN activity below homogeneous freezing threshold regardless of particle size and temperature in this study."

- *L315: Add: "… R330R along the expected homogeneous freezing threshold in Fig…"*
[Response]: Added as suggested.

- *L316: "…where the IN mode transitions from heterogeneous IN ability to homogeneous freezing…" Please consider revising this statement. During PCF, the formation of ice takes place by homogeneous freezing, i.e. PCF should not be viewed as heterogeneous ice nucleation process (Marcolli, 2020).*
[Response]: Rephrased.

L358-360: "We conclude that the lower size threshold where the BC particles exhibit IN activity below homogeneous freezing threshold at thermodynamic conditions relevant to cirrus may well lie between 300-400 nm and 200-400 nm for R2500U and R330R around -46 °C, respectively."

- *L318: "higher" not "warmer" temperature*

[Response]: Changed to "higher".

- *L324-326: You might want to revisit these lines: The number of surface defects and pores for soot particles to form ice via PCF does not necessarily scale with the degree of branching of a soot aggregate. For instance, Mahrt et al. (2020) reported that cloud processes, more compacted (less fractal, even though associated with a higher fractal dimension) soot aggregates were more potent INPs to form ice via PCF. In these more compacted, round soot aggregates the probability of having a pore suitable for PCF is higher. In fact, your results are in-line with these previous findings and further support these. Your Fig. 3 reveals that COJ300 soot is most ice active and also most round based on the SEM image, with the latter being further supported by the roundness and fractal dimension values reported for COJ300.*

[Response]: Reformulated the discussion here.

L366-375: "The IN onset results show no clear dependence on particle fractal level and surface area. Even though the more fractal and branching feature of R2500U BC particles with larger surface area do not clearly exhibit superior IN activity over R330R. Koehler et al. (2009) showed that IN was favored for oxidized hydrophilic BC, but too many hydrophilic active sites may bond water molecules, impeding ice embryo formation and thus impair IN (Pruppacher and Klett, 2010). The surface-modified, highly dispersible and spherical COJ300 with smaller $\overline{d_{pp}}$ shows better IN efficiency than fractal BC, which is consistent with the results of Mahrt et al. (2018) and Nichman et al. (2019) based on PCF mechanism. The physio-chemical properties of COJ300 particles, including oxidized surface, appropriate $\overline{d_{pp}}$ , and compacted spherical morphology, may result in higher probability to have cavities with appropriate size and hydrophilicity on particle surfaces to accommodate liquid water below bulk water saturation by the inverse Kelvin effect (Marcolli, 2014; Koop, 2017; David et al., 2019; Marcolli, 2020)."

- *L331-332: Please expand your explanation on this aspect. What you show in Fig. C1 is not the saturation ratio required for pore filling (or onset of PCF), but the negative pressure resulting from keeping water within the cavity; this needs to be revised. Please also comment on how relevant the pore geometries investigated in Marcolli (2020), i.e. cylindrical and wedge-shaped pores are for the BC agglomerates in your study.*

[Response]: We agree that Fig. C1 shows the liquid saturation pressure **drop** due to the presence of cylindrical and wedge-shaped pores, i.e. the difference between saturation pressure over a concave liquid surface and over a plain.

As for the relevance for the BC aggregates and wedge-shape pores, if we assume the primary particles are point connected, the saddle surface between two spheres can be treated like a small wedge if we observe close enough to the contact point, with the first principal radius be a few nanometers between the spheres and the second principal radius be along the circle. The second

principal radius could be an order of magnitude larger than the first principal radius, and thus can be roughly approximated to be infinite.

[Figure]

**Figure R2 5. Scheme of the wedge-like pore formation between BC primary particles.**

We removed Appendix C and this sentence so that we can focus on the IN data and discussion of morphology, hydrophilicity, etc.

L371-375: "The physio-chemical properties of COJ300 particles, including oxidized surface, appropriate $\overline{d_{pp}}$ , and compacted spherical morphology, may result in higher probability to have cavities with appropriate size and hydrophilicity on particle surfaces (Mahrt et al., 2020). Such cavities can accommodate liquid water below bulk water saturation and initiate homogeneous freezing of liquid water via PCF pathway (Marcolli, 2014; David et al., 2019; David et al., 2020)."

- *Figs. 3 (and 5): Does each dot correspond to a single RH-scan?*

[Response]: Each dot in Fig. 3 and 5 represents a separate RH scan ramp. To ensure consistency, the IN onset of a certain BC type was determined from consecutive RH scan ramps in a day.

- *Sect. 3.2: Can you determine the coating thickness from your measurements. Adding an estimate of the coating thickness to this section would be helpful, in order to estimate for what coating thickness ice nucleation by PCF becomes inhibited.*

[Response]: We can infer a coating thickness of 25 nm from the $d_m$ modal size shift. For detailed surface coverage information, SEM observation would be required, which is our future study scope.

Kulkarni et al. (2016) reported a 120 nm to 280 nm diesel BC modal size shift after $\alpha$-pinene SOA coating. Combining their results and our data together, a very rough estimation can be obtained using $d_m$ modal size shift as evidence: coating thickness of 25 nm for *n*-dodecane and $\beta$-caryophyllene SOA can inhibit PCF of COJ300.

- *L337: "deposition IN"; please see my general comment above*

[Response]: Changed to "ice formation below homogeneous freezing threshold".

L379-380: "IN onset $SS_i$ of pure SOA particles are shown as an asterisk separately to rule out the possible ice formation below homogeneous freezing threshold induced by pure SOA."

- *L346-347: It looks like the onset of toluene SOA-coated BC particles and bare BC particles is still within uncertainty.*

[Response]: We modified the discussion here and attributed the *T-10* results to potential toluene SOA phase state transition.

L389-429: "The toluene SOA mass spectrum in Fig. 6(a) exhibits higher $m/z = 44$ (COO⁻) and lower $m/z = 43$ (C$_3$H$_7$⁻) fraction signal, indicating more oxidized organic species were generated during *T-10* and *T-3* experiments (Lambe et al., 2011b), agreeing with the previous study on toluene SOA (Liu et al., 2018). The higher O/C ratio (Fig. B2) of toluene-derived SOA may enhance the hygroscopicity of the particle (Lambe et al., 2011b; Zhao et al., 2016; Liu et al., 2018) with a potential to form aqueous film on BC surface and reduce the deposition IN ability of BC particles. On the other hand, Hinks et al. (2018) showed that toluene-derived SOA contained a significant amount of oligomers under dry laboratory conditions similar to what we conducted in the PAM chamber in this study. These large oligomers could potentially reduce the hygroscopicity and alter the phase state of toluene SOA to be semi-solid or solid within the temperature range we investigated in this work (DeRieux et al., 2018; Zhang et al., 2018c; Li et al., 2020), under which the SOA can still nucleate ice (Murray et al., 2010; Berkemeier et al., 2014; Zhang et al., 2019b). The toluene SOA in *T-10* with O:C ratio over 1 (Fig. B2) has most likely already transited into solid or semi-solid glassy state at the temperature range we investigated before entering SPIN (DeRieux et al., 2018). Therefore, it is very likely that BC particles are mostly sticked with or embedded in these glassy SOA with some bare BC part exposure. Ice crystals may therefore form on the carbonaceous part of partially coated particles, whose IN onset $SS_i$ should be the same as bare COJ300. At temperatures above -43 °C, toluene SOA-coated BC particles nucleate ice at $SS_i$ ~0.1 to 0.15 above bare 350 nm COJ300, but still ~0.15 below the homogeneous freezing threshold. This might be due to the hygroscopicity enhancement of the toluene SOA-coated BC in *T*-10. BC coated by ~10-15 equivalent days atmospherically oxidized toluene SOA (*T-3* in Table 2) is less IN active than those coated by highly oxidized toluene SOA (*T-10*) at around -46 °C and -43 °C. This might also be attributed the oxidation level and the corresponding phase state difference between the SOA generated from *T-10* and *T-3* experiments, which is beyond the scope of this study and requires further detailed phase transition study for toluene SOA. Overall, two competing effects, i.e. the hygroscopicity enhancement deactivating BC IN ability, together with toluene SOA glass phase transition producing sticked or partly coated BC, may make our toluene SOA coating IN onset move towards but not fully in the homogeneous freezing regime. The toluene SOA coated diesel combustion BC (Kulkarni et al., 2016), however, nucleate ice near the homogeneous threshold, as indicated in Fig. 3(a). This might be due to the much thicker (90 nm) organic coating in their study compared to the 25 nm coating of this work, leading to a complete coverage of BC surface."

- *L349: 10-15 days is longer than the average tropospheric lifetime of soot particles. This should be reflected in your discussion.*

[Response]: We agree with the referee that the typical life time of BC particles in the troposphere can be days to weeks (Cape et al., 2012; Lund et al., 2018). However, the life time of BC particles in the tropopause can be extended to months (Pusechel et al., 1992; Yu et al., 2019).

- *Fig. 6: Showing difference spectra between each panel and the bare soot shown in Fig. 6d would facilitate showing the differences between the coatings.*

[Response]: Added BG-10 for each plot, denoted as grey bars to show the difference clearly.

- *L369: Rephrase to: "The IN onset…" … "… in Fig. 5b show that these particles nucleate ice …"*

[Response]: Rephrased as suggested.

- *L373: Can you provide a difference AMS spectra of the OH and the O3 oxidized β-caryophyllene coated BC particles?*

[Response]: Added B-0 spectra in Fig. 6(c3).

- *L384: "form" instead of "forms"*

[Response]: Modified.

- *Fig. 7:*
    - *It would be helpful to add the H/C and the O/C values of the O3 oxidized β-caryophyllene soot here as well.*

[Response]: Added. Please see our response to the next comment.

    - *Please verify that the suggested slope for the toluene coated SOA is correct. From your legend it looks like you should connect the white start in the lower left. corner to the blue star in the middle right. If that is the case, you need to revise your statement in L378-380.*

[Response]: We gratefully thank the referee for making us revisit this figure and think deeper.

We noted that the O/C and H/C ratio in Fig. 7 is closer to glyoxal SOA rather than toluene SOA (Lambe et al., 2011b), which is also true for the $f_{44}.f_{43}$ plot in modified Fig. B2. It is reported that toluene photolysis can produce glyoxal (Volkamer et al., 2001), which may lead to oligomerization under dry condition (Hinks et al., 2018). The slopes between pure toluene (hollow symbols at lower left) and T-10 and T-3 are 0.71 and 0.99 in Fig. 7, respectively. This suggests that benzene ring or carbon-carbon double bond breakage accompanied by oxygen and hydrogen atoms addition might have happened during the experiments, which is compliant with the glyoxal formation pathway in Volkamer et al. (2001). Therefore, we believe that the toluene SOA in *T-10* and *T-3* experiments represent the highly oxidized late generation products with oligomer formation.

The statement between L438-440, however, represents early stage OH oxidation reactions for *n*-dodecane and *β*-caryophyllene. OH oxidation of toluene has already passed the stage and the SOA in *T-10* and *T-3* experiments can be treated as oxidization products of toluene SOA.

Fig. 7 was modified by adding lines connecting pure toluene and *T-10* and *T-3*, and a line connecting pure *β*-caryophyllene and B-0, respectively.

[Figure]

**Figure R2 6 Modified Fig. 7**

Added "Volkamer et al. (2001) proved that glyoxal, which could facilitate the oligomerization as described by Hinks et al. (2018), can be produced from toluene reaction with OH as highly oxidized products. The slopes of T-10 and T-3 in Fig. 7 lie between 0-1 (0.71 and 0.99 for T-10 and T-3, respectively), suggests that oxygen and hydrogen atom addition accompanied by carbon-carbon double bond as well as benzene ring breakage might have happened during the toluene photooxidation experiments." between L407-411.

L404-416: "The higher O/C ratio (Fig. 7) of toluene-derived SOA may enhance the hygroscopicity of the particle (Lambe et al., 2011b; Zhao et al., 2016; Liu et al., 2018) with a potential to form aqueous film on BC surface and reduce the IN ability of BC particles. On the other hand, Hinks et al. (2018) showed that toluene-derived SOA contained a significant amount of oligomers under dry laboratory conditions similar to what we conducted in the PAM chamber in this study. Volkamer et al. (2001) proved that glyoxal, which could facilitate the oligomerization as described by Hinks et al. (2018), can be produced from toluene reaction with OH as highly oxidized products. The slopes of T-10 and T-3 in Fig. 7 lie between 0-1 (0.71 and 0.99 for T-10 and T-3, respectively), suggests that oxygen and hydrogen atom addition accompanied by carbon-carbon double bond as well as benzene ring breakage might have happened during the toluene photooxidation experiments. These large oligomers could potentially reduce the hygroscopicity and alter the phase state of toluene SOA to be semi-solid or solid within the temperature range we investigated in this work (DeRieux et al., 2018; Zhang et al., 2018c; Li et al., 2020), under which the SOA can still nucleate ice (Murray et al., 2010; Berkemeier et al., 2014; Zhang et al., 2019b). The toluene SOA in *T-10* with O:C ratio over 1 (Fig. 7) has most likely already transited into solid or semi-solid glassy state at the temperature range we investigated before entering SPIN (DeRieux et al., 2018)"

- *L392: Add: "In these studies shifts from…"*

[Response]: Rephrased.

- *L394-396: I do not think that it is the volatility that changes and allows a SOA particle to become glassy, but the phase state (here viscosity). Please reformulate.*

[Response]: Changed "volatility" to "phase state".

- *L396-398: If the suppression of the ice nucleation results from filling of the pores, the ice nucleation mechanism of the bare/uncoated particles would likely be best described by PCF, not? Please see my general comment above.*

[Response]: Yes, you are right.

L458-460: "The suppression of BC IN ability by organic coating was attributed to coverage of surface-active sites and filling of pores on BC surface when the volatility of the organic coating is relatively high and might present in liquid phase."

- *L401: Please give details about the mass loadings used in the PAM and briefly comment on typical tropospheric SOA mass loadings.*

[Response]: Change to "…with 200 to 4000 folds of typical tropospheric SOA (Tsigaridis and Kanakidou, 2003; Heald et al., 2008; Hodzic et al., 2016) mass loading in PAM chamber (~2000 to 4000 $\mu$g m$^{-3}$)…"

L462-465: "Our results suggest that less oxidized SOA (*n*-dodecane and *β*-caryophyllene derived SOA from photooxidation), with 200 to 4000 folds of typical tropospheric SOA (Tsigaridis and Kanakidou, 2003; Heald et al., 2008; Hodzic et al., 2016) mass loading in PAM chamber (~2000 to 4000 $\mu$g m$^{-3}$), are more likely to condense on seed particle and form fully coated BC particles, moving IN onset $SS_i$ to the homogeneous regime…"

- *L412-415: This discussion should be expanded and more specific examples are needed to make the claim that the characteristics of the investigated BC types is similar to ambient soot.*

[Response]: Reformulated and added specific literature examples.

L474-485: "The morphological characteristics are within the value range of typical BC emitted from combustion sources, and those collected in field observation (e.g., Lapuerta et al., 2007; China et al., 2013; China et al., 2014; Vander Wal et al., 2014; China et al., 2015b; Zhang et al., 2019a). BC primary particle size range in this study lies between 10 to 70 nm with a modal size around 25 to 40 nm,  being consistent with previous primary particle studies on combustion BC (e.g, Smekens et al., 2005; Liati et al., 2016; Joo et al., 2018). Previous field observation of transportation emission and biomass burning reported that ambient BC occupied $d_a$, circularity and roundness in the range of 130 to 940 nm, 0.19 to 0.55 and 0.32 to 0.6, respectively (China et al., 2013; China et al., 2014; China et al., 2015a; China et al., 2015b), overlapping with the range in this work. R2500U is similar to the fresh BC emitted from B737 at medium power burning conventional jet fuel in terms of morphology characteristics (Vander Wal et al., 2014). The primary particle size is consistent with BC emitted from prevalent gas turbine engines (Huang and Vander Wal, 2013). Findings in this study can be relevant to airborne aircraft emissions and ground emissions carried by updrafts to tropopause."

- *L419: The reported threshold of "dm < 200 nm" seems rather high and is misleading. For instance, Moore et al. (2017) report mode sizes of around 30 nm. While Kittelson*

*(1998) report soot aggregates up to 200 nm (see their Fig. 11) a more profound literature search should be done here to support the given threshold.*

[Response]: Rephrased to "…which are generally fractal and smaller than 200 nm with modal size ranging from 20 to 100 nm (Kittelson, 1998; Wey et al., 2006; Anderson et al., 2011; Wang et al., 2016; Moore et al., 2017; Awad et al., 2020)."

We meant to say that freshly emitted BC particles from transportation are typically smaller than 200 nm.

L488-490: "This is important for freshly emitted BC from aircraft engines and ground transportation, which are generally fractal and smaller than 200 nm with modal size ranging from 20 to 100 nm (Kittelson, 1998; Wey et al., 2006; Anderson et al., 2011; Wang et al., 2016; Moore et al., 2017; Raza et al., 2018; Awad et al., 2020)."

- *L423: Change to: "However, IN ability of small BC particles… may collapse forming PCF favoring…"*

[Response]: Changed as suggested.

- *L424-425: Why do you mention "surfaces" here and then talk about the "mesopores" in the next sentence.*

[Response]: Reorganized the sequences. Please see the response to next comment.

- *L425: The connection between the primary particle size and the mesopores is unclear and should be further elaborated.*

[Response]: Rephrased and emphasis the importance of $d_{pp}$.

L494-499: "Apart from the most spherical morphology, the smaller $d_{pp}$ of COJ300 may also offer higher probability to form smaller mesopores with appropriate size to accommodate ice crystal formation below water saturation, with particles down to 100 nm can act as efficient INP. Pores formed by BC with larger $d_{pp}$ might be too wide to accommodate liquid water at our experimental conditions. Besides, the COJ300 IN results imply that ice crystal formation may favor oxidized hydrophilic surfaces, confirming the importance of surface hydrophilicity for pore filling in PCF mechanism (David et al., 2019; David et al., 2020)."

- *L426: I suggest tuning down: "This suggest that long-lived…INPs."*

[Response]: Changed as suggested.

- *L433: Specify as: "… growth, SOA-coated soot particles…"*

[Response]: Specified as suggested.

- *L441: Delete "monodisperse", your Fig. 2 indicates a more polydisperse sample.*

[Response]: Deleted.

- *L444: Why is the R330R sample "atmospheric compacted"? The COJ300 is the most round according to your Fig. 3B and Table 1.*

[Response]: It is correct that COJ300 is the most spherical one. However, R330R is also relatively spherical based on Fig. A3 and Table 1, even though not as spherical as COJ300. With its modified surface, we prefer to refer COJ300 as "atmospheric compacted and oxidized", because oxidation may also modify the BC particle surface and make it more spherical.

Changed the description of COJ300 to "atmospheric compacted and oxidized".

L517-519: "…freshly emitted (R2500U), atmospheric compacted (R330R), and atmospheric compacted and oxidized (COJ300)."

- *L449: "deposition nucleation"; please see my general comment above. I think that in particular in the summary section you need to be careful on how to describe the ice nucleation mechanism, as in the same section (L456 and L461), you imply some ice formation via PCF.*

[Response]: Replaced the terms accordingly.

L522: "The onset of some IN below homogeneous freezing threshold, as opposed to …"

- *L451-452: See my earlier comment. The study by Mahrt et al. (2020) found more compacted soot particles to be better INP (via the PCF mechanism) compared to less compacted soot.*

[Response]: Rephrased.

L523-525: "We conclude that BC IN favors larger, spherical particles with oxidized hydrophilic surface. The highly fractal BC particles did not necessarily act as superior deposition INP over more spherical ones as would normally be anticipated from surface area. This could be attributed to PCF occurring in the pores and cavities with appropriate size offered by compacted BC particles."